# Coupling metabolomics and exome sequencing reveals graded effects of rare damaging heterozygous variants on gene function and human traits

Nora Scherer [1,2], Daniel Fässler [3], Oleg Borisov [1], Yurong Cheng[1], Pascal Schlosser [1,4,5], Matthias Wuttke [1,6], Stefan Haug[1], Yong Li [1], Fabian Telkämper[7], Suraj Patil[1,2,6,8], Heike Meiselbach [9], Casper Wong[10], Urs Berger [7], Peggy Sekula [1], Anselm Hoppmann[1], Ulla T. Schultheiss [1,6,11], Sahar Mozaffari [10], Yannan Xi[10], Robert Graham [10], Miriam Schmidts [5,12], Michael Köttgen [5,6], Peter J. Oefner[13], Felix Knauf[14], Kai-Uwe Eckardt [9,14], Sarah C. Grünert[12], Karol Estrada[10], Ines Thiele[15,16,17,18], Johannes Hertel [3,19] ✉ & Anna Köttgen [1,4,5] ✉

Genetic studies of the metabolome can uncover enzymatic and transport processes shaping human metabolism. Using rare variant aggregation testing based on whole-exome sequencing data to detect genes associated with levels of 1,294 plasma and 1,396 urine metabolites, we discovered 235 gene–metabolite associations, many previously unreported. Complementary approaches (genetic, computational (in silico gene knockouts in whole-body models of human metabolism) and one experimental proof of principle) provided orthogonal evidence that studies of rare, damaging variants in the heterozygous state permit inferences concordant with those from inborn errors of metabolism. Allelic series of functional variants in transporters responsible for transcellular sulfate reabsorption (SLC13A1, SLC26A1) exhibited graded effects on plasma sulfate and human height and pinpointed alleles associated with increased odds of diverse musculoskeletal traits and diseases in the population. This integrative approach can identify new players in incompletely characterized human metabolic reactions and reveal metabolic readouts informative of human traits and diseases.

A complex interplay of thousands of enzymes and transport proteins is involved in maintaining physiological levels of intermediates and end products of metabolism. Disturbances of their function can result in severe diseases, such as those caused by inborn errors of metabolism (IEMs), or predispose to common metabolic diseases such as type 2 diabetes or gout. While the study of rare, early-onset, autosomal recessive IEMs has uncovered many metabolite-related genes, such studies are limited by the very low number of persons homozygous for the causative variants. Conversely, genome-wide association studies (GWASs) in large populations have revealed thousands of common genetic variants associated with altered metabolite levels[1–13], but these variants' functional effects are often unknown, and their modest effect sizes limit their direct clinical impact.

Gene-based aggregation testing of rare, putatively damaging variants in population studies can address this challenge. Previously, such studies have focused almost exclusively on the circulating

**Fig. 1 | Overview of the study design.** Schematic representation of the gene-based rare variant aggregation study with plasma and urine metabolite levels using WES data of 4,737 participants from the GCKD study and their follow-up analyses. MAF, minor allele frequency; PC, principal component; PheWAS, phenome-wide association study.

metabolome[14–20]. We have shown recently that GWASs of paired plasma and urine metabolomes do not only reveal many more associations but also enable specific insights into renal metabolite handling[2]. We therefore aimed to perform gene-based testing of the aggregate effect of rare variants on the levels of 1,294 plasma and 1,396 urine metabolites quantified from 4,737 participants in the German Chronic Kidney Disease (GCKD) study with whole-exome sequencing (WES) data to identify metabolism-related genes and to understand whether the underlying rare, almost exclusively heterozygous variants permit inferences complementary to the ones obtained from the study of IEMs.

Patients with IEMs typically show severe symptoms that originate from accumulation or depletion of metabolites, while heterozygous carriers of the causative variants often show milder changes of the same or related metabolic phenotypes[21]. We hypothesized that sex-specific analysis of metabolite-associated, X chromosomal genes as well as knowledge-based, computational modeling based on sex-specific organ-resolved whole-body models (WBMs[22]; Methods) of human metabolism can inform on whether heterozygous damaging variants capture the metabolic effects of their unobserved homozygous counterparts. WBMs enable the investigation of homozygous gene defects through deterministic in silico knockout modeling. The resulting virtual IEMs reflect observed IEMs[22–25]. We further hypothesized that metabolite-associated rare variants identified in the GCKD study would show associations with related human traits and diseases in very large population studies and that the genetic effects would be proportional to their effects on metabolite levels if the implicated metabolites are molecular readouts of disease-relevant processes. The large UK Biobank (UKB) with WES data and extensive health record

linkage permits the systematic study of the aggregated and individual effects of rare, damaging, metabolite-associated variants on a wide variety of traits and diseases.

Here, we set out to perform gene-based rare variant aggregation testing to discover genes associated with metabolite levels and to characterize their genetic architecture with respect to the identified variants and across plasma and urine. We validate identified genes and variants and the range of their effects through complementary genetic approaches, with a new computational method based on WBMs[22,23] and through proof-of-principle experimental studies, and identify traits and diseases for which these metabolites represent molecular readouts.

## Results

As summarized in Fig. 1, rare, putatively damaging variants were identified in 16,525 genes based on WES data from 4,737 GCKD study participants (mean age of 60 years, 40% women; Supplementary Table 1). Metabolites were determined by nontargeted mass spectrometry and covered a wide variety of superpathways (Metabolon HD4 platform; Supplementary Table 2). Exome-wide burden tests for the association between each gene and the levels of each of 1,294 plasma and 1,396 urine metabolites (781 overlapping) were carried out using two complementary 'masks' that differed in the selection of qualifying variants (QVs; Methods) for gene-based aggregation. While the 'LoF_mis' mask contained a median of eight QVs per gene predicted to be either high-confidence loss-of-function (LoF) variants or deleterious missense or in-frame nonsynonymous variants, the 'HI_mis' mask contained a median of 16 QVs per gene predicted

as high-impact consequence (transcript ablation or amplification, splice acceptor or donor, stop-gain, frameshift, start or stop lost) or as deleterious missense variants using additional prediction scores (Methods). Both masks assume a LoF mechanism but account for different genetic architectures.

## Discovery of 192 significant gene–metabolite associations

We identified 192 significant gene–metabolite pairs across both plasma ($P$ value < $5.04 \times 10^{-9}$) and urine ($P$ value < $4.46 \times 10^{-9}$), where 43 associations were detected in both (192 + 43 associations overall; Fig. 2a and Supplementary Table 3). These involved 73 unique genes and 179 metabolites, with a comparable number of genes and metabolites identified in plasma and urine. There were 22 and 17 genes with significant associations exclusively in plasma and in urine, respectively. While the majority of associations was detected with both masks, the more inclusive 'HI_mis' mask yielded more mask-specific associations than the 'LoF_mis' mask (Fig. 2b). Amino acids and lipids were the dominating pathways among the associated metabolites (Supplementary Fig. 1). The higher proportion of implicated lipids in plasma than in urine is consistent with the absence of glomerular filtration of many lipids (Fig. 2b). Associations detected in both plasma and urine generally affected the levels of the implicated metabolite in the same direction (Fig. 2a). Sensitivity analyses evaluating additional masks and methods for aggregation testing (LoF only, sequence kernel association test (SKAT) and SKAT-optimal unified test (SKAT-O)) as well as sex-stratified and kidney function-stratified analyses supported the robustness of the main findings (Supplementary Results, Extended Data Figs. 1–3 and Supplementary Tables 4 and 5).

Previous independent studies of associations between sequencing-based rare variants and metabolite levels obtained using comparable technology have focused on plasma and serum[14,15,19,20]. Comparison of the 128 discovered gene–plasma metabolite associations in this study with previous studies[14,15,19,20] showed that 69% (88 of 128) were not reported previously, although 93% (82 of 88) of the new findings involved metabolites analyzed before (Supplementary Table 6; detailed description in the Supplementary Methods and the Supplementary Results).

The 73 unique metabolite-associated genes were strongly overrepresented among genes known to be causative for IEMs (odds ratio = 10.6, $P$ value = $1.9 \times 10^{-14}$; Supplementary Methods), with 28 (38%) of them currently known to harbor causative mutations (Supplementary Table 6). The QVs detected in our study of middle-aged and older adults were almost exclusively observed in the heterozygous state (Supplementary Data 1). Detailed annotation of QVs in the two masks (Supplementary Table 7) showed that 63 unique QVs in 15 genes and 73 unique QVs in 17 genes were listed in ClinVar as 'pathogenic' or 'pathogenic or likely pathogenic' for a corresponding monogenic disease. These observations support the notion that gene-based aggregation of rare, heterozygous, putatively damaging variants effectively identifies gene–metabolite relationships implicated in human diseases.

## Validation through independent, complementary approaches

Independent replication of our findings is complicated by differences in QVs, metabolite quantification methods and different analytical choices across studies. We therefore validated our findings using four complementary approaches: first, the large UKB permitted analysis of the same rare QVs using the same analytical choices (Methods), as in our study for two overlapping metabolites, and showed very similar effect sizes for gene–metabolite associations (Fig. 3a). Second, the UKB proteomics data[26] contain information on circulating levels of the encoded proteins of 17 genes implicated in our study. Burden tests aggregating protein-truncating and rare damaging variants revealed associations with lower levels of 15 of these proteins (in *cis*, $P$ value < $1 \times 10^{-5}$; Fig. 3b)[27], potentially explained by nonsense-mediated decay. Third, comparison of our findings to those from a previous study of the plasma metabolome[15] showed highly correlated effect sizes with those from our study, both on the variant level and the aggregated level (Spearman correlation coefficient > 0.8; Fig. 3c,d and Supplementary Table 8).

Lastly, we performed a proof-of-concept experimental validation study for an implicated gene–metabolite relationship. The $B^0AT1$ transporter, encoded by *SLC6A19*, is responsible for the uptake of neutral amino acids across the apical membrane of intestinal and kidney epithelial cells[28]. In addition to associations with the levels of the known substrates asparagine, histidine and tryptophan, we also detected associations with methionine sulfone, not yet reported as a substrate. Transport studies in CHO cells overexpressing human SLC6A19 and its co-chaperone collectrin (CLTRN) in comparison to the control indeed confirmed methionine sulfone to be a substrate of the transporter in vitro, in a similar concentration range as its known substrate isoleucine (Fig. 4a and the Methods). Specificity was shown by complete inhibition of transport activity upon application of the SLC6A19 inhibitor cinromide[29] (Fig. 4b). Together, these four complementary lines of evidence all support the validity of the detected associations.

## Prioritization and characteristics of driver variants

We next performed a forward selection procedure[15] to assess the contribution of individual QVs to their gene-based association signals (Methods). Plots that visualize the association $P$ value based on the successive aggregation of the most influential QVs (Supplementary Data 2) revealed noteworthy differences across genes and metabolites, with examples detailed in the Supplementary Results.

The inclusion of effectively neutral variants among the QVs may dilute their joint signal. We thus prioritized the variants with the strongest individual contributions that resulted in the lowest possible association $P$ value when aggregated for burden testing[15] as 'driver variants' (Methods). For each significant association signal, we identified at least two and up to 48 driver variants (median of 13; Supplementary Data 2 and Supplementary Tables 3 and 7). The proteins encoded by the vast majority of identified genes are directly involved in the generation, turnover or transport of the associated

---

**Fig. 2 | Overview of the 192 identified gene–metabolite associations across plasma and urine and their corresponding pathways. a**, Significant associations with plasma metabolites are shown on the outermost band (red; shading reflects effect direction), with genes ordered by chromosomal location across the genome. Associations with urine metabolites are shown on the middle band (blue; shading reflects effect direction). Gene–metabolite associations are based on rare variant aggregation testing from both masks. The ones labeled in gray were already reported in previous rare variant studies[14–20,24], whereas the ones labeled in black are considered new. White spaces indicate that no significant association was detected in a given matrix. For all associations detected in both matrices, effect directions are consistent. The inner band represents the superpathway of the associated metabolite. GPE, glycero-3-phosphoethanolamine; GPI, glycosylphosphatidylinositol; DC, dicarboxylic

acid. **b**, The UpSet plot shows the number of identified gene–metabolite associations by mask and matrix, color coded by the respective metabolite superpathway. Right, horizontal bar plot represents the total number of associations identified by mask and matrix. The proportion of lipids is markedly higher among associations detected with plasma metabolites as compared with urine. Top left, vertical bar plot shows the number of shared associations by mask and matrix, while the sets among which the associations are shared are indicated below each column. While the majority of associations are detected by both masks, especially the less-stringent HI_mis mask provides many mask-specific findings in both plasma and urine. The group of metabolites detected in both plasma and urine is dominated by amino acids. Part. charact., partially characterized.

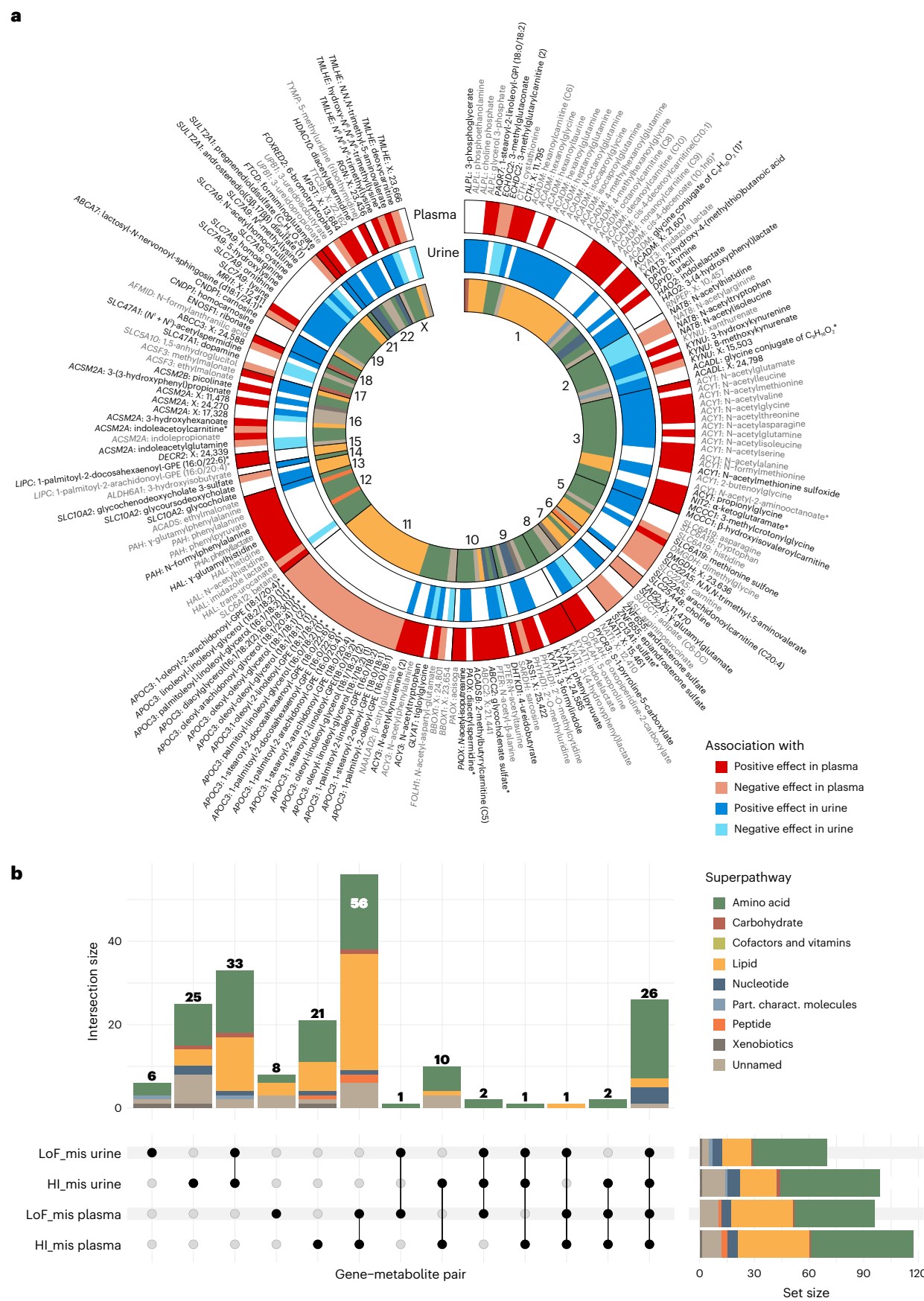

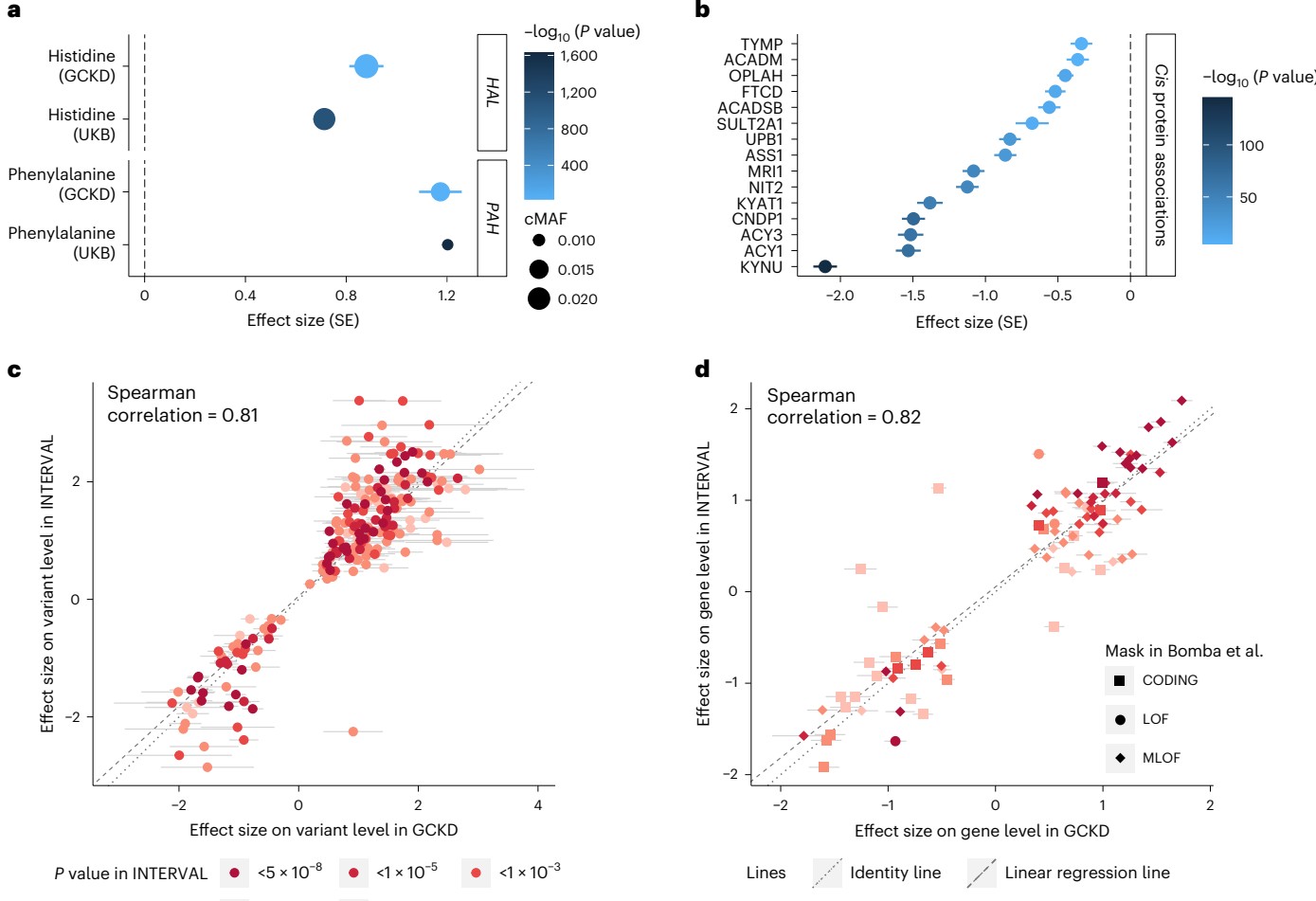

**Fig. 3 | Independent validation of findings using orthogonal approaches.**
**a**, Gene-based testing of significantly associated, available plasma metabolites among ≥261,661 UKB participants (*y* axis) using the same mask and only including QVs available in both the GCKD study and the UKB provided very similar effect sizes in the two studies (*x* axis). Bars represent corresponding standard errors (SE), symbol color reflects the −log10 (*P* value), and the size depicts the cumulative minor allele frequency of all QVs within a gene. In the UKB, gene-based burden tests were performed as implemented in REGENIE (Methods). cMAF, cumulative MAF. **b**, Plasma levels of the proteins encoded by 17 of the 73 significant, metabolite-associated genes were measured in the UKB (*N* ≥ 44,108)[27]. Among the available gene-level summary statistics[27], 15 genes showed *cis* associations with plasma protein levels with an association *P* value < 1 × 10[−5] based on a 'ptvraredmg' mask, which is similar to the masks used in the GCKD study. Genes are shown on the *y* axis. Effect sizes and the corresponding standard errors of the *cis* associations with plasma protein levels based on these summary statistics[27] are displayed on the *x* axis. Symbol color reflects the −log10 (*P* value). For all *cis* associations, the direction of effect sizes was negative, consistent with LoF variants resulting in lower plasma protein levels. **c**, Single-variant effect sizes on levels of a given plasma metabolite in the GCKD study (*x* axis) were very similar to those in the INTERVAL study (Bomba et al.[15]) (*y* axis) for all QVs involved in

significant gene–metabolite associations in the GCKD study that were also available in the INTERVAL study and showed an association *P* value < 0.1 in both studies. Horizontal bars indicate standard errors of effect sizes in the GCKD study (not available for the INTERVAL summary statistics). The depicted 200 associations involved 35 unique genes and 75 unique metabolites. Symbol color indicates the INTERVAL association *P* value. Gray lines represent identity (dotted) and the linear regression line (dashed). The strong correlation of effect sizes supports reproducibility. **d**, Summary statistics for 89 of 128 plasma gene–metabolite associations were available in the INTERVAL study (Bomba et al.). Effect sizes of gene–metabolite associations on the aggregated level in the GCKD study (*x* axis) were very similar to those in the INTERVAL study (*y* axis; Methods and Supplementary Table 8), despite differences in masks and aggregation unit. Horizontal error bars indicate the standard errors of effect sizes in the GCKD study. Standard errors were not available in the summary statistics from Bomba et al. Symbol shape indicates the corresponding mask used by Bomba et al. (LOF, high-confidence LoF variants; MLOF, LoF and missense variants combined; CODING, all rare exonic variants, splice sites and variants residing in untranslated regions). Color reflects the association *P* value in the INTERVAL study. Gray lines represent the identity (dotted) and the linear regression line (dashed).

metabolite(s). It is therefore a reasonable assumption that truly functional variants are those with the strongest individual contributions to the association signal with the implicated metabolite. Indeed, the minimum association *P* value based on only driver variants was often many orders of magnitude lower than the one obtained from all QVs, as exemplified by *DPYD* and plasma uracil (Supplementary Data 2). As expected, the proportion of splice, stop-gain and frameshift variants was higher among driver QVs, whereas nondriver QVs contained a greater proportion of missense variants (Fisher's exact test,

*P* value = 1.3 × 10[−6]; Extended Data Fig. 4a). The median effect of driver variants on metabolite levels increased from missense over start/stop lost, frameshift and stop-gain variants to variants predicted to affect splicing (Extended Data Fig. 4b). The median effect of drivers also increased with lower minor allele count and differed substantially from the one of nondrivers in each minor allele count bin (Extended Data Fig. 4c).

Lastly, evaluation of the convergence of rare and common variant association signals showed that the associations of rare and

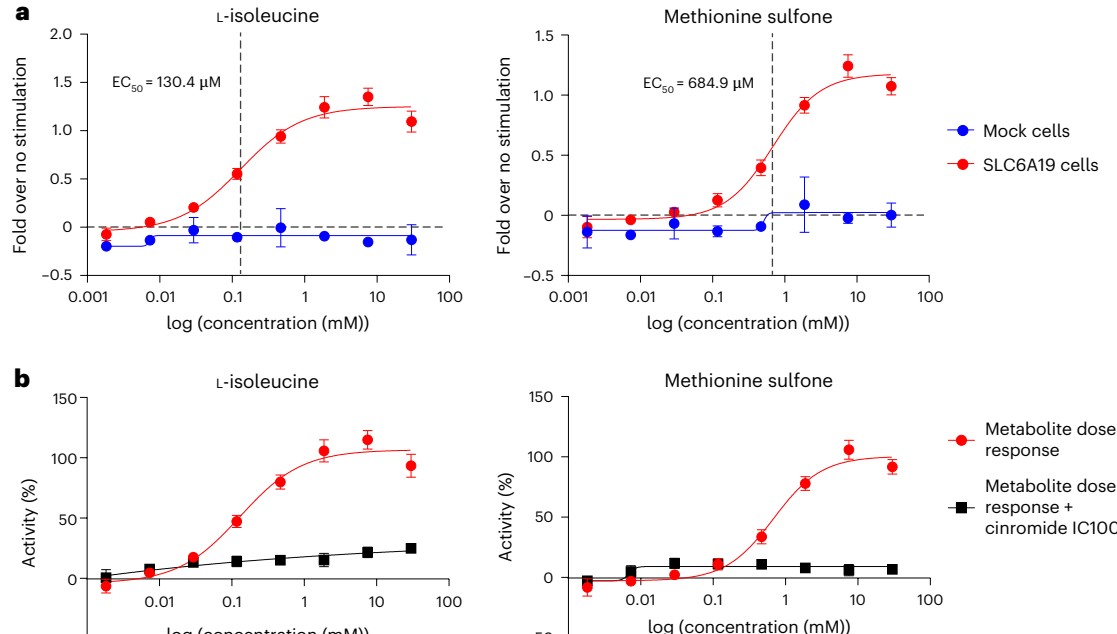

**Fig. 4 | Methionine sulfone is a direct SLC6A19 substrate in vitro. a**, Transport of L-isoleucine and methionine sulfone in CHO cells overexpressing SLC6A19 (SLC6A19 cells) and its chaperone CLTRN versus mock cells. Substrate transport activity was measured by fluorescence increase with a membrane potential dye. The x axis represents increasing concentrations of substrate, and the y axis represents fold over no-substrate-driven signal. Data were generated from four biological replicates and are represented as mean ± s.d. EC₅₀, half-maximum effective concentration. **b**, Effect of cinromide on transport of L-isoleucine and methionine sulfone in cells overexpressing SLC6A19 and CLTRN. L-isoleucine and methionine sulfone transport was abrogated by cinromide, a specific inhibitor of SLC6A19. The x axis represents increasing concentrations of substrate, and the y axis represents activity of maximal substrate-driven signal. Data were generated in the same experiment described in **a**.

common variants in the same region with a given metabolite were independent (Supplementary Results, Supplementary Table 9 and Extended Data Fig. 5).

### Heterozygous variants inform about dose–response effects

The identification of known IEM-causing variants such as in *CTH*, *PAH*, *SLC6A19* and *SLC7A9* (Supplementary Table 7) in the heterozygous state supports the notion that heterozygous QVs are functional alleles that lead to more extreme metabolic changes when present homozygously. For three genes with a homozygous QV present in more than one individual in our study, homozygous individuals tended to have more extreme metabolite levels than heterozygous ones (Extended Data Fig. 6), supporting a dose–response effect. Moreover, we had previously confirmed experimentally that heterozygous sulfate-associated QVs in *SLC26A1* detected by aggregate variant testing are indeed LoF alleles and that the encoded protein is an important player in human sulfate homeostasis[30]. However, experimental studies of each of the 2,077 QVs and 73 genes detected here are infeasible, and IEMs are so rare that no homozygous person for a given gene may have been observed yet. We therefore used three orthogonal approaches: examination of hemizygosity, in silico knockout modeling and investigation of variants prioritized through allelic series, to evaluate whether the observed metabolite-associated heterozygous variants captured similar information about a gene's function as might be derived from homozygous damaging variants in the respective gene.

### X chromosomal genes as a readout of variant homozygosity

Genes in the non-pseudo-autosomal region of the X chromosome offer an opportunity to study differences between heterozygous women and effectively homozygous (that is, hemizygous) men. We therefore investigated sex differences for the two X chromosomal genes identified in our screen, *TMLHE* and *RGN* (Supplementary Table 10).

Indeed, male carriers of QVs in *TMLHE* showed clearly higher urine levels of $N^6,N^6,N^6$-trimethyllysine, the substrate of the encoded enzyme trimethyllysine dioxygenase, than female carriers as well as markedly lower levels of its product hydroxy-$N^6,N^6,N^6$-trimethyllysine, especially when focusing on driver variants (Fig. 5 and Supplementary Table 10). In plasma, male QV carriers showed 1.15 s.d. lower levels of plasma hydroxy-$N^6,N^6,N^6$-trimethyllysine than noncarriers ($P$ value = $6 \times 10^{-44}$), whereas female QV carriers only showed 0.45 s.d. lower metabolite levels than noncarriers ($P$ value = $3 \times 10^{-4}$). A similar tendency was observed for *RGN* and urine levels of the unnamed metabolite X-23436. Levels were higher among both male and female carriers (Supplementary Table 10), suggesting that X-23436 is a metabolite upstream of the reaction catalyzed by the encoded regucalcin. Data from the GTEx Project[31] show no sex differences in gene expression across tissues. Hence, sex-differential effects on metabolite levels likely represent a dose–response effect resulting from heterozygosity versus hemizygosity of the involved QVs.

### Virtual IEMs mirror the effects of heterozygous variants

We next investigated the implicated genes' LoF by generating virtual IEMs for 24 genes that covered 60 gene–metabolite pairs via in silico knockout modeling (Methods and Extended Data Fig. 7). We compared the maximal secretion flux of the implicated metabolite into blood and/or urine between the wild-type WBM and the gene-knockout WBM. Initially, the direction of the observed gene–metabolite associations was correctly predicted by virtual IEMs with an accuracy of 73.3% in the male WBM and 76.7% in the female WBM, which is significantly better than chance (Fisher's exact test, $P$ value = $3.3 \times 10^{-3}$ (male), $P$ value = $1.5 \times 10^{-4}$ (female); Supplementary Table 11). After model curation informed by the observed gene–metabolite associations, which included the addition of metabolites (for example, 8-methoxykynurenate) and pathways as well as alteration of

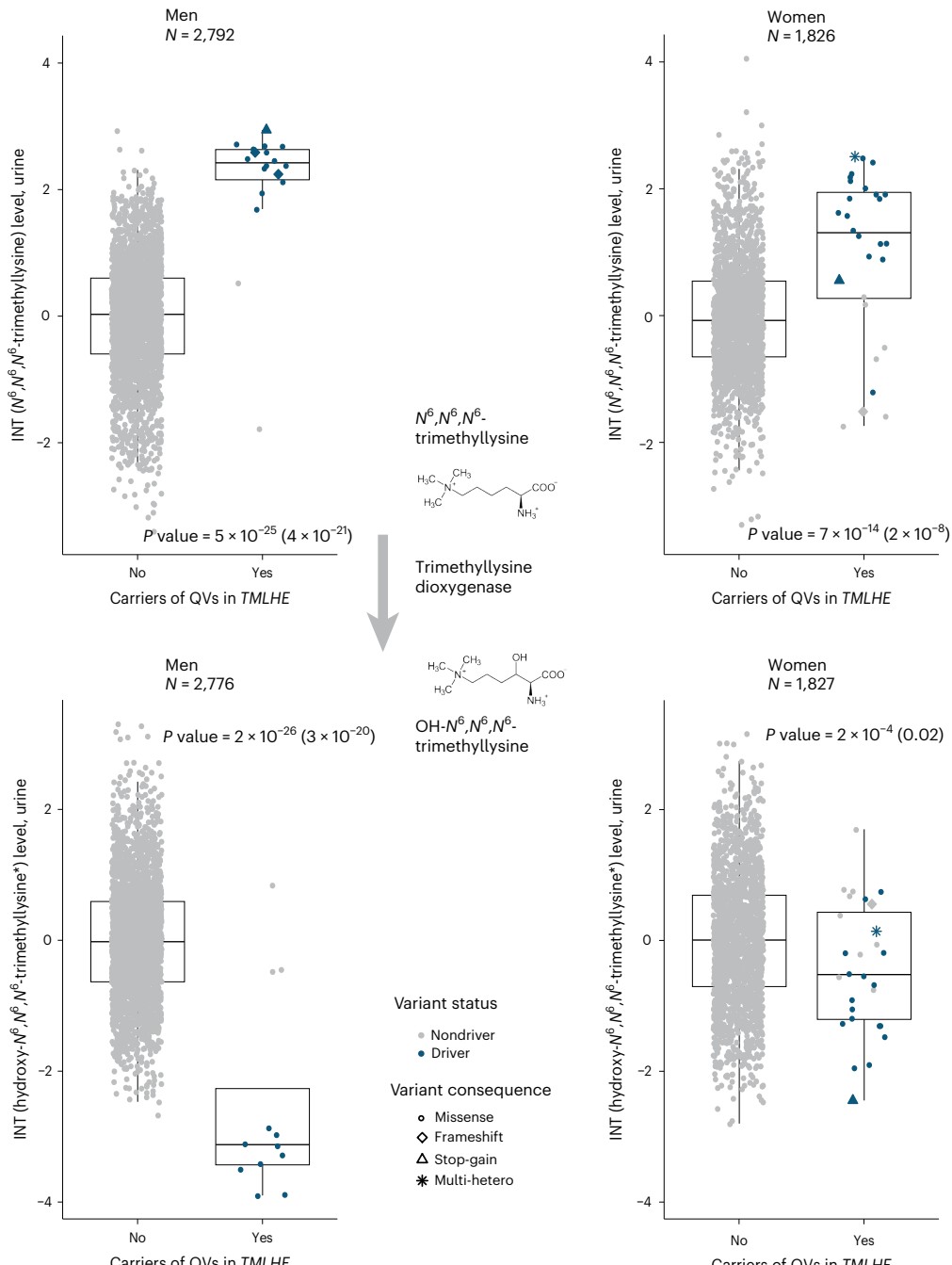

**Fig. 5 | Differences in urine metabolite levels between male and female carriers of QVs in X chromosomal *TMLHE* reflect a dose–response effect.** Top, plots represent covariate-adjusted urine levels of $N^6,N^6,N^6$-trimethyllysine after inverse normal transformation (INT; *y* axis) among male (left) and female (right) noncarriers and carriers of QVs in *TMLHE* based on the HI_mis mask (*x* axis). Symbol color and shape indicate a variant's driver status and consequence, respectively. The boxes range from the 25th percentile to the 75th percentile of metabolite levels, the median is indicated by a line, and whiskers end at the last observed value within 1.5 × (interquartile range) of the box. Among men hemizygous for a QV in *TMLHE*, the levels of the substrate $N^6,N^6,N^6$-trimethyllysine were markedly higher than in heterozygous women, reflecting more severe impairment of the encoded enzyme's function in hemizygous men. *P* values correspond to the sex-specific burden tests based on

driver variants, with *P* values based on all QVs in parentheses (Supplementary Table 10 and the Supplementary Methods). Metabolites' formulas were taken from https://commons.wikimedia.org/. Bottom, plots represent urine levels of covariate-adjusted hydroxy-$N^6,N^6,N^6$-trimethyllysine after inverse normal transformation (*y* axis) across male (left) and female (right) noncarriers and carriers of QVs in *TMLHE* based on the HI_mis mask (*x* axis). Because hydroxy-$N^6,N^6,N^6$-trimethyllysine is the product of trimethyllysine dioxygenase, the enzyme encoded by *TMLHE*, LoF QVs lead to decreased metabolite levels more strongly among men than among women. Effect sizes in men and women were significantly different (*P* value = $3 \times 10^{-4}$; Supplementary Methods). A schematic depiction of the well-studied reaction catalyzed by trimethyllysine dioxygenase[51] and of its substrate $N^6,N^6,N^6$-trimethyllysine and product hydroxy-$N^6,N^6,N^6$-trimethyllysine is included. Multi-hetero, multi-heterozygous.

constraints (for example, diet; details in the Supplementary Results and Supplementary Table 12), the number of modeled gene–metabolite associations increased to 67, and accuracy increased to 79.1%

(male, *P* value = $2.1 \times 10^{-5}$) and 83.58% (female, *P* value = $2.9 \times 10^{-7}$). These findings underline the predictive nature of the virtual IEMs for the aggregated effects of heterozygous damaging variants and

highlight opportunities to further improve WBMs by curation of the underlying knowledge base.

## Personalized WBMs capture observed metabolic changes

Virtual IEMs as described above only allow for qualitative prediction. To additionally study an equivalent to observed effect sizes, we introduced a second modeling strategy (Extended Data Fig. 7) as proof of principle, focusing on the gene *KYNU*. We successfully generated 569 microbiome-personalized[32] WBMs (Methods) and calculated the effect size of in silico *KYNU* knockout on metabolite excretion into urine against the natural variation induced by the personalized microbiomes (Supplementary Table 13). Eighteen of 257 metabolites had a modeling *P* value < 0.05/257, implicating them as potential biomarkers of the corresponding IEM kynureninase deficiency (Supplementary Table 14). The in silico effects of these 18 biomarkers, mostly belonging to tryptophan metabolism and the nicotinamide adenine dinucleotide (NAD)$^+$ de novo synthesis pathway, were significantly correlated with their observed counterparts (Supplementary Fig. 2a). Whereas two of the three metabolites with particularly large effects in both in silico modeling and the GCKD study, xanthurenate and 3-hydroxykynurenine, are known biomarkers of kynureninase deficiency[33], 8-methoxykynurenate was not. We therefore measured absolute levels of these metabolites in urine samples from a homozygous patient with kynureninase deficiency and her parents[34] (Supplementary Methods) and confirmed that, in addition to xanthurenate and 3-hydroxykynurenine, 8-methoxykynurenate also constituted a biomarker of this IEM (Fig. 6a and Extended Data Fig. 8), consistent with the association statistics from aggregate tests of heterozygous variants from the GCKD study. A similar observation was made with regard to the gene *PAH* (Fig. 6b, Supplementary Fig. 2b and Supplementary Results). Thus, in silico knockout modeling of two proof-of-principle examples faithfully captured metabolic changes observed for heterozygous variants detected in population studies and for the corresponding recessively inherited IEMs.

## Metabolites represent intermediate readouts of human traits

Allelic series describe a dose–response relationship, in which increasingly deleterious mutations in a gene result in increasingly larger effects on a trait or a disease. We hypothesized that genetic effects on metabolite levels should manifest as allelic series if the metabolite represents a molecular readout of an underlying (patho-)physiological process. As proof of principle, we investigated plasma sulfate because of solid evidence for causal gene–metabolite relationships: first, QVs in *SLC13A1* showed a significant aggregate effect on lower plasma sulfate levels (*P* value = 3 × 10⁻¹⁸, lowest possible *P* value = 2 × 10⁻²⁵). The observed association is well supported by experimental studies establishing that the encoded Na⁺–sulfate cotransporter NaS1 (SLC13A1) reabsorbs filtered sulfate at the apical membrane of kidney tubular epithelial cells[35]. Second, we had previously confirmed experimentally that plasma sulfate-associated QVs in *SLC26A1* are LoF alleles that lead to reduced sulfate transport[30], consistent with the aggregate effect of

driver variants in *SLC26A1* reaching a *P* value of 2 × 10⁻¹¹ for association with plasma sulfate (Extended Data Fig. 9). The encoded sulfate transporter SAT1 localizes to basolateral membranes of tubular epithelial cells and works in series with NaS1 to mediate transcellular sulfate reabsorption (Fig. 7a)[36,37].

Based on a growth retardation phenotype in *Slc13a1*-knockout mice[38] and an association between *SLC13A1* and lower sitting height in the UKB (*P* value = 3 × 10⁻⁸; Supplementary Tables 15 and 39), we investigated relations of six functional driver QVs in *SLC13A1* and *SLC26A1* with anthropometric measurements in the UKB (Methods). Supplementary Table 16 contains traits with which at least two QVs showed nominally significant associations (*P* value < 0.05). The genetic effect sizes on plasma sulfate levels in the GCKD study and both sitting and standing heights in the UKB were correlated (Pearson correlation coefficients of 0.57 and 0.70, respectively; Fig. 7b). These observations support a causal relationship between transcellular sulfate reabsorption and human height and designate plasma sulfate as an intermediate readout. Additionally, we observed significantly lower standing height among carriers of driver variants in *SLC13A1* and *SLC26A1* than among noncarriers in a subsample of the GCKD study (*N* = 3,239) with measured height. The aggregated effect size of driver variants in *SLC13A1* was −0.54 (corresponding to −5.17 cm when height was not inverse normal transformed, *P* value = 1.6 × 10⁻³; Supplementary Fig. 3a). For *SLC26A1*, we obtained even a stronger effect size of −0.73 (corresponding to −6.68 cm, *P* value = 1.7 × 10⁻⁶; Supplementary Fig. 3b).

The first patient homozygous for a LoF stop-gain mutation in *SLC13A1*, p.Arg12*, has just been described[39]. Aside from sitting height >2 s.d. below the normal range, the patient featured multiple skeletal abnormalities. Experimental transport studies[40] as well as the patient's fractional sulfate excretion of almost 100%[39] establish this variant as a complete LoF, resulting in renal sulfate wasting. In this study, we found that, compared with noncarriers of p.Arg12*, heterozygous carriers showed 0.95 s.d. lower plasma sulfate levels (GCKD, 22 carriers, *P* value = 9.9 × 10⁻¹⁰) and 0.08 s.d. lower sitting height (UKB, 2,480 carriers, *P* value = 2.2 × 10⁻⁷). Plasma sulfate measurements from heterozygous carriers therefore are indicative of more extreme phenotypic changes in homozygous carriers.

## Variants altering sulfate uptake and musculoskeletal traits

Rare LoF variants in *SLC13A1* and *SLC26A1* have been linked to individual musculoskeletal phenotypes through IEMs and GWASs[30,41–43]. We further investigated the association between the same six functional, sulfate-associated QVs in *SLC13A1* and *SLC26A1* and musculoskeletal disorders, fractures and injuries in the UKB, for which at least two carriers with and without disease were present (Methods). There were 116 nominally significant (*P* value < 0.05) associations with clinical traits and diseases, 113 of which were associated with increased odds of disease (Fig. 7c). For instance, the odds of various fractures ranged up to 30.7 (closed fracture of the neck, *P* value = 2.1 × 10⁻⁸, NaS1 p.Trp48*; Supplementary Table 17). While the increased odds

**Fig. 6 | Altered metabolite levels are a readout of impaired *KYNU* and *PAH* function: converging evidence from three approaches. a,b,** Three panels are shown for 8-methoxykynurenate levels associated with *KYNU* (**a**) and for phenylalanine levels associated with *PAH* (**b**) that visualize evidence from three complementary approaches. Left, covariate-adjusted inverse normal-transformed levels of the metabolite (*y* axis) among noncarriers (*N* = 4,589 and 4,562) and carriers (*N* = 25 and 151) of QVs in in the respective gene (*x* axis). Units correspond to standard deviations. The boxes range from the 25th percentile to the 75th percentile of metabolite levels, the median is indicated by a line, and whiskers end at the last observed value within 1.5 × (interquartile range) of the box. Middle, distribution of the ln-transformed secretion flux of the metabolite in mmol per day into urine (*y* axis) from minimum-norm quadratic programming (QP) simulations based on 569 and 567 microbiome-personalized WBMs without and with simulated knockout of *KYNU* and *PAH*, respectively (*x* axis).

**a**, Right, multiple-reaction monitoring (*m/z* 220.0 → 174.1) chromatograms of the diluted urine of a child with a homozygous loss of *KYNU* function[34] (patient), the heterozygous mother and the healthy father (maternal uniparental isodisomy). The signal at 12.5 min representing 8-methoxykynurenate is strongly enhanced in the patient sample. Chromatograms are normalized to urine creatinine concentrations; *y* axes are normalized to the intensity of the signal in the patient's chromatograms. **b**, Right, UV–visible chromatograms (570 nm) of amino acids (post-column derivatization with ninhydrin) in serum samples of a child with a homozygous loss of *PAH* function (c.1199+1G>C), the compound heterozygous father who additionally carries a mild mutation (c.1180G>C) and the heterozygous mother. The signal at 85.5 min represents phenylalanine (Phe). The signal at 106 min is the internal standard (IS). The reference range for phenylalanine concentrations in children is 38–137 μmol l⁻¹ and in adults is 26–91 μmol l⁻¹.

support a relationship between LoF variants in sulfate transporters and predisposition to several musculoskeletal disorders, the power to detect decreased odds was limited because of the rareness of the QVs and many of the disorders.

UKB participants who carried more than one copy of any of the six QVs were investigated more closely. The rare allele, resulting in the p.Arg272Cys substitution in NaS1, was observed in nine heterozygous carriers in the GCKD study and prioritized because of its location in a

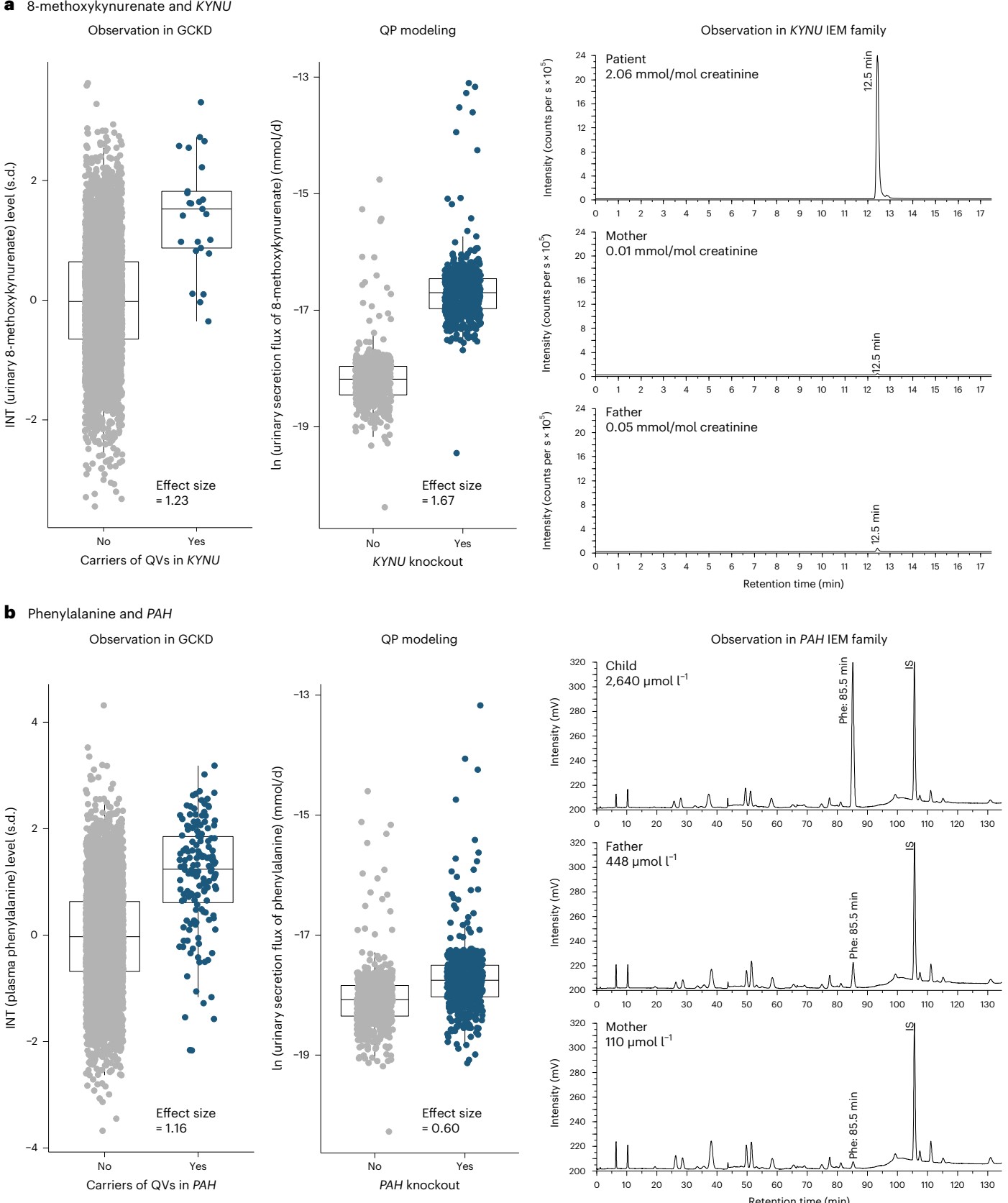

**a** 8-methoxykynurenate and *KYNU*

**b** Phenylalanine and *PAH*

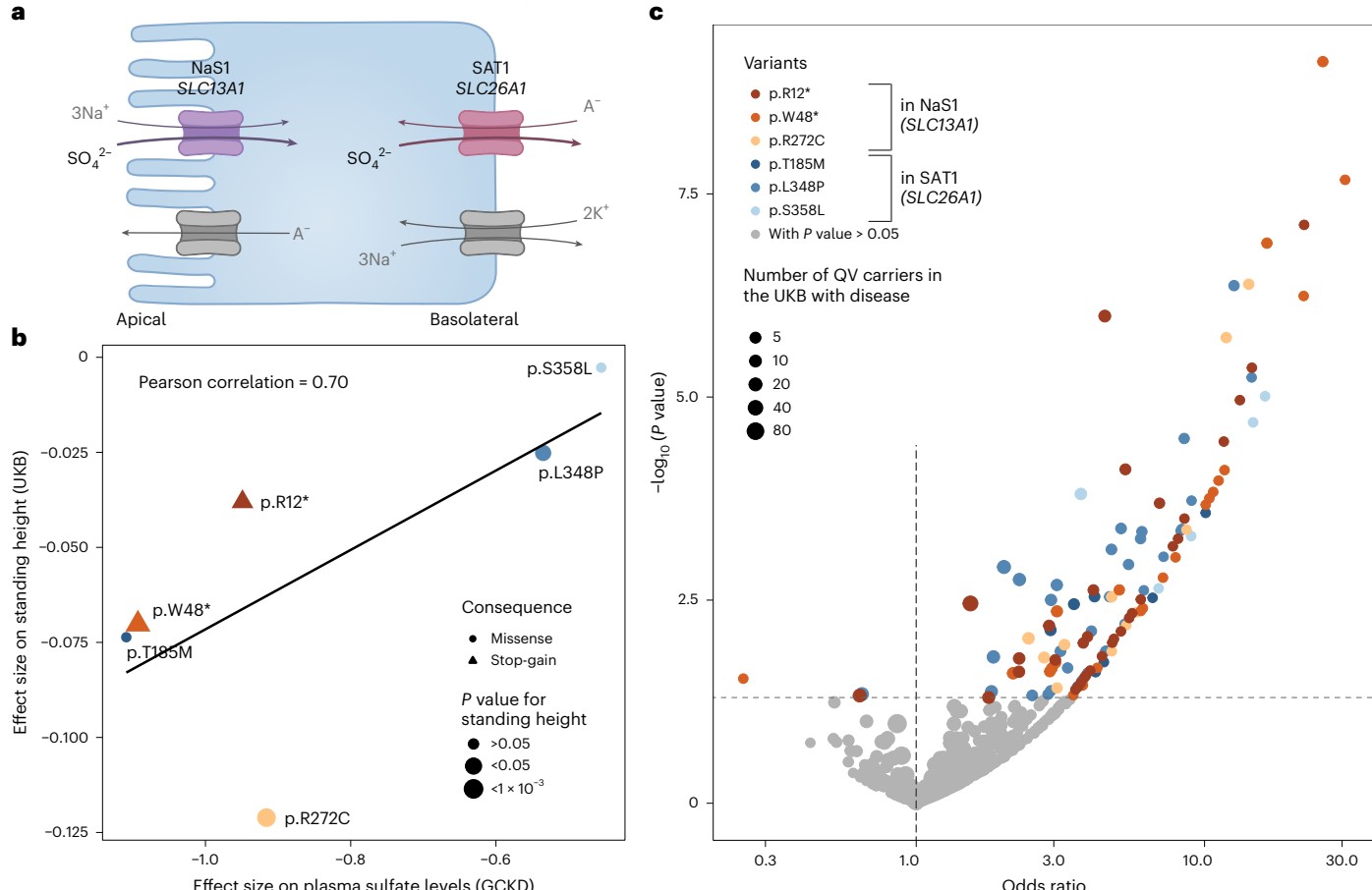

**Fig. 7 | Impact of functional QVs in *SLC13A1* and *SLC26A1* on height, musculoskeletal traits and fractures supports the role of plasma sulfate as an intermediate readout. a**, Schematic representation of the sulfate reabsorption mechanism involving NaS1 encoded by *SLC13A1* at the apical membrane and SAT1 encoded by *SLC26A1* at the basolateral membrane of epithelial cells. Figure created with https://www.biorender.com. **b**, Scatterplot shows the relation between the effect sizes of six QVs on plasma sulfate levels in the GCKD study (*x* axis) and on standing height in the UKB ($N \geq 466,907$) (*y* axis). Effect sizes correspond to single-variant association tests under additive modeling with inverse normal-transformed traits. Symbol color and shape indicate the gene (shades of red, *SLC13A1*; shades of blue, *SLC26A1*) and consequence of the QV. Symbol size represents the association *P* value with respect to height. The black line is the linear regression line through the data points. Variant effect sizes for

plasma sulfate levels are clearly correlated with the ones for standing height (Pearson correlation $r = 0.70$, allelic series). **c**, The volcano plot shows odds ratios (*x* axis) and $-\log_{10}$ (*P* values) (*y* axis) for associations of the six QVs with musculoskeletal diseases and fractures in the UKB ($N \geq 468,279$), based on a Firth regression (Methods and Supplementary Table 17). Only clinical traits for which at least two carriers were identified among both individuals with and without disease are included in the plot. Symbol color indicates the QV and whether the corresponding *P* value was nominally significant (*P* value < 0.05). Symbol size corresponds to the number of QV carriers with disease. While both increased and decreased odds of disease were observed when associations were not significant, increased odds for musculoskeletal diseases and fractures dominated for significant associations.

splice region, its high impact on plasma sulfate levels and its particularly large effect on human height (Fig. 7b). In the UKB, we found 294 heterozygous carriers of p.Arg272Cys, four persons who carried both p.Arg272Cys in NaS1 and p.Leu348Pro in SAT1 and a single person homozygous for p.Arg272Cys. Age- and sex-specific *z* scores for human height showed a clear dose–response effect (Fig. 8a and the Methods). The stronger effects among the four individuals heterozygous for LoF variants in each of the two transcellular sulfate reabsorption proteins as compared with heterozygous carriers of p.Arg272Cys only support additive effects across the pathway for human growth. Carrier status for NaS1 p.Arg272Cys was associated with increased odds of several musculoskeletal diseases such as back pain and intervertebral disk disorders as well as fractures (Fig. 8b). Homozygous persons were also identified for NaS1 p.Arg12* and SAT1 p.Leu348Pro, with similar findings (Extended Data Fig. 10). Together, these findings provide strong support that genetic variants that proxy lower transcellular sulfate reabsorption are associated with human height and several musculoskeletal traits and diseases. Prioritizing variants with strong

effects in allelic series for subsequent investigation in larger studies, even if the biomarker association rests on only a few heterozygous alleles, can therefore be an effective strategy to gain insights into the impact of rare damaging variants on human health.

**Relation of metabolite-associated genes to clinical traits**

A query of associations between the identified 2,077 QVs and 73 genes with thousands of quantitative and binary health outcomes using data from ~450,000 UKB participants (Supplementary Methods) revealed multiple biologically plausible significant and suggestive associations for genes (Supplementary Table 15) and QVs (Supplementary Table 18) but also less-studied relationships (Supplementary Results). The genes *SLC47A1*, *SLC6A19*, *SLC7A9* and *SLC22A7* were associated with one or more measures of kidney function and encode transport proteins highly expressed in the kidney[44–46]. Their localization at the apical[44–46] versus basolateral membrane of tubular epithelial kidney cells[47] corresponded to the matrix (urine versus plasma) in which they left corresponding metabolic fingerprints. This observation illustrates that rare genetic

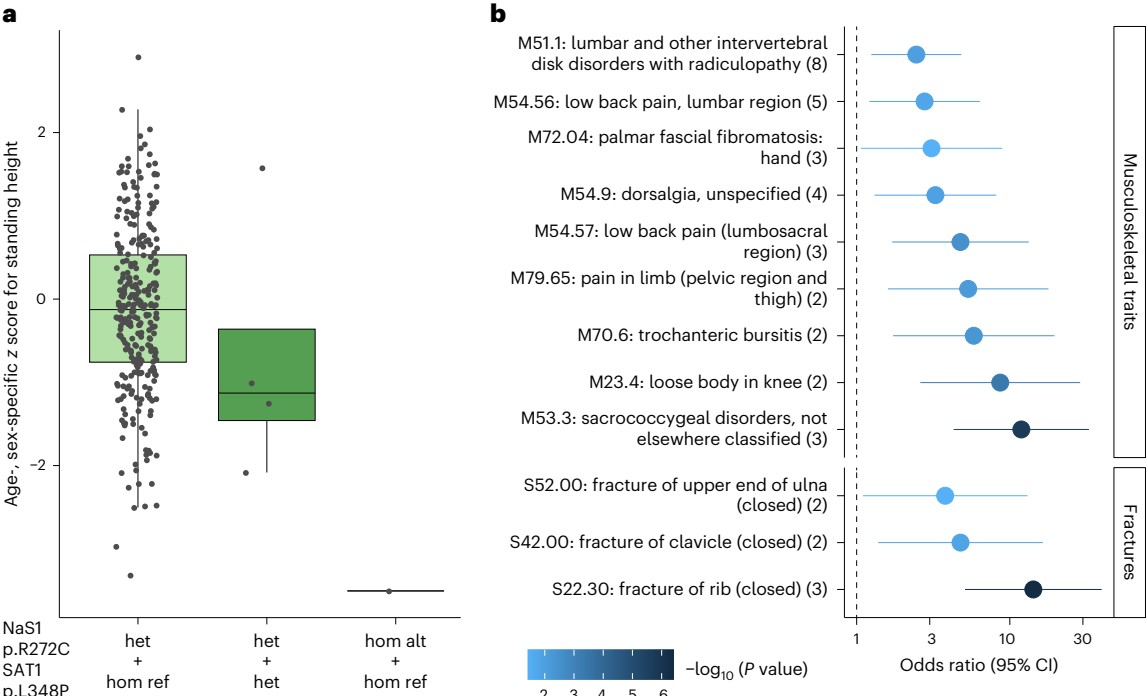

**Fig. 8 | Impact of different genotypes encoding the NaS1 p.Arg272Cys substitution on height and musculoskeletal traits and fractures. a**, The box plot shows differences in age- and sex-specific *z* scores for standing height (*y* axis; Methods) across UKB participants heterozygous and homozygous for the p.Arg272Cys-encoding allele (*x* axis). The boxes range from the 25th percentile to the 75th percentile of *z* scores, the median is indicated by a line, and whiskers end at the last observed value within 1.5 × (interquartile range) of the box. A dose–response effect is observable between heterozygous individuals (*N* = 289, median = −0.13, het + hom ref) and individuals carrying NaS1

p.Arg272Cys as well as SAT1 p.Leu348Pro (*N* = 4, median = −1.13, het + het) and one person homozygous for p.Arg272Cys (*z* score = −3.51, hom alt + hom ref). **b**, Association between the NaS1 p.Arg272Cys substitution with musculoskeletal diseases and fractures from the UKB (*N* ≥ 468,279), for which at least two carriers were identified among both individuals with and without disease (*y* axis). Numbers in parentheses indicate the number of p.Arg272Cys carriers with a respective disease. Odds ratios and their corresponding 95% confidence intervals (CI; *x* axis) are based on Firth regression (Methods). The symbol color reflects the −log$_{10}$ (*P* value). Only associations with *P* value < 0.05 are shown.

variants associated with clinical markers of organ function can leave specific signatures in organ-adjacent biofluids that reflect their roles in cellular exchange processes.

## Discussion

We performed a comprehensive screen of the aggregate effect of rare, putatively damaging variants on the levels of 1,294 plasma and 1,396 urine metabolites from paired specimens of 4,737 persons. The majority of the 192 identified gene–metabolite relationships have not been reported yet[14–20,24] and include plasma- and urine-exclusive associations that reflect organ function. The findings were validated through primary data analysis for metabolites available in the UKB, investigation of previously published summary statistics from sequencing-based genetic studies of the plasma metabolome, integration of orthogonal plasma proteomics data and proof-of-concept experimental studies that confirmed a new metabolite association with the transport protein encoded by *SLC6A19*.

We show, via several genetic, computational and experimental approaches that the rare, almost exclusively heterozygous metabolite-associated variants in our study capture similar information about a gene's function as can be obtained from the study of rare IEMs but are observed much more frequently and permit insights into graded effects of impaired gene function. First, 38% of identified genes in our study are known to harbor causative mutations for autosomal recessively inherited IEMs that often exhibit concordant but more extreme changes in the implicated metabolite, as exemplified by elevated urine levels of cystine in cystinuria (MIM 220100, *SLC7A9*) or tryptophan in Hartnup disease (MIM 234500, *SLC6A19*).

Second, men exhibited significantly larger effects of rare QVs in non-pseudo-autosomal X chromosomal genes on metabolite levels than women. This observation is consistent with male hemizygosity as an approximation of female homozygosity for a given variant and with the known greater penetrance and severity of X-linked disorders in men than in women[48].

Third, in silico knockout in a virtual metabolic human, that is, full loss of gene function, was predictive for observed metabolic changes associated with variant heterozygosity. Predicted changes on metabolite levels upon in silico gene knockout were also reflected in absolute metabolite quantification of patients with IEM homozygous for a LoF mutation in the respective genes, *KYNU*[34] and *PAH*. Thus, deterministic, knowledge-based in silico modeling generated context for better biological interpretation also of heterozygous variants, while genetic screens of metabolite levels in population studies permit the identification of knowledge gaps and errors in WBMs. Our modeling pipeline for generating virtual IEMs, which we make publicly available to substantiate evidence from rare variant aggregation tests, will constitute a valuable resource in particular to scrutinize genes for which an IEM has yet to be observed.

Fourth, the presence of different causal QVs affecting a given metabolic reaction or pathway enabled the investigation of allelic series. The resulting dose–response relationships proxy a range of target inhibition, which represents desirable information for drug development and is relevant because enzymes and transporters are attractive drug targets. Plasma sulfate-associated functional QVs in *SLC13A1* and *SLC26A1* showed a clear dose–response effect between the degree of genetically inferred impaired transcellular sulfate reabsorption

and lower human height. This observation is biologically plausible, because defects in genes linked to sulfate biology often result in perturbed skeletal growth and development[49]. In particular, constitutive knockouts of *Slc13a1* and *Slc26a1* in mice do not only cause hyposulfatemia and renal sulfate wasting[38,50] but also general growth retardation in *Slc13a1*-knockout mice[38]. Interestingly, the missense variant p.Thr185Met in SAT1 exhibited the largest effect on sulfate. We have previously shown experimentally a dominant negative mechanism of this variant[30], providing another mechanism of how heterozygous variants may promote insights into an effectively full loss of gene function. Moreover, our findings for the p.Arg272Cys variant in NaS1 show that even very few, heterozygous copies of a metabolite-prioritized QV can give rise to the detection of homozygous individuals and hitherto unreported disease associations in subsequent larger studies. These observations suggest that the importance of impaired transcellular epithelial sulfate transport for musculoskeletal diseases, fractures and injuries deserves additional study and should be further substantiated through conditional or mediation analyses if plasma sulfate levels become available in the UKB.

Potential limitations of our study include a focus on participants of European ancestry with moderately reduced kidney function, potential violations of assumptions underlying burden tests, in silico prediction of QV pathogenicity and of whole-body modeling and the use of semi-quantitative rather than absolute metabolite levels. Arguments mitigating each of these concerns are detailed in the Supplementary Discussion.

In conclusion, exome-wide population studies of rare, putative LoF variants can reveal potentially causal relationships with metabolites and highlight metabolic biomarkers informative of the degree of impaired gene function that can translate into graded associations with human traits.

## Online content

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

[1]Institute of Genetic Epidemiology, Faculty of Medicine and Medical Center, University of Freiburg, Freiburg, Germany. [2]Spemann Graduate School of Biology and Medicine, University of Freiburg, Freiburg, Germany. [3]Department of Psychiatry and Psychotherapy, University Medicine Greifswald, Greifswald, Germany. [4]Department of Epidemiology, Johns Hopkins Bloomberg School of Public Health, Baltimore, MD, USA. [5]Centre for Integrative Biological Signalling Studies, Albert-Ludwigs-Universität Freiburg, Freiburg, Germany. [6]Department of Medicine IV, Nephrology and Primary Care, Faculty of Medicine and Medical Center, University of Freiburg, Freiburg, Germany. [7]Laboratory of Clinical Biochemistry and Metabolism, Department of General Pediatrics, Adolescent Medicine and Neonatology, Medical Center, Faculty of Medicine, University of Freiburg, Freiburg, Germany. [8]Faculty of Biology, University of Freiburg, Freiburg, Germany. [9]Department of Nephrology and Hypertension, University Hospital Erlangen, Friedrich-Alexander-Universität Erlangen–Nürnberg, Erlangen, Germany. [10]Research, Maze Therapeutics, South San Francisco, CA, USA. [11]SYNLAB MVZ Humangenetik Freiburg, Freiburg, Germany. [12]Department of General Pediatrics, Adolescent Medicine and Neonatology, Medical Center, Faculty of Medicine, University of Freiburg, Freiburg, Germany. [13]Institute of Functional Genomics, University of Regensburg, Regensburg, Germany. [14]Department of Nephrology and Medical Intensive Care, Charité–Universitätsmedizin Berlin, Berlin, Germany. [15]School of Medicine, University of Galway, Galway, Ireland. [16]Ryan Institute, University of Galway, Galway, Ireland. [17]Division of Microbiology, University of Galway, Galway, Ireland. [18]APC Microbiome Ireland, Cork, Ireland. [19]German Centre for Cardiovascular Research (DZHK), partner site Greifswald, Greifswald, Germany. ✉e-mail: johannes.hertel@med.uni-greifswald.de; anna.koettgen@uniklinik-freiburg.de

## Methods

### Study design and participants

The GCKD study is an ongoing prospective cohort study of 5,217 participants with moderate chronic kidney disease who were enrolled from 2010 to 2012 and are under regular nephrologist care. Inclusion criteria were an age between 18 and 74 years and an eGFR between 30 and 60 ml min$^{-1}$ per 1.73 m$^2$ or an eGFR >60 ml min$^{-1}$ per 1.73 m$^2$ with a UACR >300 mg per g or with a urinary protein-to-creatinine ratio >500 mg per g[52]. Biomaterials, including blood and urine, were collected at the baseline visit, processed and shipped frozen to a central biobank for storage at −80 °C[53]. Details on the study design and participant characteristics have been published[52,54]. The GCKD study was registered in the national registry for clinical studies (DRKS 00003971) and approved by local ethics committees of the participating institutions[52]. All participants provided written informed consent.

### Whole-exome sequencing and quality control

Genomic DNA was extracted from whole blood and underwent paired-end 100-bp WES at Human Longevity, using the IDT xGen version 1 capture kit on the Illumina NovaSeq 6000 platform. More than 97% of consensus coding sequence (CCDS) release 22 (ref. 55) had at least 10-fold coverage. The average coverage of the CCDS was 141-fold read depth. Exomes were processed from their unaligned FASTQ state in a custom-built cloud compute platform using the Illumina DRAGEN Bio-IT Platform Germline Pipeline version 3.0.7 at AstraZeneca's Centre for Genomics Research, including alignment of reads to the GRCh38 reference genome (https://ftp.ncbi.nlm.nih.gov/genomes/all/GCA/000/001/405/GCA_000001405.15_GRCh38/) and variant calling[56].

Sample-level quality control included removal of samples from participants who withdrew consent, duplicated samples, those with an estimated VerifyBamID contamination level >4%[57], samples with inconsistency between reported and genetically predicted sex, samples not having chromosomes XX or XY, samples having <94.5% of CCDS release 22 bases covered with ≥10-fold coverage[55], related samples with kinship >0.884 (KING, kinship version 2.2.3)[58] and samples with a missing call rate >0.03. Furthermore, only samples with available high-quality DNA microarray genotype data and without outlying values (>8 s.d.) along any of the first ten genetic principle components from a principal component analysis[59] were kept, for a final sample size of 4,779 samples.

Variant-level quality control was performed similar to that in ref. 56, excluding variants with coverage <10, heterozygous variants with a one-sided binomial exact test $P$ value $<1 \times 10^{-6}$ for Hardy–Weinberg equilibrium, variants with a genotype quality score <30, single-nucleotide variants with a Fisher's strand bias score (FS) >60 and insertions and deletions with an FS >200, variants with a mapping quality score <40, those with a quality score <30, variants with a read position rank-sum score <−2, those with a mapping quality rank-sum score <−8, variants that did not pass the DRAGEN calling algorithm filters, heterozygous genotype called variants based on an alternative allele read ratio <0.2 or >0.8 and variants with a missing call rate >10% among all remaining samples. This resulted in 1,038,062 variants across the autosomes and the X chromosome.

### Variant and gene annotation

Variants from WES were annotated using the Variant Effect Predictor (VEP) version 101 (ref. 60) with standard settings, including the canonical transcript, gene symbol and variant frequencies from gnomAD version 2.1 (https://gnomad.broadinstitute.org/). VEP plugins were used to add the REVEL (version 2020-5)[61] and CADD (version 3.0)[62] scores and to downgrade LoF variants using LOFTEE (version 2020-8)[63]. Furthermore, we added multiple in silico prediction scores using dbNSFP version 4.1a[64].

For interpretation, genes were annotated for their potential function as enzymes using UniProt (https://www.uniprot.org/)[65] and as transporters using data from Gyimesi and Hediger[66].

### Metabolite identification and quantification

Metabolite levels were quantified from stored plasma and spot urine as published by Schlosser et al.[2]. In brief, nontargeted mass spectrometry analysis was conducted at Metabolon. Metabolites were identified by automated comparison of the ion features in the experimental sample to a reference library of chemical standards. Known metabolites reported in this study were identified with the highest confidence level of identification of the Metabolomics Standards Initiative[67,68], unless marked with an asterisk. Unnamed biochemicals of unknown structural identity were identified by virtue of their recurrent nature. For peak quantification, the area under the curve was used, followed by normalization to account for interday instrument variation.

### Data cleaning of quantified metabolites

Data cleaning, quality control, filtering and normalization of quantified metabolites in plasma and urine in the GCKD study were performed using an in-house pipeline[2]. Samples and metabolites were evaluated for duplicates; missing and outlying values and metabolites with low variance were excluded. Levels of urine metabolites were normalized using the probabilistic quotient[69] derived from 309 endogenous metabolites with <1% missing values to account for differences in urine dilution. After removing metabolites with <300 individuals with WES data, the remaining 1,294 plasma and 1,396 urine metabolites (Supplementary Table 2) were inverse normal transformed before gene-based aggregation testing. Therefore, effect sizes based on effects of aggregated rare variants on the semi-quantitative metabolite measurements have 1 s.d. as a unit.

### Additional variables

Serum and urine creatinine were measured using an IDMS-traceable enzymatic assay (Creatinine Plus, Roche). Serum and urine albumin levels were measured using the Tina-quant assay (Roche–Hitachi Diagnostics). GFR was estimated with the CKD-EPI formula[70] from serum creatinine. UACR was calculated using urinary albumin and creatinine measurements. Full information on WES data, covariates and metabolites was available for 4,713 persons regarding plasma metabolites and for 4,619 persons regarding urine metabolites. Genetic principal components were derived based on principal component analysis on the basis of genotype data using flashpca[71].

### Rare variant aggregation testing on metabolite levels

We performed burden tests to combine the effects of rare, putatively damaging variants within a gene on metabolite levels assuming a LoF mechanism that results in concordant effect directions on metabolite levels[72]. The selection of high-quality QVs into masks based on their frequency and annotated properties is a state-of-the-art approach in variant aggregation studies[73]. Annotations from VEP version 101 (ref. 60) were used to select QVs within each gene for aggregation in burden tests. Because genetic architectures of damaging variants vary across genes, two complementary masks for the selection of QVs were defined. Both masks were restricted to contain only rare variants in canonical transcripts with MAF <1%. All variants that were predicted to be either high-confidence LoF variants or missense variants with a MetaSVM score >0 or in-frame nonsynonymous variants with a fathmm-XF-coding score >0.5 were aggregated into the first mask, termed LoF_mis. The second mask, termed HI_mis, contained all variants that were predicted either to have a high-impact consequence defined by VEP (transcript ablation, splice acceptor variant, splice donor variant, stop-gain variant, frameshift variant, start/stop lost variant, and transcript amplification) or to be missense variants with a REVEL score >0.5, a CADD PHRED score >20 or an

M-CAP score >0.025. Only genes with an HGNC symbol that were not read-throughs and that contained more than three QVs in at least one of the masks were kept for testing, resulting in 16,525 analyzed genes. Burden tests were carried out as implemented in the seqMeta R package version 1.6.7 (ref. 74), adjusting for age, sex, ln(eGFR) and the first three genetic principal components as well as serum albumin for plasma metabolites and ln(UACR) for urinary metabolites, respectively[2]. Genotypes were coded as the number of copies of the rare allele (0, 1, 2) on the autosomes and also on the X chromosome for women. For men, genotypes in the non-pseudo-autosomal region of the X chromosome were coded as (0, 2). Statistical significance was defined as nominal significance corrected for the number of tested genes and principal components that explained more than 95% of the metabolites' variance ($0.05/16{,}525/600 = 5.04 \times 10^{-9}$ in plasma, $0.05/16{,}525/679 = 4.46 \times 10^{-9}$ in urine). For significant gene–metabolite associations, single-variant association tests between each QV in the respective mask and the corresponding metabolite levels were performed under additive modeling, adjusting for the same covariates using the seqMeta R package version 1.6.7 (ref. 74). Sensitivity analyses that evaluated all significant gene–metabolite pairs with regard to additional gene-based tests as well as across strata of sex and kidney function are summarized in the Supplementary Methods and Supplementary Tables 4 and 5.

## Assessment of QV contributions and driver variants

The investigation of the genetic architecture underlying gene–metabolite associations and the prioritization of QVs according to their contribution to the gene-based association signal were performed using the forward selection procedure from Bomba et al.[15]. First, for each QV $v$, the $P$ value $P_v$ is calculated by performing the burden test aggregating all QVs other than $v$. Second, for each QV $v$, the difference $\Delta_v$ between $P_v$ and the total $P$ value of the burden test including all QVs is calculated. Subsequently, QVs are ranked by the magnitude of $\Delta_v$. QVs not contributing to the gene signal or even having an opposite effect can provide a negative $\Delta_v$. Finally, burden tests are performed by adding the ranked QVs one after the other until the lowest $P$ value is reached, starting with the greatest $\Delta_v$. This identified a set of QVs that contained only variants that contributed most to the gene–metabolite association signal (that is, led to a stronger association signal) and did not contain variants that introduced noise (that is, neutral variants or those with a small or even opposite effect on metabolite levels). The resulting set of selected variants that led to the lowest possible association $P$ value was designated 'driver variants' for the respective gene–metabolite association. Driver variants within a gene might differ for different associated metabolites.

## Relation of QVs in *SLC13A1* and *SLC26A1* to musculoskeletal traits

WES and biomedical data of the UKB were used to investigate allelic series of functional QVs in *SLC13A1* and *SLC26A1* with hypothesized related clinical traits and diseases. We focused on *SLC13A1* driver variants with experimental validation or that likely result in a severe consequence (stop-gain, splicing) to select truly functional QVs. Among these, the stop-gain variant encoding p.Arg12*, for which a complete LoF has experimentally been validated[40], the stop-gain substitution p.Trp48*, for which associations with decreased serum sulfate levels[42] and skeletal phenotypes[41] were reported, and the missense variant encoding p.Arg272Cys, located in a splice region, were available in the UKB. For *SLC26A1*, we selected driver QVs for which reduced sulfate transport activity had previously been shown[30], of which p.Leu384Pro, p.Ser358Leu and p.Thr185Met were available in the UKB. All 6 QVs passed the '90pct10dp' QC filter, defined as at least 90% of all genotypes for a given variant, independent of variant allele zygosity, had a read depth of at least 10 (https://biobank.ndph.ox.ac.uk/ukb/ukb/docs/UKB_WES_AnalysisBestPractices.pdf).

Analyses were performed on the UKB Research Analysis Platform. Participants with all ancestries were included into the analysis but excluding strongly related individuals, defined as those that were excluded from the kinship inference process and those with ten or more third-degree relatives. After individual-level filtering, 468,292 individuals remained for analyses. Of these, ten participants were homozygous for one of the six QVs and 7,280 persons were heterozygous for at least one of the QVs. For these homozygous or heterozygous persons, we determined age- and sex-specific $z$ scores of their quantitative anthropometric measurements, enabling interpretation of their measurements compared with noncarriers of the same age and sex. Age- and sex-specific distributions were inverse normal transformed before calculating $z$ scores.

The association between each of the six functional QVs with medical diagnoses defined by International Classification of Diseases version 10 (ICD-10) codes based on UKB field 41202 (primary or main diagnosis codes across hospital inpatient records) was investigated. We selected musculoskeletal diseases (ICD-10 codes starting with 'M') and fractures and injuries (ICD-10 codes starting with 'S' and containing 'fracture', 'dislocation' or 'sprain' terms). To avoid unreliable estimates, traits were restricted to those with at least two rare variant carriers among both individuals with and without disease. The association was examined using Fisher's exact test under dominant modeling and Firth regression under additive modeling ('brglm2' R package[75]). We included sex, age at recruitment, sex × age and the first 20 genetic principal components (UKB field 22009) as covariates in the regression model. The association with quantitative anthropometric traits was assessed after inverse normal transformation via linear regression, additive genotype modeling and adjusting for the same covariates.

## Gene-based tests for metabolite associations in the UK Biobank

We performed gene-based tests for significantly associated metabolites available in the UKB to validate our findings using the same settings for analysis as those in our study. Because metabolite levels in the UKB were quantified by Nightingale Health's metabolic biomarker platform focusing on lipids, only two (histidine and phenylalanine) of the 122 significantly associated plasma metabolites were available.

Histidine and phenylalanine values (UKB data fields 23463 and 23468) were inverse normal transformed. Sample and variant QC was performed, and covariates were included as described in the previous paragraph. A total of 260,000 individuals were available for analysis. Association analysis for the two identified gene–metabolite pairs, histidine and *HAL* as well as phenylalanine and *PAH*, was performed based on burden tests as implemented in REGENIE version 3.3 in two steps using the HI_mis mask, selecting only QVs that were present in the GCKD study to ensure reproducibility of rare variant effects between the studies.

## Setup of the whole-body model and mapping

The sex-specific and organ-resolved WBM covers 13,543 unique metabolic reactions and 4,140 unique metabolites based on the generic genome-scale reconstruction of human metabolism, Recon3D[23], and adequate physiological and coupling constraints[22,24].

Of all observed significant gene–metabolite pairs from the GCKD study, 51 genes and 69 metabolites could be mapped onto Recon3D. For 36 of 51 genes, their associated metabolites could be mapped, resulting in 69 unique gene–metabolite pairs. To investigate perturbations in gene $G$, we first identified all reactions $R_G = \{r_{G_1}, \ldots, r_{G_n}\}$ of the corresponding encoded enzymes or transporters in the WBM[76]. We included those genes (27 of 36) in the generation of virtual IEMs that were exclusively causal for a non-empty set of reactions (that is, for a gene $G$, associated with reactions $R_G = \{r_{G_1}, \ldots, r_{G_n}\}$, there did not exist a gene $H$ that was associated with any reaction of $R_G$) and metabolites with urinary excretion reactions, leading to the exclusion of *SLC22A7* and *SULT2A1*.

## In silico knockout modeling via linear programming

Knockout simulations were based on maximizing the flux of the excretion or demand reaction of the metabolite of interest $M$ under different conditions in a steady state setting ($Sv = 0$), where $S$ is the stoichiometric matrix (rows, metabolites; columns, reactions), and $v$ is the flux vector through each reaction, adhering to specific constraints ($v_l \leq v \leq v_u$)[22,77]:

$$\max_{v} c^T v,$$
$$\text{subject to } Sv = 0, \tag{1}$$
$$v_l \leq v \leq v_u.$$

For simulating a wild-type model for gene $G$, we solved the linear programming (LP) problem stated in equation (1), choosing the linear objective as the sum of all corresponding fluxes of reactions in $R_G$:

$$S_G := \max \sum_{k=1}^{n} v_{G_k},$$
$$\text{subject to } Sv = 0, \tag{2}$$
$$v_l \leq v \leq v_u.$$

First, we checked whether $S_G > 10^{-6}$, a criterion implemented in the function checkIEM_WBM of the PSCM toolbox for deciding whether the corresponding reactions could carry any flux[22,78]. All reactions except the *TMLHE*-associated reactions passed this criterion.

Next, we maximized the flux of two key reactions: the urine excretion reaction (for example, $EX_M[u]$) and the created unbounded demand reaction (for example, $DM_M[bc]$), designed to reflect accumulation in the blood compartment. First, we unbounded the upper bound of the urine excretion reaction. Next, we maximized the corresponding fluxes of metabolite $M$ as the LP problem stated in equation (1) under the additional constraint that $\sum_{k=1}^{n} v_{G_k} = S_G$, providing the maximal urine excretion and the maximal flux into blood given the constraint setting. Finally, to simulate the complete LoF, we blocked all reactions in all organs catalyzed by gene $G$ by setting $v_{G_1} = ... = v_{G_n} = 0$. We derived maximum fluxes as in the wild-type model. Subsequently, we tested whether the knockout resulted in an increase, a decrease or no change in $EX_M[u]$ and $DM_M[bc]$ for each mapped gene–metabolite pair that was significant in the GCKD cohort.

From the initial 36 genes mapped onto Recon3D, 24 genes and their mapped metabolites fulfilled all criteria (exclusively causal, reactions of the genes carry flux, urinary excretion reaction present), leading to 60 modeled gene–metabolite pairs. After curation of the male and female models, 26 genes (*TMLHE* and *KYAT1* added) and 67 gene–metabolite pairs could be computed (Supplementary Methods).

LP simulations were carried out in Windows 10 using MATLAB 2021a (MathWorks) as the simulation environment, ILOG CPLEX version 12.9 (IBM) as the LP solver, the COBRA Toolbox version 3.4 (ref. 78) and the PSCM toolbox[22].

## Microbiome personalization of whole-body models

Microbiome-personalized WBMs were generated by creating community models based on the genome-scale reconstructions of microbes in the AGORA1 resource[79,80]. Models have been shown to accurately reflect aspects of the fecal host metabolome[80,81]. Briefly, from microbe identification and relative abundance data of a metagenomic sample, genome-scale reconstructions of the identified microbes are joined together and connected via a lumen compartment, where they can exchange metabolites to form a microbial community[82,83]. Each microbial community model is then integrated in the WBM by connecting the microbiota lumen compartment to the large intestinal lumen of the WBM. Microbial community models ($n = 616$) were based on publicly available metagenomics data from Yachida et al.[32] and then embedded into the male WBM to form 616 personalized WBMs.

## In silico knockout modeling using quadratic programming

While maintaining the same conditions as outlined in equation (1), rather than maximizing a linear objective, we minimized a quadratic objective for each personalized WBM:

$$\min_{v} \frac{1}{2} v^T Q v,$$
$$\text{subject to } Sv = 0, \tag{3}$$
$$v_l \leq v \leq v_u.$$

Here, $Q$ is a diagonal matrix, with $10^{-6}$ on its diagonal, a value recommended in the COBRA Toolbox[78]. Because of convexity attributes, equation (3) allows for calculation of a unique flux distribution. For each solution $v^*$, we obtained the corresponding urine excretion reactions of the measured and mapped metabolites. For knockout simulations, the associated reactions of gene $G$ were set to zero ($v_{G_1} = ... = v_{G_n} = 0$). Then, equation (3) was solved if possible. An optimal quadratic programming (QP) solution could be computed for 582 wild-type models, 590 *KYNU*-knockout WBMs and 588 *PAH*-knockout WBMs, which led to 569 paired QP–*KYNU* solutions and 567 paired QP–*PAH* solutions. We analyzed urine secretion fluxes for 257 metabolites covered in the GCKD urine metabolome data and 272 metabolites covered in the GCKD plasma metabolome data that had non-zero flux values. For *KYNU*, the urine compartment was analyzed, as biomarker quantification for the corresponding IEM is done in urine. Analogously for *PAH*, the blood metabolome data were analyzed as the clinically relevant compartment. The QP simulations were carried out using the high-performance computing facility, called the Brain-Cluster, of the University of Greifs-wald, employing MATLAB 2019b (MathWorks), ILOG CPLEX version 12.10 (IBM) as the quadratic programming solver and the COBRA Toolbox version 3.4 (ref. 78).

## Statistical analysis of the in silico simulation results

The Fisher–Freeman–Halton test was used to determine significance when comparing the in vivo and in silico signs from LP modeling. Statistical analysis of the QP solutions was conducted based on the paired wild-type and knockout fluxes via fixed-effect linear regression for panel data[84]. We used ln(urine secretion flux) as the response variable, the knockout status as the sole predictor (wild type versus knockout) and the personalized microbiome as a fixed effect. Significance thresholds were set to 0.05/257 (*KYNU*) and 0.05/272 (*PAH*). Importantly, the entire variance in the regression models had two sources: (1) the knockout and (2) the microbiome personalization. Significance testing of the in silico regression coefficient of the knockout variable therefore delivers a test of whether the knockout explains substantial amounts of variance in comparison to the variance induced by randomly sampled microbiome communities. The in silico regression coefficients were then correlated with the burden-derived observed regression coefficients of gene–metabolite associations from the GCKD study, and significance was determined using the standard test for Pearson correlations.

## Experiments on transport activity of SLC6A19

**Generation of cells.** Human *SLC6A19* (NM_001003841.3 → NP_0010 03841.1) and human *CLTRN* (*TMEM27*) (NM_020665.6 → NP_065716.1) cDNA was synthesized at Life Technologies Gene Art and cloned into a T-REx inducible expression vector. Both vectors were transfected into CHO T-REx cells and selected with neomycin and hygromycin. Mock cells were made by transfecting with only the TMEM27 vector and selection using hygromycin. Stable pools were then selected by measuring doxycycline-inducible uptake of neutral amino acids (for example, isoleucine) by measuring changes in membrane potential using the FLIPR Tetra system. The selected stable cell pools were then serially diluted to generate single-cell clones, which were subsequently selected based on function using the FLIPR assay and hSLC6A19 and hTMEM27 expression using qPCR.

**FLIPR membrane potential assay.** CHO T-REx cells stably expressing doxycycline-inducible hSLC6A19 and hTMEM27 were seeded in a 384-well plate and incubated overnight with 1 µg ml⁻¹ doxycycline. The next day, cells were washed and then incubated with Tyrode's buffer (sodium free) with FMP-Blue-Dye, which is a membrane potential dye, for 60 min. The cells were then incubated with standard Tyrode's buffer (130 mM NaCl) with and without cinromide for 10 min before incubation with standard Tyrode's buffer alone or with eight increasing concentrations of methionine sulfone and isoleucine, both with maximum concentrations of 30 mM. The FLIPR Tetra system was used to read FMP-Blue-Dye fluorescence as a measurement of membrane depolarization as a result of substrate-driven electrogenic net influx of $Na^+$. Data were analyzed and represented in two ways: (1) for data comparison with the mock cell line, transport activity was presented as fold over non-substrate-driven signal with the formula (fluorescence signal − median of fluorescence signal with no substrate)/(median of fluorescence signal with no substrate); and (2) for data comparison with cinromide, transport activity was presented as a percent of maximum substrate-driven fluorescence signal with the formula 100 × (fluorescence signal − median of fluorescence signal with no substrate)/(median of fluorescence signal with substrate).

**Reporting summary**

Further information on research design is available in the Nature Portfolio Reporting Summary linked to this article.

## Data availability

The summary statistics of all significant gene–metabolite associations based on burden tests using two masks as well as all involved QVs with annotations are available in Supplementary Tables 3 and 7, respectively. Genotype, metabolite, protein and phenotype data were obtained from the UKB (https://www.ukbiobank.ac.uk/) and the GCKD study (https://www.gckd.org/). This research has been conducted using the UKB resource under application number 64806. The following external data resources were used: the GRCh38 reference genome (https://ftp.ncbi.nlm.nih.gov/genomes/all/GCA/000/001/405/GCA_000001405.15_GRCh38/), alignment of reads; the GTEx Project (https://gtexportal.org/home/), investigation of gene expression and QTLs across tissues; the AstraZeneca PheWAS Portal (https://azphewas.com/), search for gene- and variant-level associations of detected genes and QVs; the OMIM catalog (https://www.omim.org/), query for monogenic disorders and traits related to identified genes; the Genomics England PanelApp (https://panelapp.genomicsengland.co.uk/panels/467/ version 4.0), search for known IEMs related to the detected genes; the Open Targets Platform (https://platform.opentargets.org/), search for drug target status and the corresponding indication for identified genes; the ClinVar archive (https://www.ncbi.nlm.nih.gov/clinvar/), query for clinical significance and the corresponding trait or disease of detected QVs; microbiome abundance data (https://static-content.springer.com/esm/art%3A10.1038%2Fs41591-019-0458-7/MediaObjects/41591_2019_458_MOESM3_ESM.xlsx) and the AGORA resource of genome-scale microbial reconstructions (https://github.com/VirtualMetabolicHuman/AGORA/), creating in silico microbiome models; organ-resolved, sex-specific whole-body metabolic reconstructions for the male and female WBM Harvey_1_04b and Harvetta_1_04c (https://www.digitalmetabolictwin.org/copy-of-reconstructions), creating (personalized) WBMs; the Virtual Metabolic Human database (https://vmh.life/), identifying reactions carried out by corresponding proteins.

## Code availability

Analyses were performed using publicly available software: for variant and gene annotation, VEP version 101 (https://www.ensembl.org/info/docs/tools/vep/index.html) with plugins REVEL version 2020-5, CADD version 3.0, LOFTEE version 2020-8, dbNSFP version 4.1a; for rare variant aggregation testing (burden test), seqMeta R package version 1.6.7 (https://rdrr.io/cran/seqMeta/); for GWAS of common variants and metabolites to compare with rare variant results, REGENIE version 2.2.4 (https://rgcgithub.github.io/regenie/); for Firth regression, the brglm2 R package (https://cran.r-project.org/web/packages/brglm2/index.html); for gene-based testing in the UKB, REGENIE version 3.3; for creating in silico microbiome models and whole-body modeling, COBRA Toolbox version 3.4 (https://opencobra.github.io/cobratoolbox/stable/index.html); for in silico whole-body LP modeling, the stoichiometrically constrained modeling (PSCM) toolbox (https://github.com/opencobra/cobratoolbox/tree/master/src/analysis/wholeBody/PSCMToolbox); source codes for personalized whole-body modeling, constraint settings, creating microbiome community models, personalized WBMs, performing curation steps and simulations for in silico LP and QP modeling (https://github.com/SysPsyHertel/CodeBase/tree/main/Scripts_Scherer_WBM); for comparing in silico gene knockouts with linear regressions for panel data, the plm R package version 2.6-4 (https://cran.r-project.org/web/packages/plm); for general coding, R (versions 3.6.3 and 4.0.5) was used. Detailed information on used software is also provided in the respective sections of the Methods.

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

## Acknowledgements

We thank N. Sallee for supporting the collaboration with Maze and for providing scientific insights about the functional experiments on SLC6A19. The work of N.S., M.S., O.B., S.H., P. Schlosser, M.W., M.K., P. Sekula and A.K. was funded by German Research Foundation (DFG) project ID 431984000 (SFB 1453). N.S. and Y.L. were supported by DFG KO 3598/4-2 (to A.K.). Germany's Excellence Strategy (Centre for Integrative Biological Signalling Studies, EXC-2189, project ID 390939984) supported the work of P. Schlosser, M.K., M.S. and A.K. The work of J.H., D.F. and A.K. was supported by DFG project ID 499552394 (SFB 1597/1). S.P. was funded by H2020 MSCA-ITN-2019 ID:860977 (TrainCKDis). P. Sekula was supported by DFG SE 2407/3-1. The work of U.T.S. was supported by the German Federal Ministry of Education and Research (BMBF) within the framework of the e:Med research and funding concept (grant 01ZX1912B). The work of P. Schlosser was supported by DFG project ID 1050086601 (SCHL 2292/2-1). I.T. was funded by the European Research Council under the European Union's Horizon 2020 research and innovation program (757922), Science Foundation Ireland under grant number 12/RC/2273-P2 and a Horizon Europe grant (101080997). Genotyping and urine metabolomics in the GCKD study were supported by Bayer Pharma. WES was funded by AstraZeneca. Plasma metabolomics has received funding from the Innovative Medicines Initiative 2 Joint Undertaking (JU) under grant agreement 115974. The JU receives support from the European Union's Horizon 2020 research and innovation program and the EFPIA and the JDRF. Any dissemination of results reflects only the authors' view; the JU is not responsible for any use that may be made of the information it contains. Mechanistic studies were funded and executed by Maze Therapeutics. The GCKD study was and is supported by the BMBF (FKZ 01ER 0804, 01ER 0818, 01ER 0819, 01ER 0820 and 01ER 0821) and the KfH Foundation for Preventive Medicine. Unregistered grants to support the study were provided by corporate sponsors (listed at https://gckd.org). We are grateful for the willingness of the patients to participate in the GCKD study. The enormous effort of the study personnel at the various regional centers is highly appreciated. We thank the large number of nephrologists who provide routine care for the patients and collaborate with the GCKD study. The GCKD investigators are listed in the Supplementary Note. We thank the staff at the Computing Centre of the University of Greifswald for their support in using the high-performance computing services of the University of Greifswald.

## Author contributions

Design of the study: A.K. and J.H. Recruitment and management of the study: K.-U.E., A.K., H.M. and U.T.S. Processing of WES data: M.W. Metabolite quantification and interpretation: N.S., Y.C., P. Schlosser and P.J.O. Bioinformatics and statistical analysis: A.K., D.F., J.H., I.T., A.H., M.W., N.S., O.B. and P. Schlosser. Clinical genetics: F.T., U.B., S.C.G. and M.S. SLC6A19 transport studies: C.W., S.M., Y.X., R.G. and K.E. Interpretation of results: A.K., D.F., J.H., I.T., N.S., O.B., P. Schlosser, P. Sekula, M.W., Y.L., Y.C., F.T., U.B., S.C.G., F.K., M.K., M.S., P.J.O., S.P., A.H. and S.H. Wrote the paper: A.K., D.F., J.H. and N.S. Critically read and approved the paper: N.S., D.F., O.B., Y.C., P. Schlosser, M.W., S.P., S.H., A.H., H.M., F.T., U.B., S.C.G., P. Sekula, U.T.S., Y.L., M.K., P.J.O., F.K., K.-U.E., I.T., M.S., J.H. and A.K. A.K. and J.H. jointly supervised this project.

## FundingInformation

## Competing interests

C.W., S.M., Y.X., R.G. and K.E. are employees of and own shares in Maze Therapeutics. A.K. reports a sponsored research collaboration agreement with Maze Therapeutics. The other authors declare no competing interests.

## Additional information

**Extended data** is available for this paper at https://doi.org/10.1038/s41588-024-01965-7.

**Correspondence and requests for materials** should be addressed to Johannes Hertel or Anna Köttgen.

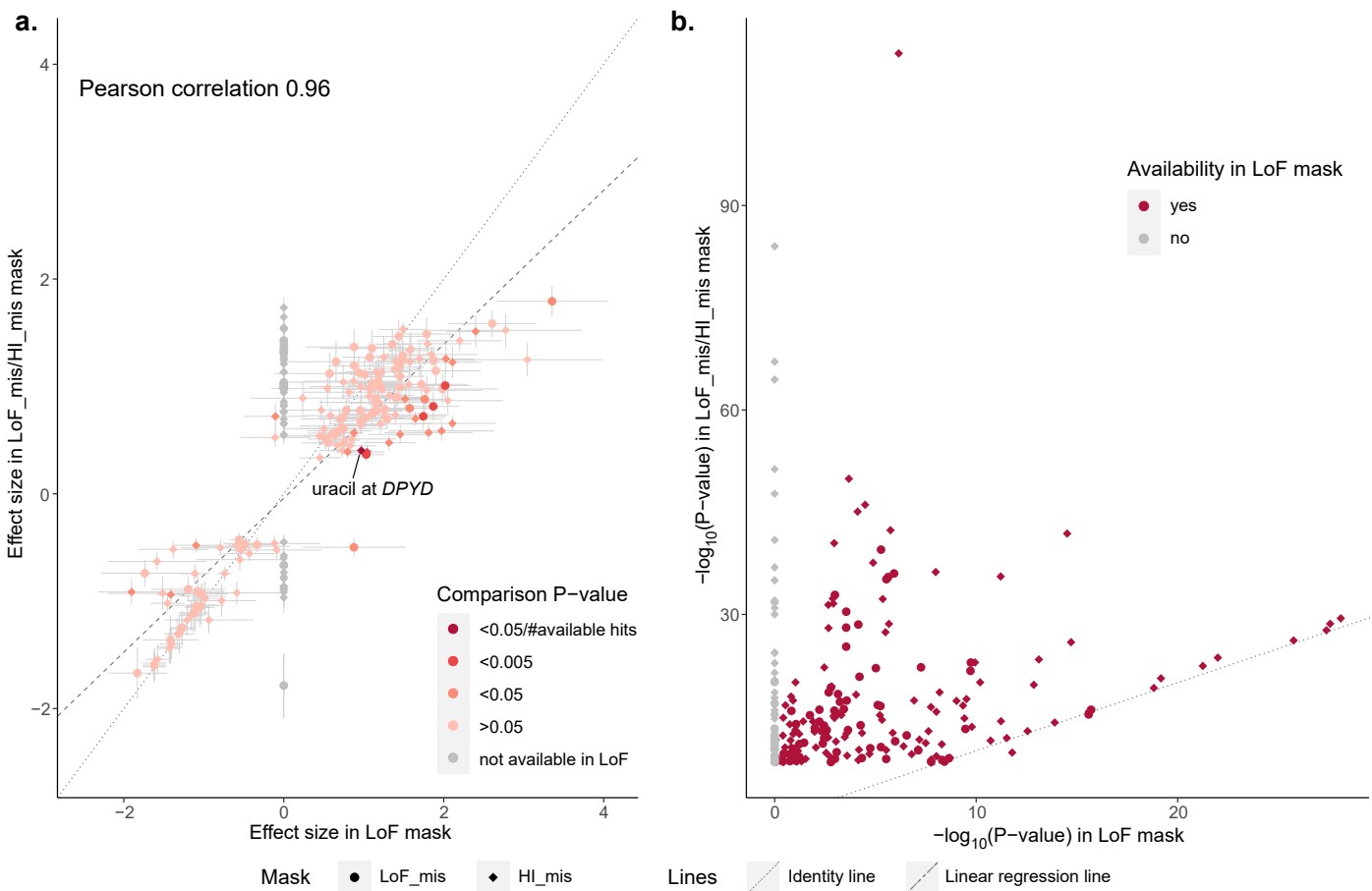

**Extended Data Fig. 1 | Comparison of gene-metabolite associations based on the main analysis masks LoF_mis/HI_mis with a LoF only mask in the GCKD study. (a)** Effect size of gene-metabolite associations using burden tests based on the LoF_mis/HI_mis masks (y-axis) vs. those based on the LoF mask (x-axis) (Supplementary Table 4). Error bars represent the standard errors of the effect sizes. Symbol shape indicates the corresponding mask. Color reflects the comparison P-value between effect sizes based on the test statistic Z (see Supplementary Methods), indicating that the effect sizes between the LoF_mis/HI_mis and the LoF mask differ significantly (P-value < 0.05/178 adjusted

for the number of associations available based on the LoF mask) just for one association of *DPYD* with uracil. Gray symbols reflect the 57 gene-metabolite pairs for which no gene-based test could be performed based on the LoF mask. Gray lines represent the identity (dotted) and the linear regression line (dashed). **(b)** -$\log_{10}$(P-value) of gene-metabolite associations using burden tests based on the LoF_mis/HI_mis masks (y-axis) vs. those based on the LoF mask (x-axis). Symbol shape indicates the corresponding mask. Color reflects the availability of gene-metabolite pairs based on the LoF mask. The gray dotted line represents the identity line.

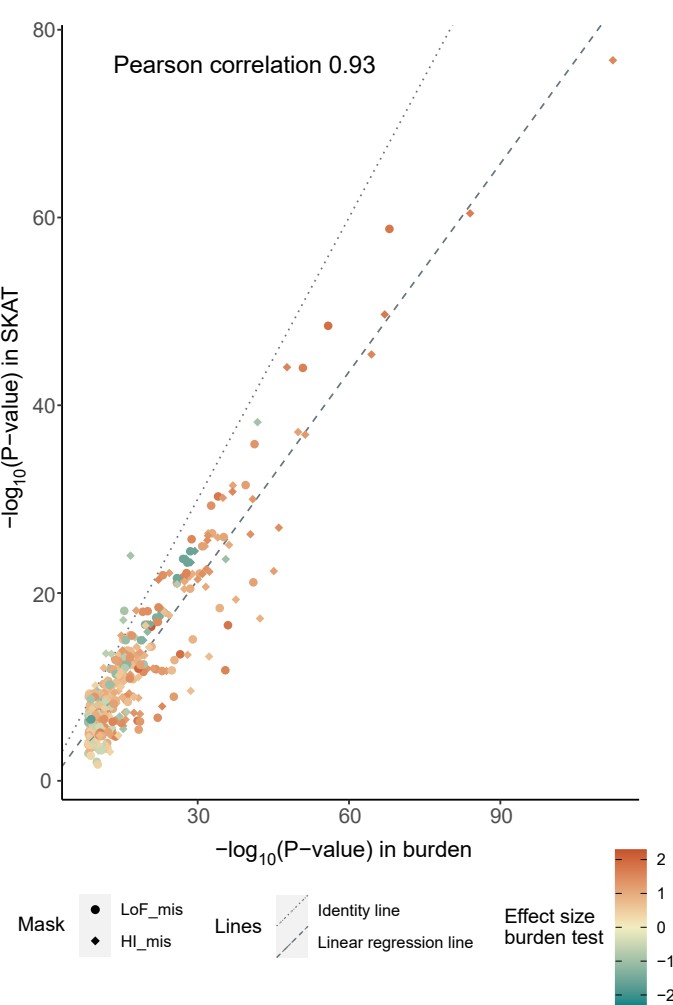

**Extended Data Fig. 2 | Comparison of P-values of significant gene-metabolite associations between burden and SKAT tests in the GCKD study.** The -log$_{10}$ (P-value) of gene-metabolite associations based on the burden test (x-axis) vs. those based on the SKAT test (y-axis) (Supplementary Table 4). Symbol shape indicates the corresponding mask. Color reflects the effect size provided by the burden test. Gray lines represent the identity (dotted) and the linear regression line (dashed).

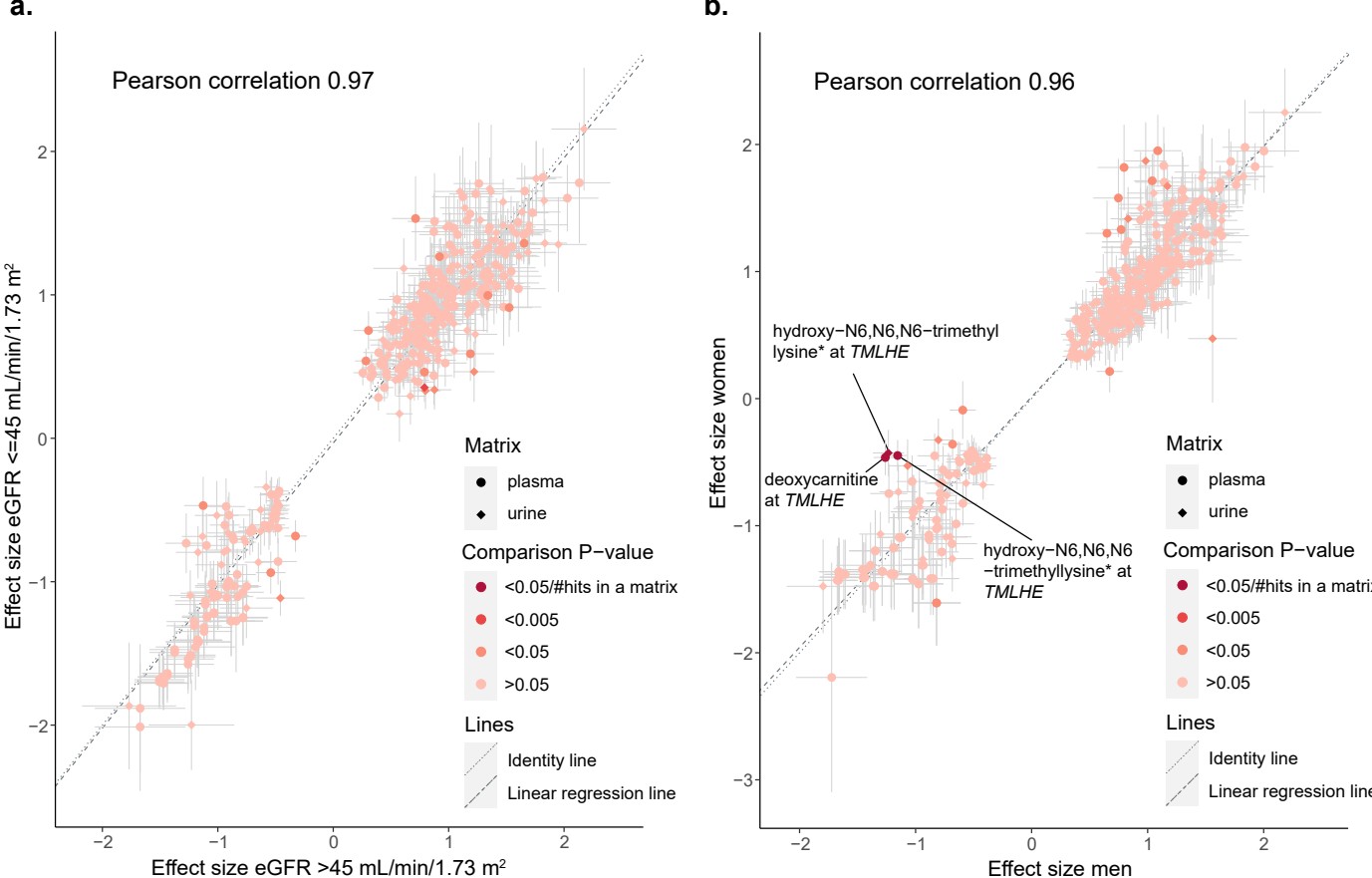

**Extended Data Fig. 3 | Comparison of effect sizes across strata of eGFR and sex in the GCKD study. (a)** Effect size of gene-metabolite associations among individuals with eGFR >45 mL/min/1.73 m² (N = 2,528, x-axis) vs. those with eGFR <=45 mL/min/1.73 m² (N = 2,185, y-axis) (Supplementary Table 5). Error bars represent the standard errors of the effect sizes. Symbol shape indicates the corresponding matrix. Color reflects the comparison P-value between both strata, indicating that effect sizes between individuals with high and low eGFR did not significantly differ for any gene-metabolite association, defined as P-value < 0.05/128 for plasma and P-value < 0.05/107 for urine. Gray lines represent the identity (dotted) and the linear regression line (dashed).

**(b)** The effect size of gene-metabolite associations among men (N = 2,837, x-axis) and women (N = 1,876, y-axis). Error bars represent the standard errors of the effect sizes. Symbol shape indicates the corresponding matrix. Color reflects the comparison P-value between both strata. Gene-metabolite associations with effect sizes that significantly differ between men and women (P-value < 0.05/128 for plasma or <0.05/107 for urine) are labeled and are exclusively observed for the X-chromosomal *TMLHE* gene, where hemizygous men show more extreme metabolite levels than heterozygous women. Gray lines represent the identity (dotted) and the linear regression line (dashed).

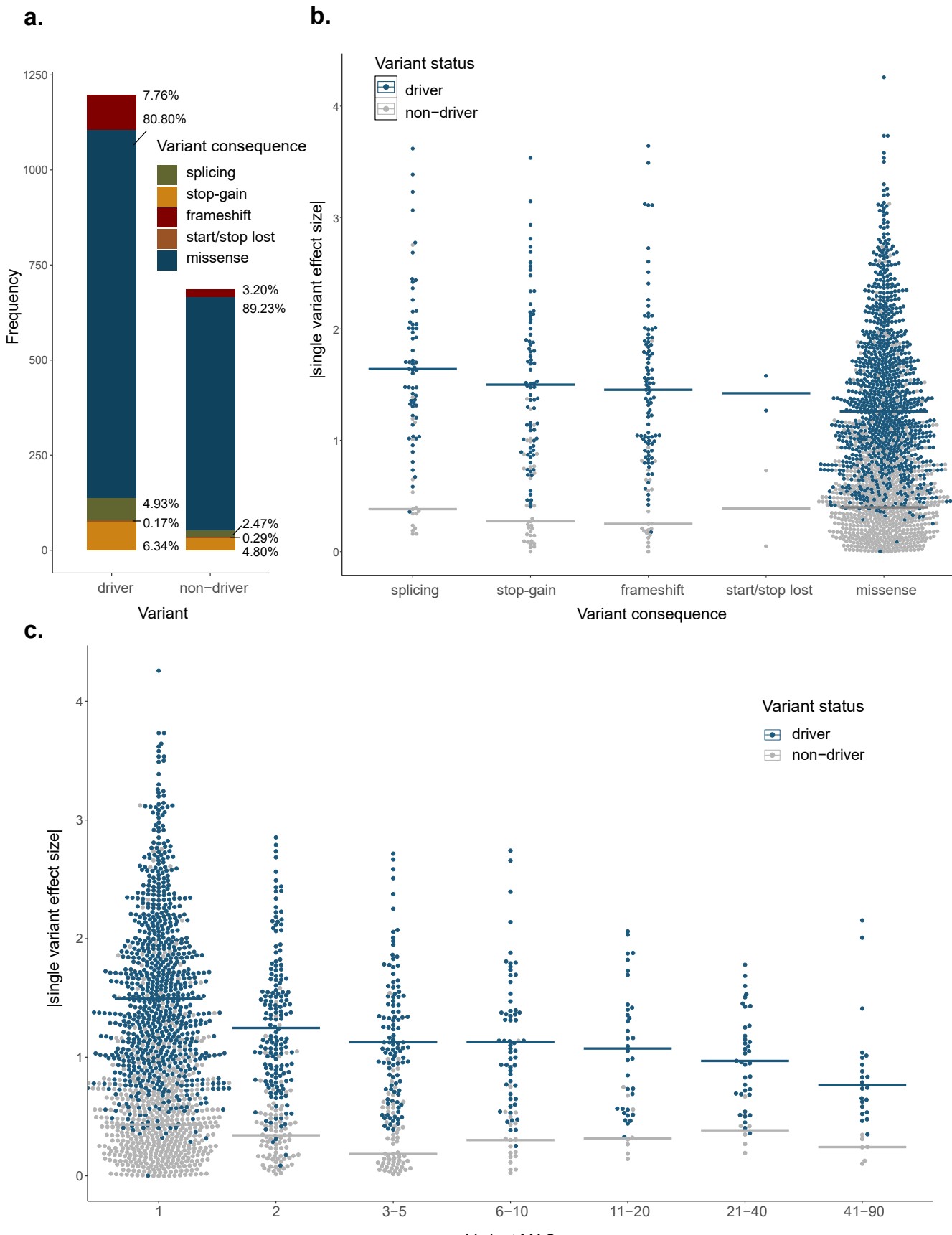

**Extended Data Fig. 4 | See next page for caption.**

**Extended Data Fig. 4 | Driver variants show a more severe impact on metabolite levels compared with non-drivers in terms of consequence and effect size. (a)** The bar plot represents the absolute frequency (y-axis) of each of the QVs' consequences with their proportions noted next to them, separately for driver and non-driver variants (x-axis). Whereas driver variants contain more splicing, stop-gain and frameshift variants, the proportion of missense variants is higher among non-driver variants (Fisher's exact test: P-value = 1.3e-6). **(b)** The swarm plot shows differences in absolute effect sizes for QVs (y-axis) across the five different consequence classes (x-axis). The color reflects the variant status (driver versus non-driver variant). Horizontal lines represent the median of the absolute effect sizes separately for driver and non-driver variants. The median effect of driver variants on metabolite levels increases when ordering

the consequence classes with respect to severity (missense, start/stop-lost, frameshift, stop-gain, splicing). **(c)** The swarm plot shows differences in absolute effect sizes for QVs (y-axis) across groups of variants by minor allele count (MAC) bins (x-axis). Color reflects variant status (driver versus non-driver). Horizontal lines represent the median of the absolute effect sizes separately for driver and non-driver variants. The median effect of driver variants on metabolite levels increases with decreasing MAC bin, supporting that ultra-rare variants tend to have larger effects than those observed more frequently. In case one gene was significantly associated with levels of more than one metabolite, only the QVs from the strongest gene-metabolite associations are included (for only one matrix and only one mask) in each panel, to prevent counting variants multiple times, resulting in 1,885 QVs that were included in each panel.

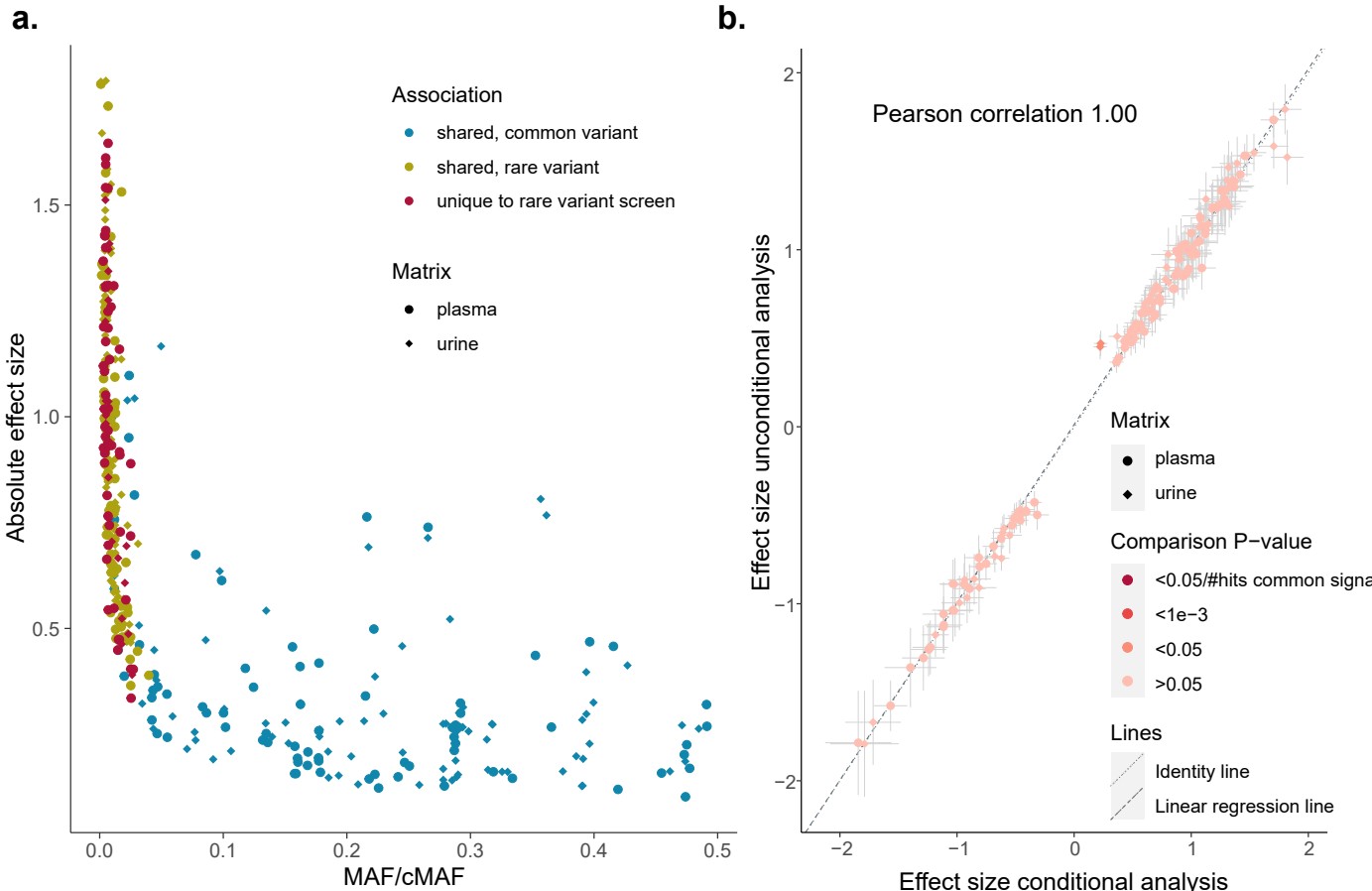

**Extended Data Fig. 5 | Comparison and integration of rare and common variant association signals with metabolite levels within the same locus.**
**(a)** The scatter plot shows the absolute effect size (y-axis) of association signals with metabolite levels based on aggregating rare variants within a gene using burden tests and based on the common variant with the lowest significant individual association P-value within the same locus (±500 kb around the gene) under additive modeling (Supplementary Methods), across different cumulative minor allele frequencies (cMAF, for aggregated rare variants) and minor allele frequencies (MAF, for common variants) (x-axis). Colors indicate whether the corresponding association signal is based on shared rare or common variants or whether it is unique to the rare variant screen. The shape represents the matrix

of the corresponding metabolite. The absolute effect size tends to increase with decreasing MAF/cMAF. **(b)** Effect size of 157 gene-metabolite associations with conditioning on associated common variants within the gene region (x-axis) vs. without conditioning on common variants (y-axis) (Supplementary Table 9). Error bars represent the standard errors of the effect sizes. Symbol shape indicates the corresponding matrix. Color reflects the comparison P-value between both analyses, indicating that effect sizes between (un)conditional analyses did not significantly differ for any gene-metabolite association, defined as P-value < 0.05/157. Gray lines represent the identity (dotted) and the linear regression line (dashed).

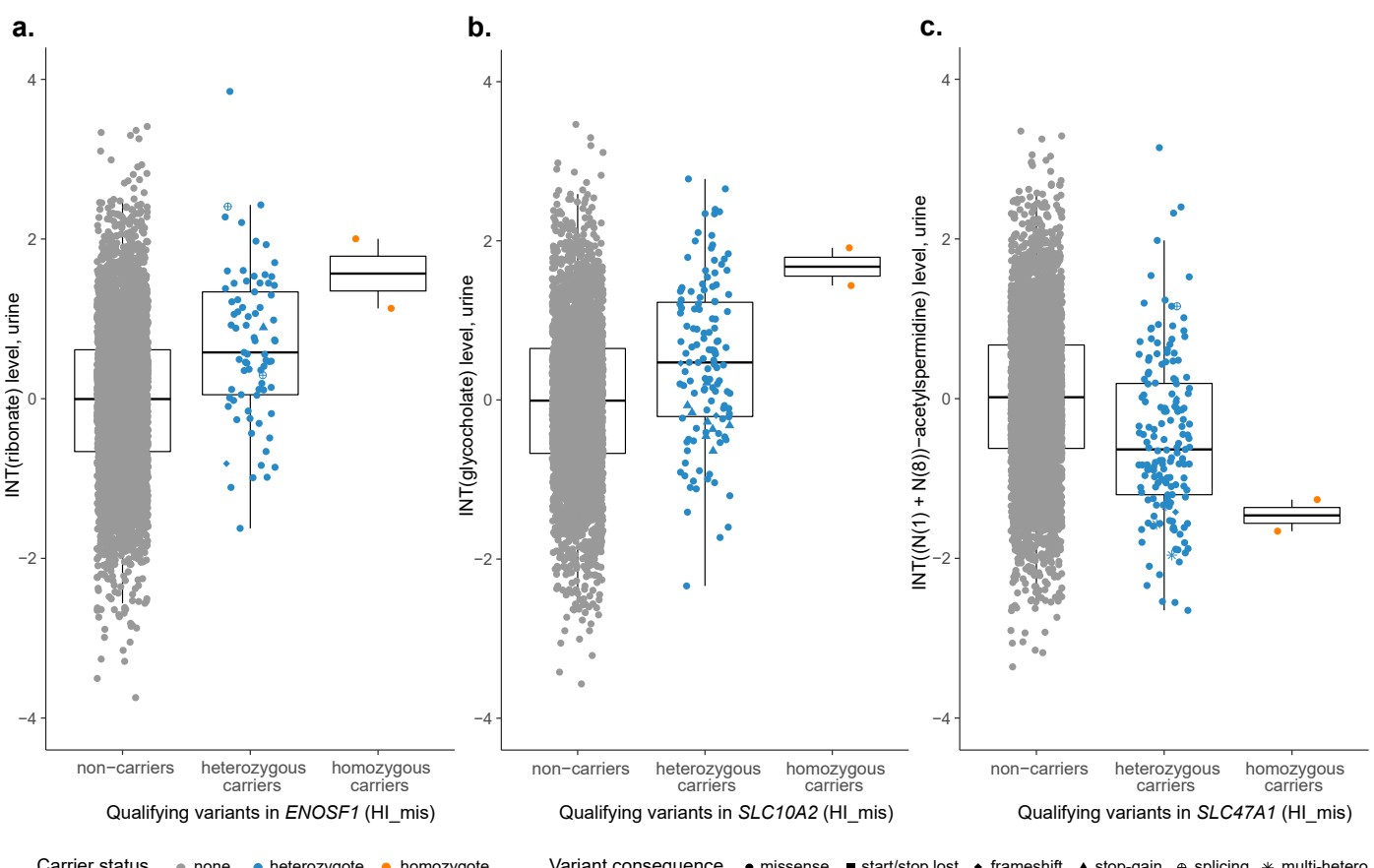

**Extended Data Fig. 6 | Metabolite levels by QV carrier status for significantly associated genes with more than one homozygous carrier.** Urine metabolite levels after inverse normal transformation and covariate-adjustment are shown on the y-axis, among non-carriers and heterozygous and homozygous carriers of QVs in the HI_mis mask on the x-axis. Symbol color and shape indicate a variant's carrier status and consequence, respectively. Carriers of multiple heterozygous QVs are denoted by an asterisk. Boxes range from the 25th to the 75th percentile of metabolite levels, the median is indicated by a line, and whiskers end at the last observed value within 1.5*(interquartile range) away from the box. The median levels of ribonate (N = 4,618) **(a)**, glycocholate (N = 3,753) **(b)**, and (N(1) + N(8))–acetylspermidine (N = 4,619) **(c)** are all more extreme for the homozygous than the heterozygous carriers, reflecting a dose-response effect.

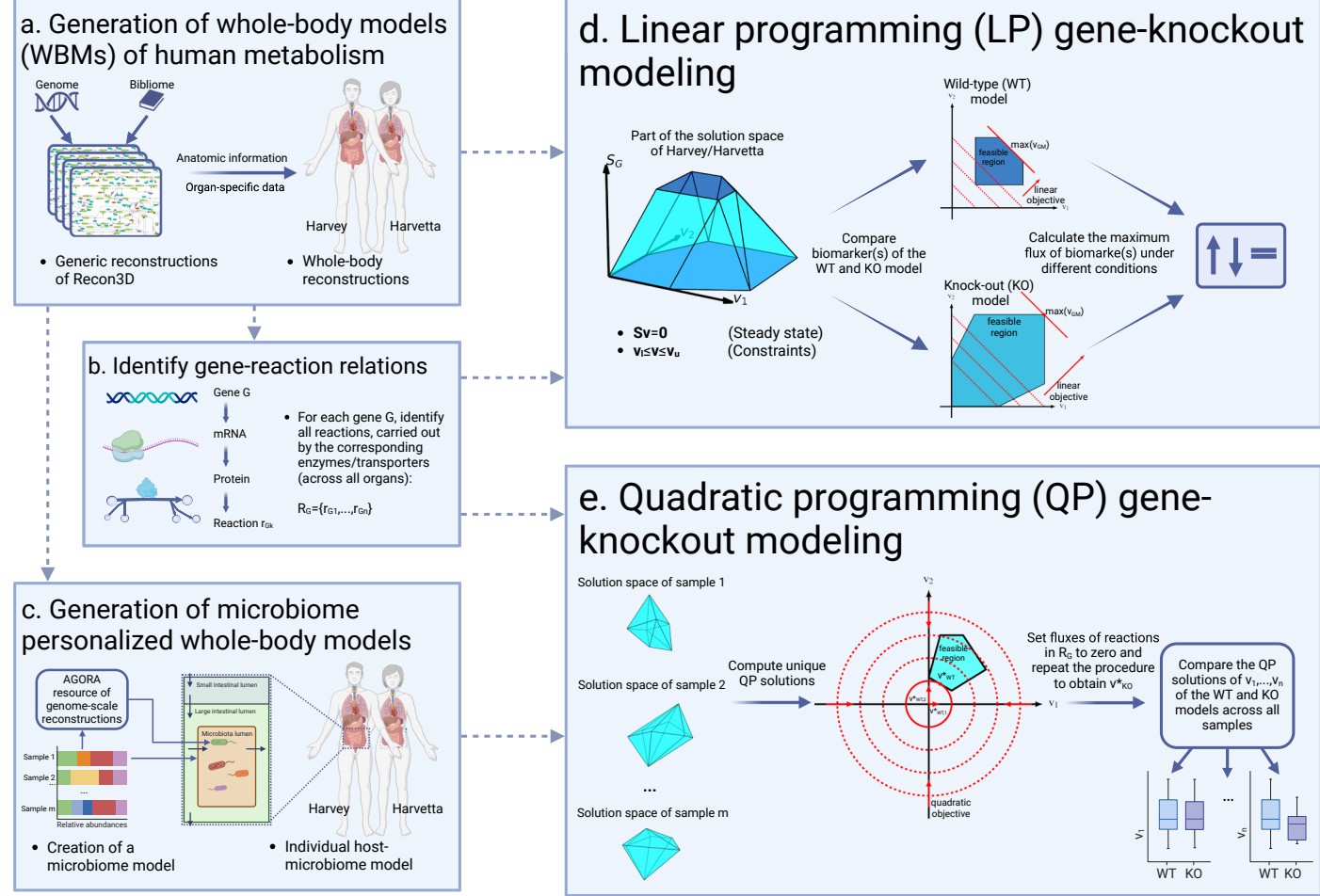

**Extended Data Fig. 7 | Illustration of microbiome-personalized whole-body models of *in silico* knockout modeling.** (a) Generation of whole-body models (WBMs) of human metabolism: Generic reconstructions of human metabolism of Recon3D are used and pruned by organ-specific data and anatomical information, to form differentiated, organ-specific male and female models of human metabolism. The models are derived from Thiele et al. 2020, PMID: 32463598. **(b)** Identify gene-reaction relations: For each Gene *G*, it is essential to identify every reaction carried out by the corresponding enzyme/transporter across all organs. The corresponding reactions $R_G = \{r_{G_1}, \ldots, r_{G_n}\}$ are used for gene-knockout modeling in the linear programming (LP) and quadratic programming (QP) knockout methodologies. **(c)** Generation of microbiome-personalized whole-body models: Utilizing abundance-data of metagenomic samples, personalized gut microbiome community models are generated via combining the individually present genome-scale reconstructions of microbes in a common lumen compartment. The lumen compartment is then integrated with the whole-body model, thereby creating an individualized host-microbiome model. **(d)** LP gene-knockout modeling: The system of mass balance equations at steady state, as defined by the stoichiometric matrix $S$ and constraints for each flux, forms a convex space of possible solutions for a vector $v$ of fluxes of a WBM. The sum of fluxes of reactions in $R_G$ (expressed as $S_G = v_{G_1} + \ldots + v_{G_n}$) is maximized, and this maximal value is subsequently

imposed as a constraint in the wild-type model. Setting all fluxes corresponding to the reactions in $R_G$ to zero forms the knockout model. The maximum flux of a biomarker $v_{GM}$ is computed in both models and compared with each other. **(e)** QP gene-knockout modeling: Each microbiome-personalized whole-body model spans up its individual wild-type solution space under steady state and constraint settings. The solution of the quadratic objective of the flux vector under steady state and constraints for each reaction of the WBM forms the wild-type solution for Harvey/Harvetta, denoted as $v^*_{WT} = (v^*_{WT,1}, \ldots, v^*_{WT,n})$. Setting each flux of reaction in $R_G$ to zero and repeating the procedure yields the QP solution after knockout, $v^*_{KO}$. The unique QP solutions across all samples allows for screening for differences within each reaction of the WBM to find possible biomarkers, and for the computation of effect sizes for each reaction. The visualization of generic reconstructions of Recon3D of Extended Data Fig. 7 was generated using a screenshot of ReconMap3 (https://www.vmh.life/#reconmap) and further edited with Inkscape (https://inkscape.org/). The 3D visualizations in Extended Data Fig. 7 and Extended Data Fig. 7 were produced using Python (https://www.python.org/), while the 2D images with the coordinate systems were created using LaTeX (https://www.latex-project.org/). The remaining subfigures, along with the final composition of the complete figure, were created using BioRender.com, integrating all previously mentioned elements. Figure created with BioRender.com.

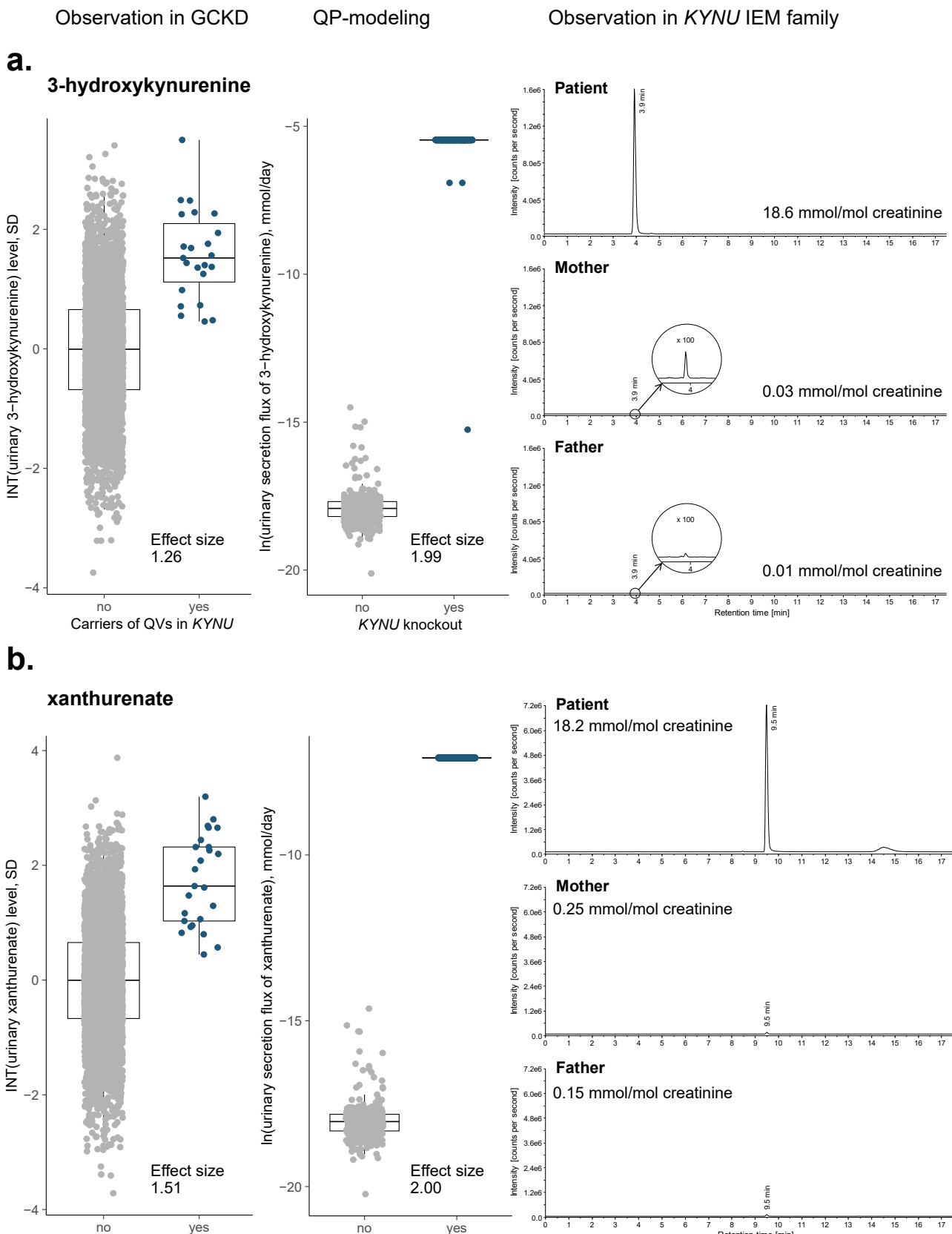

**Extended Data Fig. 8 | See next page for caption.**

**Extended Data Fig. 8 | Elevated urine levels of 3-hydroxykynurenine and xanthurenate are a readout of impaired *KYNU* function: converging evidence from three approaches.** Three panels are shown for 3-hydroxykynurenine **(a)** and xanthurenate **(b)** each: the left panel represents inverse-normal transformed, covariate-adjusted urine levels of the respective metabolite (y-axis) among non-carriers and carriers of QVs in *KYNU* (x-axis). Units correspond to standard deviations. The boxes range from the 25th to the 75th percentile of metabolite levels, the median is indicated by a line, and whiskers end at the last observed value within 1.5*(interquartile range) away from the box. The middle panel represents the distribution of the ln-transformed urinary secretion flux of the respective metabolite in mmol/day into urine (y-axis) from min-norm simulations based on solutions of 569 microbiome-personalized whole-body models without and with simulated knockout of *KYNU* (x-axis). The right panel shows multiple reaction monitoring (MRM, $m/z$ 225.0 → 162.1, 206.0 → 160.1) chromatograms of the diluted urines of a child with a homozygous, autosomal-recessively inherited loss of *KYNU* function (patient), the mother and the father. The signals at 3.9 min (3-hydroxykynurenine) and 9.5 min (xanthurenate) are strongly enhanced in the patient sample. Chromatograms are normalized to urine creatinine concentrations; y-axes are normalized to the intensity of the signals in the patient's chromatograms. All three independent approaches arrive at the conclusion that elevated levels of 3-hydroxykynurenine and xanthurenate in urine are a readout of impaired *KYNU* function.

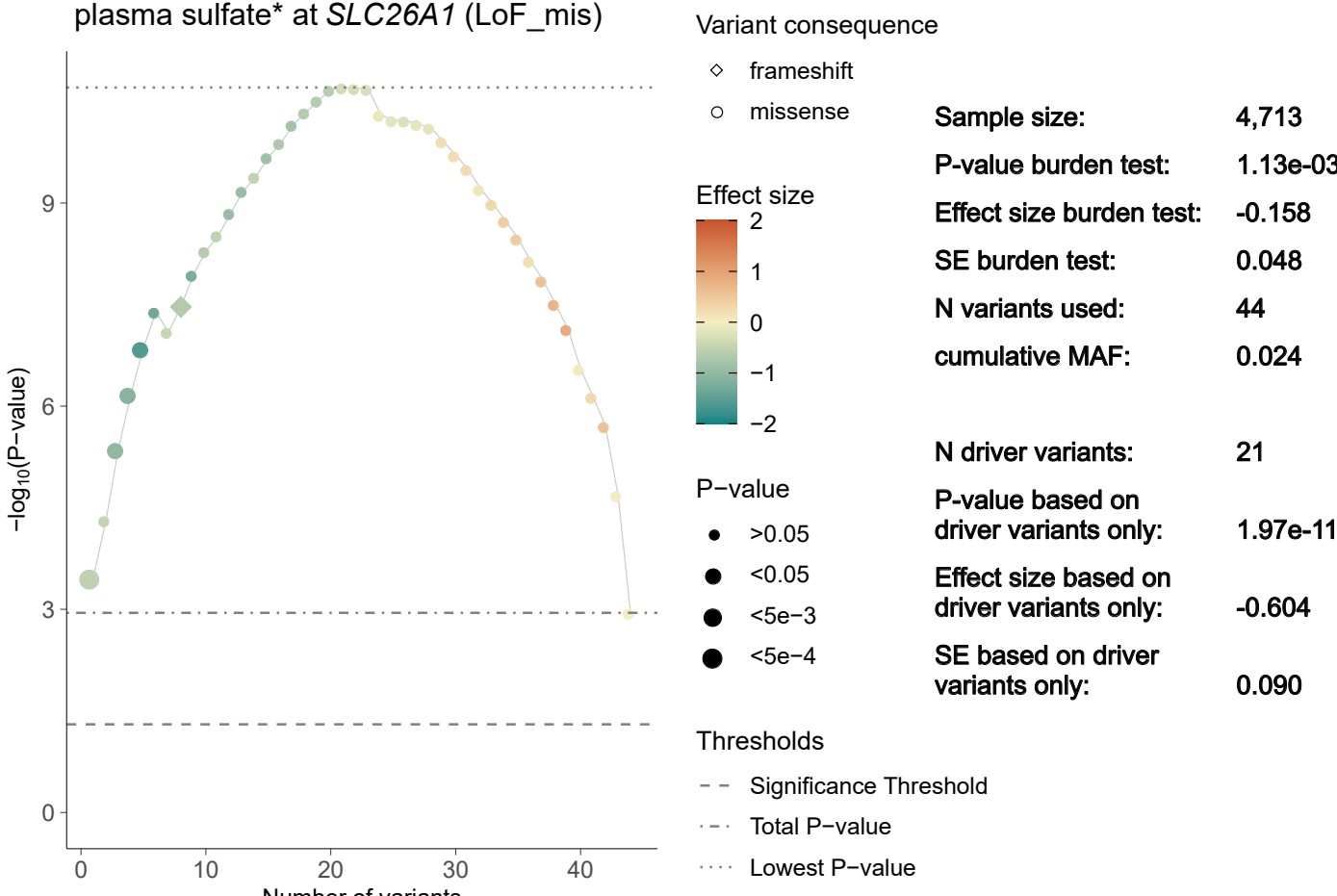

**Extended Data Fig. 9 | Contribution of individual QVs in *SLC26A1* to their gene-based association signal with plasma sulfate levels.** The symbols visualize the -log₁₀(P-value) (y-axis) with regard to plasma sulfate levels for the successive aggregation of the most influential QVs in *SLC26A1* with respect to the forward selection procedure (Bomba *et al*, PMID: 35568032, Methods) based on burden tests for the mask LoF_mis. The number of QVs aggregated for burden testing is shown on the x-axis. Symbol shape indicates the variant's consequence. The symbol color and size reflect the effect size and the P-value of the variant based on its single-variant association test. The gray dashed lines represent the significance threshold (-log₁₀(0.05)), the total -log₁₀(P-value) of the aggregate variant test including all QVs in *SLC26A1* for the mask LoF_mis, and the -log₁₀(lowest P-value) that can be reached by aggregating only the driver variants from the forward selection procedure. Summary statistics shown on the right refer to the burden tests aggregating all QVs and only driver variants. For the latter, a clear association of *SLC26A1* with plasma sulfate levels is observed.

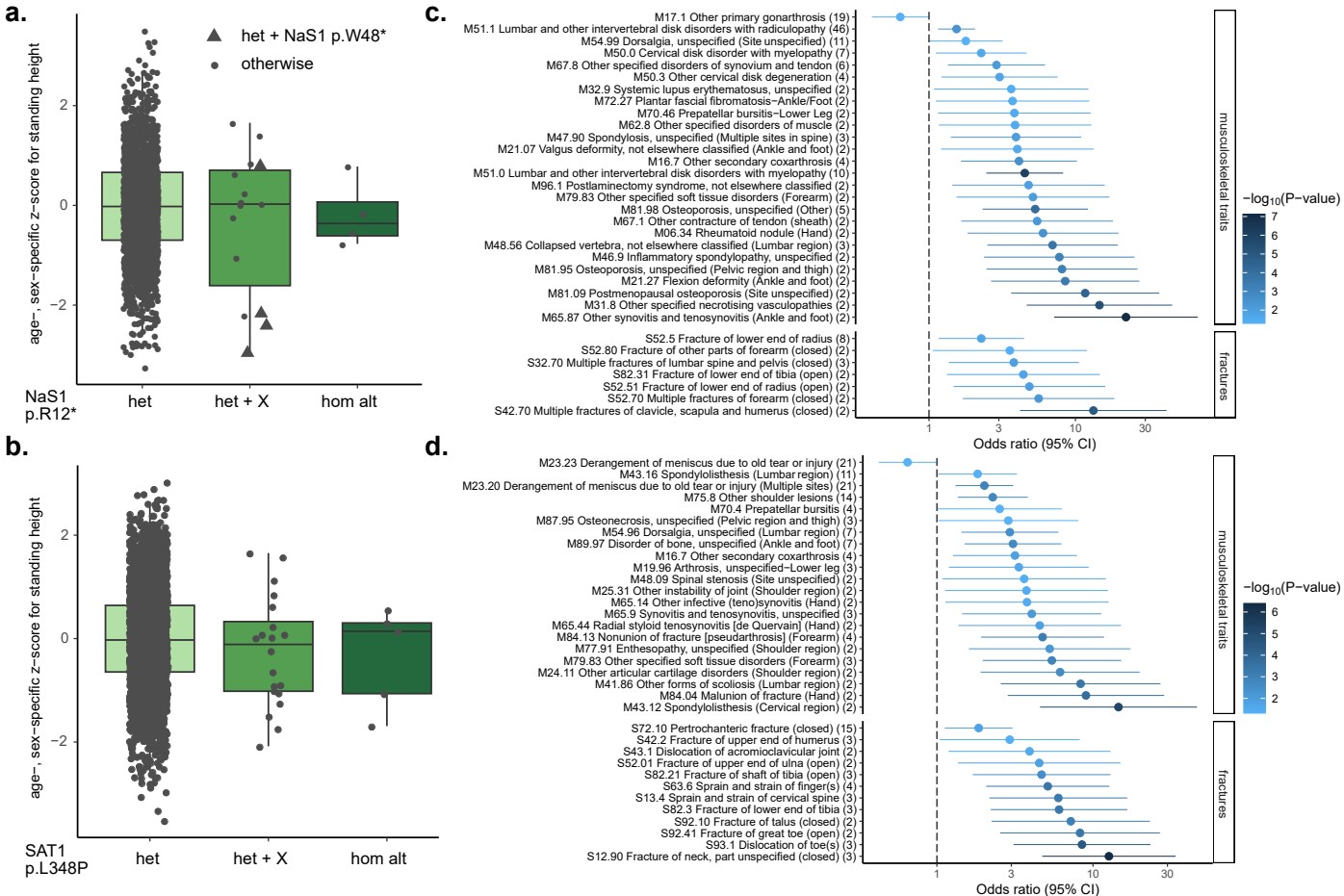

**Extended Data Fig. 10 | Impact of different genotypes encoding NaS1 p.Arg12* and SAT1 p.Leu348Pro on height and musculoskeletal traits and fractures.** The boxplots on the left show differences in age- and sex-specific z-scores for standing height (y-axis, Methods) across individuals in the UKB heterozygous and homozygous for the NaS1 p.Arg12*-encoding allele **(a)** and for the SAT1 p.Leu348Pro-encoding allele **(b)** (x-axis). The boxes range from the 25th to the 75th percentile of z-scores, the median is indicated by a line, and whiskers end at the last observed value within 1.5*(interquartile range) away from the box. Heterozygous individuals carrying only NaS1 p.Arg12* (N = 2,460) and SAT1 p.Leu348Pro (N = 3,096), respectively, are depicted in the 'het' category. Individuals carrying a variant at two different DNA positions are shown in the category 'het + X' (N = 15 for NaS1 p.Arg12* and N = 20 for SAT1 p.Leu348Pro).

For the NaS1 p.Arg12* stop-gain variant, multi-heterozygous individuals, who additionally carry the NaS1 p.Trp48* stop-gain variant, are indicated with differently shaped symbols, emphasizing that carrying two stop-gain variants in NaS1 seems to lead to a more severe phenotype. The forest plots on the right show associations between the NaS1 p.Arg12* **(c)** and SAT1 p.Leu348Pro **(d)** carrier status with musculoskeletal diseases and fractures from the UKB (N ≥ 468,279), for which at least 2 carriers were identified among individuals with and without disease (y-axis). Number in parentheses indicate the number of carriers with a given disease. Odds ratios and their corresponding 95% confidence interval (x-axis) are based on Firth regression (Methods). The symbol color reflects the -log₁₀(P-value). Only associations with P-value < 0.05 are shown.

## Reporting Summary

## Statistics

For all statistical analyses, confirm that the following items are present in the figure legend, table legend, main text, or Methods section.

| n/a | Confirmed | |
|---|---|---|
| ☐ | ☒ | The exact sample size (*n*) for each experimental group/condition, given as a discrete number and unit of measurement |
| ☐ | ☒ | A statement on whether measurements were taken from distinct samples or whether the same sample was measured repeatedly |
| ☐ | ☒ | The statistical test(s) used AND whether they are one- or two-sided *Only common tests should be described solely by name; describe more complex techniques in the Methods section.* |
| ☐ | ☒ | A description of all covariates tested |
| ☐ | ☒ | A description of any assumptions or corrections, such as tests of normality and adjustment for multiple comparisons |
| ☐ | ☒ | A full description of the statistical parameters including central tendency (e.g. means) or other basic estimates (e.g. regression coefficient) AND variation (e.g. standard deviation) or associated estimates of uncertainty (e.g. confidence intervals) |
| ☐ | ☒ | For null hypothesis testing, the test statistic (e.g. *F*, *t*, *r*) with confidence intervals, effect sizes, degrees of freedom and *P* value noted *Give P values as exact values whenever suitable.* |
| ☒ | ☐ | For Bayesian analysis, information on the choice of priors and Markov chain Monte Carlo settings |
| ☒ | ☐ | For hierarchical and complex designs, identification of the appropriate level for tests and full reporting of outcomes |
| ☐ | ☒ | Estimates of effect sizes (e.g. Cohen's *d*, Pearson's *r*), indicating how they were calculated |

*Our web collection on statistics for biologists contains articles on many of the points above.*

## Software and code

Policy information about availability of computer code

| Data collection | The data for the GCKD study was collected using the software Askimed (https://www.askimed.com/). |
|---|---|
| Data analysis | - Software-tools for processing of whole-exome sequencing data: Illumina DRAGEN Bio-IT Platform Germline Pipeline v3.0.7 at Astra Zeneca's Centre for Genomics Research<br>Software-tools for QC of whole-exome sequencing data: KING --kinship v2.2.3<br>- Software-tools for variant and gene annotation: Variant Effect Predictor (VEP) v101 with plugins REVEL v2020-5, CADD v3.0, LoFtee v2020-8, dbNSFP v4.1a, gnomAD v 2.1<br>- Software-tools for rare variant aggregation testing in the GCKD study: seqMeta R-package v1.6.7<br>- Software-tools for GWAS of metabolites to compare with rare variant results: REGENIE v2.2.4<br>- Software-tools for Firth regression: R-package "brglm2"<br>- Software-tools for gene-based testing in the UKB: REGENIE v3.3<br>- Software-tools for in silico whole-body modeling: COBRA Toolbox v3.4 (https://opencobra.github.io/cobratoolbox/stable/index.html), Matlab 2019b and 2021a, Ilog Cplex v12.9 and v12.10, physiologically and stoichiometrically constrained modeling (PSCM) toolbox (https://github.com/opencobra/cobratoolbox/tree/master/src/analysis/wholeBody/PSCMToolbox), R-package "plm" for regression of panel data.<br>- Source codes for personalized whole-body modelling: https://github.com/SysPsyHertel/CodeBase/tree/main/Scripts_Scherer_WBM<br>- Data bases, publicly available: Ensembl VEP tool, GTEx Project, Ensembl Biomart, AstraZeneca PheWAS Portal, OMIM catalog, Genomics England PanelApp v4.0, Open Targets Platform, ClinVar archive, Virtual Metabolic Human database<br>- Miscellaneous: R v3.6.3 and v4.0.5<br>References or website addresses are provided in the manuscript. |

For manuscripts utilizing custom algorithms or software that are central to the research but not yet described in published literature, software must be made available to editors and reviewers. We strongly encourage code deposition in a community repository (e.g. GitHub). See the Nature Portfolio guidelines for submitting code & software for further information.

March 2021

# Data

Policy information about availability of data

All manuscripts must include a data availability statement. This statement should provide the following information, where applicable:
  - Accession codes, unique identifiers, or web links for publicly available datasets
  - A description of any restrictions on data availability
  - For clinical datasets or third party data, please ensure that the statement adheres to our policy

Data preparation, quality control, data modeling and statistical analyses of the data presented in this manuscript were performed at the Institute of Genetic Epidemiology, Medical Center - University of Freiburg, Freiburg (Germany) and at the Department of Psychiatry and Psychotherapy, University Medicine Greifswald, Greifswald (Germany), unless otherwise mentioned in the Methods.

The summary statistics of all significant gene-metabolite associations based on burden tests using two masks as well as all involved QVs with annotations are available in Supplementary Table 3 and Supplementary Tables 7a, b, respectively. Genotype, metabolite, protein and phenotype data were obtained from the UKB (https://www.ukbiobank.ac.uk/) and the GCKD study (https://www.gckd.org/). This research has been conducted using the UK Biobank Resource under Application Number 64806.

The following external data sources were used: GRCh38 reference genome (https://ftp.ncbi.nlm.nih.gov/genomes/all/GCA/000/001/405/ GCA_000001405.15_GRCh38/): alignment of reads; GTEx Project (https://gtexportal.org/home/): investigation of gene expression and QTLs across tissues; AstraZeneca PheWAS Portal (https://azphewas.com/): search for gene- and variant-level associations of detected genes and QVs; OMIM catalog (https:// www.omim.org/): query for monogenic disorders and traits related to identified genes; Genomics England PanelApp (https://panelapp.genomicsengland.co.uk/ panels/467/ version v4.0): search for known IEM related to the detected genes; Open Targets Platform (https://platform.opentargets.org/): search for drug target status and corresponding indication for identified genes; ClinVar archive (https://www.ncbi.nlm.nih.gov/clinvar/): query for clinical significance and corresponding trait/disease of detected QVs. Microbiome abundance data (https://static-content.springer.com/esm/art%3A10.1038%2Fs41591-019-0458-7/ MediaObjects/41591_2019_458_MOESM3_ESM.xlsx) and the AGORA resource of genome-scale microbial reconstructions (https://github.com/ VirtualMetabolicHuman/AGORA/): Creating in silico microbiome models; Organ-resolved, sex-specific whole-body metabolic reconstructions, Harvey_1_04b and Harvetta_1_04c, which are updated versions of the current public models v1_03c (https://www.digitalmetabolictwin.org/copy-of-reconstructions): Creating (personalized) whole-body models; Virtual Metabolic Human database (https://vmh.life/): identifying reactions carried out by corresponding genes.

# Human research participants

Policy information about studies involving human research participants and Sex and Gender in Research.

| | |
|---|---|
| Reporting on sex and gender | Persons of both sexes were included in all analyses. In the context of this study, biological sex was used.  X chromosomal genetic variants were included in all analyses. |
| Population characteristics | Please see Supplementary Table 1:<br>Characteristics of the GCKD study overall: N=4,737<br>Mean age (SD), years: 60.26 (11.88)<br>Female sex, % (n): 39.75% (1883)<br>Mean BMI (SD), kg/m²: 29.8 (5.97)<br>Mean systolic blood pressure (SD), mm Hg: 139.57 (20.38)<br>Mean Hemoglobin A1c (SD), mmol/mol: 45.82 (11.29)<br>Diabetes, % (n): 35.61% (1687)<br>Mean eGFR (SD), ml/min/1.73m²: 49.42 (18.21)<br>Median urinary albumin-to-creatinine ratio (IQR), mg/g: 49.28 (9.3-375.78)<br>Mean albumin (SD), g/l: 38.33 (4.25) |
| Recruitment | The GCKD study is an ongoing prospective observational cohort study of participants with CKD. Between 2010 and 2012, 5,217 adult persons with CKD under regular care by nephrologists provided written informed consent and were enrolled into the study at nine participating study centers across Germany (see below). Participants were included if they met inclusion criteria, but - as in all epidemiological studies - it cannot be excluded that eligible persons with many or severe comorbidities were less likely to participate than other eligible participants. For this project, all participants with available plasma or urine collected at the baseline visit and with available whole-exome sequencing data were selected (N=4,737). |
| Ethics oversight | The GCKD Study was registered in the national registry for clinical studies (DRKS 00003971) and approved by all local ethic committees of the nine participating centers (Universities or Medical Faculties of Aachen, Berlin, Erlangen, Freiburg, Hannover, Heidelberg, Jena, München, Würzburg). |

Note that full information on the approval of the study protocol must also be provided in the manuscript.

# Field-specific reporting

Please select the one below that is the best fit for your research. If you are not sure, read the appropriate sections before making your selection.

☒ Life sciences          ☐ Behavioural & social sciences          ☐ Ecological, evolutionary & environmental sciences

For a reference copy of the document with all sections, see nature.com/documents/nr-reporting-summary-flat.pdf

# Life sciences study design

All studies must disclose on these points even when the disclosure is negative.

| | |
|---|---|
| Sample size | We included all 4,737 GCKD participants with available plasma or urine metabolite quantification and with available whole-exome sequencing data. The entire sample was therefore utilized, and no selection was made. |
| Data exclusions | Metabolites were excluded for high proportions of missingness (>93%). Samples were excluded if no high-quality whole-exome sequencing data was available. This is clearly described in the methods. |
| Replication | Replication of gene-metabolite associations based on aggregating rare damaging variants is difficult because of the non-availability of the same rare damaging variant in an independent cohort.<br>Therefore, we evaluated reproducibility of our findings by different means :<br>- We compared our identified gene-metabolite associations to those from 8 published studies that focused on rare variant aggregation testing of metabolite levels. For one published study of the plasma metabolome (based on MS-quantification) that made summary statistics accessible we compared effect sizes on the variant- and gene-level.<br>- For overlapping metabolites, we conducted gene-based tests in the UKB using the same variants and tests as in our study.<br>- We investigated the role of our metabolite-associated genes in currently known inborn errors of metabolism (IEM).<br>- We used whole-body models where in silico gene knockouts were modeled to validate our identified gene-metabolite pairs.<br>- We performed a proof-of-concept experimental validation study for an implicated metabolite not yet shown to be involved in the encoded protein's function.<br>When information was available, each of these approaches supported the validity of findings. |
| Randomization | Not relevant to this study because this is an observational study. |
| Blinding | Not relevant to this study because this is an observational study. |

# Reporting for specific materials, systems and methods

We require information from authors about some types of materials, experimental systems and methods used in many studies. Here, indicate whether each material, system or method listed is relevant to your study. If you are not sure if a list item applies to your research, read the appropriate section before selecting a response.

### Materials & experimental systems

| n/a | Involved in the study |
|---|---|
| ☒ | ☐ Antibodies |
| ☐ | ☒ Eukaryotic cell lines |
| ☒ | ☐ Palaeontology and archaeology |
| ☒ | ☐ Animals and other organisms |
| ☐ | ☒ Clinical data |
| ☒ | ☐ Dual use research of concern |

### Methods

| n/a | Involved in the study |
|---|---|
| ☒ | ☐ ChIP-seq |
| ☒ | ☐ Flow cytometry |
| ☒ | ☐ MRI-based neuroimaging |

## Eukaryotic cell lines

Policy information about cell lines and Sex and Gender in Research

| | |
|---|---|
| Cell line source(s) | Chinese Hamster Ovary (CHO) cells with Tetracycline-Regulated Expression (T-REx) system. Cell line obtained from and engineered at Axxam. |
| Authentication | Cells were not authenticated. |
| Mycoplasma contamination | Cell lines previously tested negative for mycoplasma. Cells in this experiment were not tested for mycoplasma. |
| Commonly misidentified lines (See ICLAC register) | No commonly misidentified cell lines were used. |

## Clinical data

Policy information about clinical studies

All manuscripts should comply with the ICMJE guidelines for publication of clinical research and a completed CONSORT checklist must be included with all submissions.

| | |
|---|---|
| Clinical trial registration | This study is an observational study (DRKS 00003971). |
| Study protocol | The study protocol and design has been published (PMID: 21862458). |

| Data collection | Between 2010 and 2012, 5,217 adult persons with CKD under regular care by nephrologists provided written informed consent and were enrolled into the study at nine participating study centers across Germany (Aachen, Berlin, Erlangen, Freiburg, Hannover, Heidelberg, Jena, München, Würzburg).<br><br>Data was collected during GCKD study visits by trained personnel in any of the nine study centers following a published pre-specified protocol and standard operating procedures, and captured with the software Askimed (https://www.askimed.com/).<br><br>For this project, all participants with available plasma or urine collected at the baseline visit and with available whole-exome sequencing data were selected (N=4,737).<br><br>The participants are currently followed for clinical outcomes for more than 10 years. |

| Outcomes | The predefined outcomes of this study were metabolite levels in plasma and urine, which was defined before study initiation by the authors. Non-targeted MS analysis was performed at Metabolon, Inc., from plasma and urine samples collected at the study's baseline visit, as described in detail in the publication. |

