## [Peer Review File · Nature Genetics]

Peer Review Information

Manuscript Title: Coupling metabolomics and exome sequencing reveals graded effects of rare damaging heterozygous variants on gene function and human traits

Corresponding author name(s): Professor Anna Köttgen, Professor Johannes Hertel

Editorial Notes:

Transferred manuscripts This document only contains reviewer comments, rebuttal and decision letters for versions considered at Nature Genetics.

Reviewer Comments & Decisions:

Decision Letter, initial version:

28th Nov 2023

Dear Professor Köttgen,

Your Article, "Coupling of metabolomics and exome sequencing reveals graded effects of rare damaging heterozygous variants on gene function and resulting traits and diseases" has now been seen by 3 referees. You will see from their comments copied below that while they find your work of considerable potential interest, they have raised quite substantial concerns that must be addressed. In light of these comments, we cannot accept the manuscript for publication, but would be very interested in considering a substantially revised version that addresses these serious concerns.

We hope you will find the referees' comments useful as you decide how to proceed. If you wish to submit a substantially revised manuscript, please bear in mind that we will be reluctant to approach the referees again in the absence of major revisions.

To guide the scope of the revisions, the editors discuss the referee reports in detail within the team with a view to identifying key priorities that should be addressed in revision. In this case, we think all three referees have identified important aspects of the study design and the analyses that need to be substantially improved. We particularly ask that you address their technical comments as thoroughly as possible with appropriate revisions. We hope that you will find the prioritized set of referee points to be useful when revising your study.

If you choose to revise your manuscript taking into account all reviewer and editor comments, please

highlight all changes in the manuscript text file. At this stage we will need you to upload a copy of the manuscript in MS Word .docx or similar editable format.

*2) If you have not done so already please begin to revise your manuscript so that it conforms to our Article format instructions, available here. Refer also to any guidelines provided in this letter.

Please be aware of our guidelines on digital image standards.

[redacted]

If you wish to submit a suitably revised manuscript we would hope to receive it within 6 months. If you cannot send it within this time, please let us know. We will be happy to consider your revision so long as nothing similar has been accepted for publication at Nature Genetics or published elsewhere. Should your manuscript be substantially delayed without notifying us in advance and your article is eventually published, the received date would be that of the revised, not the original, version.

Nature Genetics is committed to improving transparency in authorship. As part of our efforts in this direction, we are now requesting that all authors identified as 'corresponding author' on published papers create and link their Open Researcher and Contributor Identifier (ORCID) with their account on the Manuscript Tracking System (MTS), prior to acceptance. ORCID helps the scientific community achieve unambiguous attribution of all scholarly contributions. You can create and link your ORCID from the home page of the MTS by clicking on 'Modify my Springer Nature account'. For more

information please visit please visit www.springernature.com/orcid.

Thank you for the opportunity to review your work.

Sincerely,
Wei

Wei Li, PhD
Senior Editor
Nature Genetics
New York, NY 10004, USA
www.nature.com/ng

Reviewers' Comments:

Reviewer #1:

Remarks to the Author:

The study from Köttgen and colleagues advances our understanding of the role that rare sequence variants play in regulating plasma and urinary metabolites. With 60% of the identified gene-metabolite relationships being novel, findings from this study will boost existing knowledge. However, I do have some concerns about the examples that have been provided in the text to substantiate the idea that rare variant-based mQTLs (driven exclusively by heterozygous carriers) can capture information similar to the clinical manifestations of their unobserved homozygous counterparts.

Major Comments:

- * This study has been conducted in a CKD ascertained cohort, so it shouldn't be referred to as a 'population-based' study throughout the text.
- * The design of this study means that there might be differences in the mQTLs uncovered in this study (e.g., 'disease-specific mQTLs') compared to a general population-based setting. This point should be adequately emphasised in the Discussion.
- * Participants in this study are likely to be / have been exposed to CKD medications. This will have an effect on plasma and urinary metabolite levels that could contaminate interpretation. Is medication information available in these participants? If so, a complementary analysis should be provided that adjusts for medication use in the mQTL analysis.
- * Page 7 lines 1-4: It is mentioned that 60% of the significant gene-metabolite associations were novel compared to published sequence-based studies (refs 14-20, 24). It would be informative to know what % of these novel associations pertain to metabolites that were analysed in previous studies (but did not achieve significance) compared to what % pertain to metabolites that were uniquely analysed in this study (no equivalent/close proxy available in prior studies). For the subset of the novel gene-metabolite associations for which the metabolites were analysed previously, it would be important to see a comparison between the effect size estimates (current study vs previous study/ies) to determine if they are significantly different. This might also point toward differences driven by studying a CKD-ascertained sample on the reported mQTL effects.

Minor Comments:

* Page 4 lines 10-12: While this tends to be the usual issue with GWAS loci, it is less so for metabolite GWAS where the likely causal gene underlying a GWAS signal is often more obvious (e.g., gene encoding an enzyme / transporter relevant to the metabolite). The authors could reword this to something along the lines of "the functional effect of the associated variants is often unknown and the modest effect sizes limits their clinical impact".

* For the section 'Association of metabolite-associated alleles and genes with human traits and diseases' (page 12), can a formal enrichment analysis be done to demonstrate that the metabolite-associated genes are indeed enriched for genes associated with *clinical endpoints* in UK Biobank? There are now numerous public resources for exome-based clinical phenotype-wide studies that could enable this. Using clinical biomarkers / quantitative traits observed in the UK Biobank (as has been done in this section) might not be adequate for this purpose since these clinical biomarkers don't necessarily indicate association with relevant clinical outcomes. For e.g., page 14 lines 1-4: association of SLC7A9 and SLC6A19 with a renal biomarker (e.g., creatinine) doesn't demonstrate the point that the urinary metabolites these genes were associated with actually captures genes that are relevant to renal disease pathobiology.

* Similarly, on page 7 lines 6-8, an enrichment analysis should be performed (and corresponding p-values provided) to show whether the metabolite-associated genes are indeed enriched for drug targets and known IEM genes.

* Methods: In addition to the 'LoF_mis' and 'HI_mis' models, both of which combine LoF and missense variants, could a LoF only model be run to potentially detect gene-metabolite relationships for which effect sizes might be greater and thus enriched in this model?

* Page 10 lines 16-18: apart from the two X chromosome genes that have been highlighted, there might be scope to systematically test for sex-specific mQTL effects (i.e., interaction analysis for sex) across the exome and all tested metabolites?

* In the section 'Allelic series: metabolites represent intermediate readouts of pathophysiological processes', the highlighted example of the association between QVs in SLC13A1 / SLC26A1 with plasma sulfate and human height doesn't necessarily imply that the association of the genes with the two traits are causally related since they could represent pleiotropic effects. A conditional analysis testing the effect of these QVs on the two traits can help disentangle that.

Reviewer #2:

Remarks to the Author:

General comments: This is a very interesting manuscript written by a group with the top expertise in this area of research – I appreciate the new insights it has provided into genetic regulation of human metabolome based on WES and the examples of new computational strategies to validate the findings. It would be helpful to see (a) a little bit more emphasis and a tighter connectivity on/between metabolites regulated by rare variants and clinical/biochemical phenotypes, (ii) contemplation of the potential effect on gene/protein expression (if feasible given how rare these variants are), (iii) a little bit more information (possibly with illustrative examples) on how in silico knockout modelling and microbiome-personalised WBMS work.

Specific comments:

(1) I'd like to see a little bit more background that justifies a focus on metabolites of kynurenine pathway. Indeed, it is increasingly recognised that genetically driven changes in kynurenine pathway may lead to worse cardiovascular and renal outcomes (Pharmacol Rep, 2022;74:27-39, Kidney

International 2022;102:492-505). Can the authors comment on the health consequences of rare variants-driven changes in circulating/urinary levels of metabolites from this pathway? Did they associate with kidney function or blood pressure, for example? If yes, was their directionality and the directionality of genetic effects on metabolites consistent with the expected drop in biochemical activity of this pathway (possibly via reduced NAD⁺ synthesis) leading to either higher BP or drop in kidney function?

(2) In the same spirit, it would be nice to see more evidence linking together the genetically mediated effects on metabolites with some of the clinical outcomes at least for some of the highlighted examples, e.g. renal transporter genes SLC47A1, SLC6A19, SLC7A9, and SLC22A7.

(3) For rare and very rare alleles, a look up of their effects on gene and protein expression in relevant tissues may not be feasible - the alleles are so rare there is a good chance that they will not be present in individuals within reference panels and/or difficult to impute. I wonder whether the authors have contemplated looking up some of their most "frequent" alleles e.g to see if they can be traced there and if yes quantify the effect on the relevant gene expression?

(4) It would be most helpful to see the concept of in silico knockout modelling and microbiome-personalised WBMS using e.g. pictorial representation

(5) For rare alleles with demonstrated effect on metabolites; have the authors captured at least a few where they had rare homozygous genotype? If yes, was the effect on metabolites following the additive model of inheritance?

(6) RE: X chromosome analysis: have pseudoautosomal regions been excluded from the study

(7) Please add gene names to Tables ST3

Reviewer #3:

Remarks to the Author:

The study reports genetic associations with serum and urine metabolites using gene-based test for rare WES variants in the German Chronic Kidney Disease (GCKD) study. WES variants were selected based on predictive annotations. Metabolites were generated using non-targeted mass spectrometry analysis from Metabolon. The study identified several significant metabolite-gene associations, with a small overlap of metabolite-gene associations among serum and urine metabolites. There are several follow-up analyses including an attempt to identify variants that are drivers of the gene-based associations, and some exploration of in silico data and simulation approaches to uncover function as well as explorations in the UK Biobank for relationships with clinical traits. The authors are experienced in the field and have published similar studies using metabolites in this cohort. There are some interesting aspects of the project including the simulation of gene-knockouts and the report of one patient with a homozygous KYNU mutation. The paper reads like two different projects that have been combined in one report, instead of a sequential follow-up of results.

Major concerns.

1. The study participants have chronic kidney disease, and a low glomerular filtration rate (GFR) accounts for the largest variation in metabolite levels in both plasma and urine. Therefore, results are heavily influenced by the GFR and associations may be confounded by chronic kidney disease. Adjusting for GFR does not completely account for this. It would be important to provide results among participants with normal GFR for comparison.

2. The major limitation of the paper is the lack of replication of the gene-based results in an independent sample. This is even more important in the context of gene-based analysis focused on rare variant associations, and subsequent analyses of "variant drivers". The follow-up analyses are

based on these non-validated findings, and it is difficult to assess the validity of these additional results in the context of the initial findings.

3. Overall, a large amount of the first part of the paper is descriptive and focused on the number of associations with little detail on novel metabolite-gene associations, which are not clearly highlighted. The follow-up analyses also do not focus on novel associations and seem to change the focus on specific results such as patient mutation and in silico simulations.

4. There are several assertions throughout the paper that are beyond the reported findings, and they should be either removed or clearly stated as hypothesis. The paper should clearly state that no functional assays were performed for any of the genes/variants.

Specific questions:

The study used a burden test for gene-based analysis, and it is unclear why new methods for collapsing rare variants were not chosen.

Details on the two approaches to select variants need to be included when results are first cited, and the type of variants selected clearly stated. There is some ambiguity in the description throughout the text that makes one think that all the variants selected were predictive damaging.

It would be important to know the number of variants tested per gene for each of the two approaches used for all significant associations.

The whole paragraph under "Identification and properties of 192 significant gene-metabolite associations" is a long description of number of associations without any specific details on type of metabolites or genes, which does not provide many insights on findings. There is likely a high correlation among these metabolites, which could account for the multiple associations with the same gene, which needs to be explored.

It was unclear how the comparison of significant gene associations between this study and published one was done. Contrasting the type of metabolomic platform and the strategy used for variant selection may provide more context on the comparisons.

The focus of the paper is on inborn errors of metabolism (IEM) but the cohort includes older adults with chronic kidney disease and no known IEM.

On Page 6, the statement "The proportion of lipids was substantially higher among associated metabolites detected in plasma compared to urine, consistent with the absence of glomerular filtration of many lipids." Clarify if this means that some lipid metabolites were not available in the urine, and this explains the differences in associations for lipids.

Page 7, 2nd paragraph. The results for drug targets should focus on newly associated genes and not all the 73 identified genes, given several are already reported. Provide this information in the main text. Table S4 column "T" IEM Status, the interpretation of the listed numbers needs to be included and add a column showing which associations are new.

Page 7, differences in the associations using the two approaches for gene-based analysis. The most likely explanation is power than differences in genetic architecture, given some variants were removed in the stricter definition based on annotations. The reduced number of variants within each gene is

shown in Figure S3 and could be listed in the main text.

Page 8, 1st paragraph, statement that some genes show differences in association with a metabolite in serum and urine, is not easy to verify in supplementary material. Perhaps include side to side results in supplementary figures? Same for statements 3 and 4 in the same paragraph, which are not easily queried in results.

Include the definition of "driver variants" in the main text and clarify if there is more than one driver variant at a gene. It seems that these variants were selected based on lowest p-values. Figure S3 does not highlight the driver variants as mentioned in the text.

It seems that some driver variants were outliers in the gene burden test. It would be of interest to report their allele count, which I could not find in Tables S5a and b.

Page 9, analyses of common variants, clarify if variants were from WES or GWAS/imputed variants.

The summary sentence on page 9, 2nd paragraph, seems to be out of place and is not based on results.

Page 9, paragraph under "Heterozygous variants". There is a change in the focus from driver variants to all variants identified in all genes for further assessments (hemizygoty, in silico knockout modeling, and allelic series). This is somewhat puzzling given the large effort in identifying driver variants and showing their stronger associations with genes.

Page 10, related to sex differences in associations in X-chromosome genes, report the interaction p-value by sex.

Virtual IEM, this is interesting, but you will need to clarify why the number of genes queried was reduced to 25. The parameters for the simulation should be included as well as the rationale for testing separately in men and women. These simulations are not clear-cut validation of associations, so some rewording on statements is needed. Also, include some references on validation of these simulation approaches. The results from one patient homozygous KYNU mutation support predictions for one gene pathway but not the other genes investigated. This part of the paper seems to be a different focus from the first part and needs to be better integrated into the whole report.

The associations with health outcomes in the UK Biobank are focused on the original 73 genes, not novel associations. The UK Biobank has metabolomics and could be used for replication of gene-based results in addition to PHEWAS.

Page 13, the 2nd paragraph starting with "We have previously shown", clearly state what is newly reported and what is already published given this publication used the same dataset. Same for the description of allele series in Page 14, what is new and what is already published? The ambiguity in the statements makes one think that these are all new results.

Page 15, related to allelic series: I am not sure if Pearson correlation is the best test for testing the genetic effect sizes on plasma sulfate levels in the GCKD study and sitting and standing height in the UKB and concluding that there is a "causal effect". Perhaps use a causal model? This analysis is using driver variants, so clarify why some analysis used all variants and others just the driver variants. The last paragraph on this page is a comment on a published study and it needs to be moved to the

discussion.

Page 17, the conclusion that “these findings provide convincing evidence that lower transcellular sulfate reabsorption is associated with numerous adverse musculoskeletal traits and diseases” is an overstatement as the evidence is based on associations and not functional assays.

Page 17, discussion. The sentences related to “heterozygous variants identified in a population sample permit insights into graded effects of impaired gene function without the need to identify patients with a corresponding biallelic IEM” is an overstatement of the paper findings, given the gene-based results were not replicated and the study is using relative levels of metabolites. Therefore, except for the one KYNU patient results that used absolute metabolite quantification, inference related to disease do not apply.

I may have missed this but the “impaired epithelial transcellular sulfate reabsorption” was not tested in assays and is based on association findings with phenotypes. I suggest rephrase this statement.

Methods:

Report number of missing metabolites in serum and urine and how missing was handled.

GFR was estimated using the race-based equation, which is no longer used. The updated 2021 equation should be used instead.

Add references for the need for adjustments for serum albumin and urine albumin for serum and urine metabolite analyses.

A sensitive analysis excluding participants with low GFR should be done to confirm the main findings, as associations may be driven by chronic kidney disease.

Which threshold for significance was determined in the single rare variant analysis?

Results for conditional analysis at genes need to be included to sort out if the associations with common variants explain the associations with rare variants.

Figures:

Figure 1. change “Novel genes” to “novel gene associations”.

Figure 2a circus plot is hard to read the metabolite names and the color code is too complex. I find this type of graph unhelpful as they mask important details. Consider simplifying it.

Figure 2b, it is not very helpful to show broad categories of metabolites and you could move this to a supplemental material.

Figure 3, remove statement about the enzyme function as this was not measured in the study.

Measured metabolites are relative levels and not absolute levels so one cannot infer if they are within normal or abnormal range.

Figure 5a, this type of graph is usually shown when there is a functional experiment performed, which does not seem to be the case and can mislead the interpretation of results. I suggest that this is removed. Figure 5c, label that results are from the UK Biobank.

Figure 6a, label for NaS1 p.R272C has two het categories. Fig 6b, add the number of participants that had the NaS1 p.Arg272Cys variant and an outcome. Is this the “Number of QV carriers with disease”? it would be better to show the estimates and 95% confidence intervals and N cases/controls for these results as the display is confusing.

Supplementary material:

Table S1. Add % chronic kidney disease based on GFR and urine albumin.

Figures S1 and S2. The statement citing these figures says that most of the variants were observed in heterozygous state, which is not shown. The significant gene-metabolite associations are from collapsing all variants within the gene, but the figures include each allele, and there is no way to know if a participant has multiple alleles. Although the overall mean metabolite level between groups is different, there is a large overlap in the distribution of the metabolite, and results are likely unadjusted. These figures need to be revised.

For Figure S5, provide the comparison for the effect estimates or p-values for driver/non-driver variants by allele count as I suspect some of the associations are driven by a very low allele count.

Figure S6, include the definition of shared and unique associations. A conditional analysis likely would be a better approach to sort out this. This figure does not provide insights on results at the genomic regions including the relation between the absolute aggregated effect size of rare variants with the presence of a GWAS signal in the region, as stated in the main text.

Data availability

"Further data is available on personal request." Is not appropriate and authors should include link to the genotype, metabolite and phenotype data for the GCKD and UK Biobank.

A role of pharma on this study needs to be clarified given funding source of genotype and metabolomics. Conflicts of interest will need to be updated based on these clarifications.

Author Rebuttal to Initial comments

Reviewers' Comments:

Reviewer #1:

Remarks to the Author:

The study from Köttgen and colleagues advances our understanding of the role that rare sequence variants play in regulating plasma and urinary metabolites. With 60% of the identified gene-metabolite relationships being novel, findings from this study will boost existing knowledge. However, I do have some concerns about the examples that have been provided in the text to substantiate the idea that rare variant-based mQTLs (driven exclusively by heterozygous carriers) can capture information similar to the clinical manifestations of their unobserved homozygous counterparts.

Response: Thank you for the comments and thoughtful suggestions, which we have addressed as outlined below.

Major Comments:

1. This study has been conducted in a CKD ascertained cohort, so it shouldn't be referred to as a 'population-based' study throughout the text.

Response: We have removed “population-based” when referring to the GCKD study throughout the manuscript.

2. The design of this study means that there might be differences in the mQTLs uncovered in this study (e.g., 'disease-specific mQTLs') compared to a general population-based setting. This point should be adequately emphasised in the Discussion.

Response: We agree and have now emphasized this point in the discussion (page 21, lines 486-488, 490-494). Our previous work^{1,2}, as well as the new validation (see Reviewer 3, point 2) and kidney-function stratified analyses (see Reviewer 3, point 1) carried out as part of the requested revisions, show that genetic effect sizes on metabolite levels are comparable between our study of individuals with moderately impaired kidney function and those from several population-based studies. This suggests that metabolite levels have a stable genetic component in individuals with and without reduced kidney function.

3. Participants in this study are likely to be / have been exposed to CKD medications. This will have an effect on plasma and urinary metabolite levels that could contaminate interpretation. Is medication information available in these participants? If so, a complementary analysis should be provided that adjusts for medication use in the mQTL analysis.

Response: We performed sensitivity analyses of all significant findings by including additional covariates into the model that indicated the use of common medications prescribed in CKD (ACE inhibitors, ARBs, beta blockers, calcium channel blockers, anti-diabetic medications, diuretics, and lipid-lowering medications. SGLT2-inhibitors were not yet on the market at the time of metabolite measurements). Adjustment for the use of these medications left the associations between the aggregated variants and the relevant metabolites virtually unchanged. A formal test to detect a difference in their effect sizes with

and without adjustment for the use of common CKD medications did not yield any significant differences (all p-values >0.7, Pearson correlation between effect sizes: 0.9998; **Reviewer Figure 1**).

Reviewer Figure 1: Effect sizes of associations with metabolite levels with and without adjustment for medication.

4. Page 7 lines 1-4: It is mentioned that 60% of the significant gene-metabolite associations were novel compared to published sequence-based studies (refs 14-20, 24). It would be informative to know what % of these novel associations pertain to metabolites that were analysed in previous studies (but did not achieve significance) compared to what % pertain to metabolites that were uniquely analysed in this study (no equivalent/close proxy available in prior studies). For the subset of the novel gene-metabolite associations for which the metabolites were analysed previously, it would be important to see a

comparison between the effect size estimates (current study vs previous study/ies) to determine if they are significantly different. This might also point toward differences driven by studying a CKD-ascertained sample on the reported mQTL effects.

Response: We agree that these are relevant points, which we have now addressed by using data from four previously published studies that reported the effects of rare genetic variants on the plasma/serum metabolome using a comparable technology to ours³⁻⁶. There were no previous WES-based studies of the urine metabolome. For the 128 significant gene-metabolite relationships in the GCKD study that involved 122 plasma metabolites, these comparisons yielded the following main findings:

- For 95% of the associations (122/128), the corresponding metabolite was analyzed in at least one of the four studies.
- With respect to the same four studies, 73% (93/128) of the associations were novel and 94% (87/93) of these novel associations involved a metabolite analyzed in at least one of the four previous studies.

We have included these proportions in the manuscript on page 7, lines 157-164. Plausible explanations why so many associations were novel, although the corresponding metabolite had been analyzed previously, include the relatively small sample size and, consequently, lower power of some of these previous studies^{5,6}, differences in qualifying variants, and various differences in analytical choices (see below).

We have also performed three types of new analyses to compare effect sizes of our findings to those from previous studies as rigorously as possible: 1) For overlapping metabolites, we conducted new gene-based tests in the UKB where we could use exactly the same variants, data transformations, tests, and gene transcripts as in our study, and found highly similar effect sizes (please see response to Reviewer 3, point 2, for details). 2) We compared effect sizes for individual qualifying variants to those from Bomba *et al*³, a previous large study of the plasma metabolome (based on MS-quantification) that made these summary statistics accessible. We found an excellent correlation between genetic effect sizes on metabolite levels of informative QVs in both studies, with a Spearman correlation coefficient of 0.81. 3) We compared aggregated effect sizes of units: while the study of Bomba *et al* did not provide summary statistics for aggregated effect estimates on a gene-level, they did provide such estimates on a window-level, where each gene was divided into windows based on the exon structure and the number of QVs. We found that effect sizes of significant gene-metabolite associations detected in our study correlated strongly with those based on the window of the respective gene with the lowest P-value in Bomba *et al* (Spearman correlation coefficient of 0.82), despite differences in masks, aggregate variant tests, covariates, and the unit used for aggregation.

Thus, we conclude that our results are in excellent agreement with the available validation data. We believe this is an important new addition to our manuscript, for which we thank the Reviewer and that we have included in a new main Figure 3 and corresponding text in the Results section (page 8, lines 179-192), the Abstract and the Discussion (page 18f, lines 436-438).

Minor Comments:

5. Page 4 lines 10-12: While this tends to be the usual issue with GWAS loci, it is less so for metabolite GWAS where the likely causal gene underlying a GWAS signal is often more obvious (e.g., gene encoding an enzyme / transporter relevant to the metabolite). The authors could reword this to something along the lines of "the functional effect of the associated variants is often unknown and the modest effect sizes limits their clinical impact".

Response: We agree and have adapted the language accordingly (page 4, lines 83f).

6. For the section 'Association of metabolite-associated alleles and genes with human traits and diseases' (page 12), can a formal enrichment analysis be done to demonstrate that the metabolite-associated genes are indeed enriched for genes associated with *clinical endpoints* in UK Biobank? There are now numerous public resources for exome-based clinical phenome-wide studies that could enable this. Using clinical biomarkers / quantitative traits observed in the UK Biobank (as has been done in this section) might not be adequate for this purpose since these clinical biomarkers don't necessarily indicate association with relevant clinical outcomes. For e.g., page 14 lines 1-4: association of SLC7A9 and SLC6A19 with a renal biomarker (e.g., creatinine) doesn't demonstrate the point that the urinary metabolites these genes were associated with actually captures genes that are relevant to renal disease pathobiology.

Response: While we did not state in the manuscript that metabolite-associated genes were enriched for association with clinical traits and diseases, we now attempted to perform formal enrichment testing by leveraging data from the UK Biobank, the largest accessible dataset that should therefore have the best statistical power. Based on a systematic analysis that integrated the WES data with clinical diseases and diagnoses as described in Wang *et al*⁷ and shared through the AZ portal (<https://azphewas.com/>), 102 genes showed significant associations ($p < 1e-08$) with any binary clinical trait. However, none of these 102 genes overlapped with the 73 unique metabolite-associated genes detected in our study. Therefore, formal enrichment analyses provided no meaningful result. FinnGen, another very large European sample sharing genetic associations across the phenome (https://www.finnngen.fi/en/access_results), only provided genotyped rather than sequenced data and therefore contained only a median of two qualifying

variants per gene (as compared to 16 for the HI_mis mask in the GCKD study), which was not sufficient for further analyses. We conclude that while we detect significant associations between the identified metabolite-associated genes with selected clinical endpoints such as *APOC3* with “Disorders of lipoprotein metabolism and other lipidaemias”, formal enrichment testing could not be performed.

We agree with the Reviewer’s last statement, i.e., that continuous measures of kidney function do not equate to associations with binary kidney disease endpoints, which we have not detected after stringent correction for multiple testing. To avoid the impression that clinical endpoints are enriched for associations with metabolite-related genes, as well as in response to a request by Reviewer 3 to improve the flow of the manuscript, we have moved a short summary of these findings to the end of the Results section (page 18, lines 413-428) and now present more detailed results in the Supplementary Results (page 11f). We do believe, however, that Supplementary Tables 13 and 16 contain interesting associations with clinical endpoints of suggestive significance, and provide them as a resource to enable readers to explore such associations.

7. Similarly, on page 7 lines 6-8, an enrichment analysis should be performed (and corresponding p-values provided) to show whether the metabolite-associated genes are indeed enriched for drug targets and known IEM genes.

Response: Thank you for this helpful suggestion, which we followed. We found that significant genes detected in our study were strongly overrepresented among genes known to be causative for inborn errors of metabolism (odds ratio: 10.6, P-value = $1.9e-14$). For drug targets, we observed that such overrepresentation increased across phases of clinical trials (phase 1-4: OR 1.4, phase 2-4: OR 1.5, phase 3-4: 1.6, phase 4 only: 2.1). This was not statistically significant, although phase 4 only results approached statistical significance (P-value = 0.068). We included the new results related to IEMs in the manuscript (page 7, lines 167f) and updated the Supplementary Methods accordingly (page 4). As our study does not further evaluate any potential drugs and to focus on the main messages, we have moved information related to drug targets to Supplementary Table 5.

8. Methods: In addition to the 'LoF_mis' and 'HI_mis' models, both of which combine LoF and missense variants, could a LoF only model be run to potentially detect gene-metabolite relationships for which effect sizes might be greater and thus enriched in this model?

Response: Thank you for raising this interesting point, which we addressed analytically. We found that the absolute genetic effect sizes on metabolite levels based on a LoF only model indeed tend to be greater (Reviewer Figure 2a), with an excellent Spearman correlation of genetic effect sizes between our main models and the LoF model of 0.96. A formal test comparing effect sizes across models found that only one of the associations differed significantly between the LoF only and the main models (*DPYD* and uracil, P-value= 0.00018<0.05/#available hits). Of note, 57 associations significant with our main models could not be analyzed with the LoF only model, because no qualifying variants were detected. These 57 included well-established positive controls. Lastly, for all except for three associations, the association P-value provided by the LoF only model was higher, i.e., less significant, compared to those provided by the two main models (Reviewer Figure 2b). Hence, most of the significant associations would not have been detected based on the LoF only model.

We included a summary of these findings in the Supplementary Results (page 5), along with **Reviewer Figure 2** as a new Supplementary Figure 2, and added the summary statistics based on the LoF only model to a new Supplementary Table 4.

Reviewer Figure 2: Comparison between LoF_mis/Hi_mis masks and a LoF only mask in the GCKD study with regard to effect sizes (a) and $-\log_{10}(\text{P-values})$ (b) of gene-metabolite associations.

9. Page 10 lines 16-18: apart from the two X chromosome genes that have been highlighted, there might be scope to systematically test for sex-specific mQTL effects (i.e., interaction analysis for sex) across the exome and all tested metabolites?

Response: We refrained from performing exome-wide interaction analysis for two reasons: first, the statistical power to detect significant associations is reduced in this setting, which is important in light of the need to stringently control for multiple testing. Secondly, many of the QVs are only detected in one or two carriers. Stratification would result in partly different QVs within a gene in men and women, so that detected differences could not confidently be attributed to differences in sex.

To still address the Reviewer's valid point, we have now performed sex-stratified analyses for all significant gene-metabolite associations and formally tested for interaction by sex. There were only three

associations that showed significant differences ($p < 0.05/128 = 3.9 \times 10^{-4}$ for plasma and $p < 0.05/107 = 4.7 \times 10^{-4}$ for urine based on the number of tested associations) in effect sizes between men and women. All of these were associations between metabolites and loss-of-function variants in the X-chromosomal gene *TMLHE* gene (plasma and urine hydroxy-N6,N6,N6-trimethyllysine, plasma deoxycarnitine). For all three associations, men showed significantly larger absolute effect sizes compared to women, which can be explained by their hemizygous genotypes, effectively corresponding to homozygosity in women (see page 11f, lines 265-271, and corresponding discussion). Sex-stratified results for the *TMLHE* gene were already included in Supplementary Table 8. We have now added the comparison of all gene-metabolite associations in men and women to the manuscript as a new Supplementary Figure 4b and describe these findings in the Supplementary Results (page 6).

10. In the section 'Allelic series: metabolites represent intermediate readouts of pathophysiological processes', the highlighted example of the association between QVs in *SLC13A1* / *SLC26A1* with plasma sulfate and human height doesn't necessarily imply that the association of the genes with the two traits are causally related since they could represent pleiotropic effects. A conditional analysis testing the effect of these QVs on the two traits can help disentangle that.

Response: We agree that pleiotropic effects cannot be excluded, although putative loss-of-function variants in genes encoding sulfate transport proteins should most immediately affect sulfate levels more than other traits that could potentially introduce pleiotropy. Unfortunately, conditional analyses in the well-powered UKB study, in which the association with height was assessed, is not possible because plasma sulfate measurements are not available, and statistical power in the GCKD study is limited because of the much lower number of variant carriers. We therefore believe that the growth defect / lower size described in sulfate transporter KO mice^{8,9} is more compelling evidence of a causal relationship between a loss-of-function in the genes and height. We have now included a statement that once sulfate becomes available in the UKB, conditional analyses should be carried out (page 21, lines 484f).

Reviewer #2:

Remarks to the Author:

General comments: This is a very interesting manuscript written by a group with the top expertise in this area of research – I appreciate the new insights it has provided into genetic regulation of human metabolome based on WES and the examples of new computational strategies to validate the findings. It would be helpful to see (a) a little bit more emphasis and a tighter connectivity on/between metabolites regulated by rare variants and clinical/biochemical phenotypes, (ii) contemplation of the potential effect

on gene/protein expression (if feasible given how rare these variants are), (iii) a little bit more information (possibly with illustrative examples) on how in silico knockout modelling and microbiome-personalised WBMS work.

Response: We appreciate the positive feedback and thank the Reviewer for the constructive suggestions, which we have addressed as outlined below.

Specific comments:

(1) I'd like to see a little bit more background that justifies a focus on metabolites of kynurenine pathway. Indeed, it is increasingly recognised that genetically driven changes in kynurenine pathway may lead to worse cardiovascular and renal outcomes (Pharmacol Rep, 2022;74:27-39, Kidney International 2022;102:492-505). Can the authors comment on the health consequences of rare variants-driven changes in circulating/urinary levels of metabolites from this pathway? Did they associate with kidney function or blood pressure, for example? If yes, was their directionality and the directionality of genetic effects on metabolites consistent with the expected drop in biochemical activity of this pathway (possibly via reduced NAD⁺ synthesis) leading to either higher BP or drop in kidney function?

Response: Our work highlights examples that shed light on a common theme from different angles, namely that the aggregated effects of heterozygous variants permit complementary insights into mechanisms that are classically studied through full loss-of-function models (e.g., autosomal-recessive IEMs, knockout animals, as well as our newly developed computational models to simulate such a scenario). We selected the *KYNU* gene as one such example, because it nicely illustrates the complementary and consistent information drawn from these different approaches, including the extension to the corresponding IEM as shown through data from a patient with kynureninase deficiency in our care. To clarify that we did not choose *KYNU* because of the importance of the kynurenine pathway, and to show that the converging lines of evidence hold true not only for the *KYNU* gene, we have now moved the previous Figure 4a (that contained many metabolites of the kynurenine pathway) to a new Supplementary Figure 11a, and instead added new population-, modeling-, and IEM patient-evidence for an additional gene, *PAH*, to Figure 6 (previously Figure 4). New Supplementary Figure 11 now contains, in addition to the effect size comparisons of *KYNU*-related metabolites from whole-body modeling to the ones observed in the GCKD study, an analogous such panel for *PAH*-related metabolites. Significant correlations between modeled and observed effect sizes are observed for both genes. Lastly, we have clarified the motivation for selecting *KYNU* and *PAH* as an example (page 13, lines 308-310).

Although *KYNU* was not chosen because of the importance of kynurenine pathway metabolites for cardiovascular or renal outcomes, we now performed additional analyses to address the Reviewer's question. We systematically explored whether rare, putative LoF variants in the *KYNU* gene, associated with kynurenine pathway metabolite levels in our study, were related to kidney function or blood pressure in the UKB, the largest study with the best statistical power to detect such associations⁷. For binary traits, we focused on diseases of the circulatory system (Chapter IX), of the genitourinary system (Chapter XIV), and on symptoms signs and abnormal clinical and laboratory findings not elsewhere classified (Chapter XVIII). For quantitative traits, we included traits from Chapter XVIII, factors influencing health status and contact with health services (Chapter XXI), NMR Metabolomics and Olink proteomics. Both on the variant- and gene-level (after excluding the synonymous mask), no significant association defined as in the original publication⁷ were identified, except for associations in *cis* with the protein encoded by the *KYNU* gene itself. Since the absence of such associations is already reflected in Supplementary Tables 13 and 16, we did not make changes to the manuscript. We do however agree with the Reviewer that there is growing evidence that genetically driven changes in the kynurenine pathway may lead to worse cardiovascular and renal outcomes, as underscored by studies of common variants^{10,11}, even if our specific investigations of the aggregated effects of rare, putative loss-of-function variants in the heterozygous state among adult participants of population-based studies did not translate into multiple-testing corrected significant changes in blood pressure or kidney function.

(2) In the same spirit, it would be nice to see more evidence linking together the genetically mediated effects on metabolites with some of the clinical outcomes at least for some of the highlighted examples, e.g. renal transporter genes *SLC47A1*, *SLC6A19*, *SLC7A9*, and *SLC22A7*.

Response: We agree with the Reviewer about the value of linking the genetic evidence to clinical outcomes. We have highlighted a few examples, as mentioned above, in the manuscript to emphasize the point that the matrix-specific metabolome (either urine or plasma) is a sensitive readout of the loss-of-function of the encoded proteins depending on their tissue and cellular localization. In response to a comment by Reviewer 3 to improve the flow of the manuscript, we have now moved a summary of the findings in this paragraph to the end of the main results section (page 18, lines 413-428), and clarified that the matrix-specific metabolomic fingerprints can be reflective of the functions of adjacent organs, as also shown by association with respective measures of organ function (even in the absence of associations with clinical endpoints).

There were no significant associations with binary kidney diseases after correcting for multiple testing for *SLC47A1*, *SLC6A19*, *SLC7A9*, and *SLC22A7* (statement now included on page 12 of the Supplementary Results). Nevertheless, we note that the comprehensive material provided in the corresponding Supplementary Tables (13 and 16) contains suggestive associations between a QV in *SLC47A1* and chronic renal failure. Studies that test the aggregate effect of rare and ultra-rare variants

such as ours have limited statistical power to detect multiple-testing corrected associations or perform causal inference studies with binary disease outcomes. However, the publication of genes containing rare-metabolite associated variants in our study will now enable the scientific community to test for associations with specific diseases in a hypothesis-driven manner, without the need to account for multiple testing as stringently, making the Supplementary Tables a convenient resource.

(3) For rare and very rare alleles, a look up of their effects on gene and protein expression in relevant tissues may not be feasible - the alleles are so rare there is a good chance that they will not be present in individuals within reference panels and/or difficult to impute. I wonder whether the authors have contemplated looking up some of their most "frequent" alleles e.g to see if they can be traced there and if yes quantify the effect on the relevant gene expression?

Response: Thank you very much for this excellent suggestion. We have now performed the suggested analyses. It turned out that the gene expression analyses were not feasible because of the very small sample size of most tissues in GTEx. Hence, there were not enough carriers of QVs. The analysis of protein levels, however, was informative and yielded the following main findings:

- Of 73 significant genes detected in our study, 17 of them had their plasma protein levels measured as part of the Olink plasma proteomics available in the UKB¹².
- 15 of these 17 genes were associated with the levels of their encoded proteins in *cis* with an association P-value <1e-5.
- For these *cis* associations, the pvraredmg mask (containing protein truncating variants and rare damaging missense variants similar to our masks) provided the lowest association P-value among masks tested in¹², and the effect direction was negative in all cases (consistent with loss of function).

We believe that this is strong complementary evidence that putative LoF variants in these metabolite-associated genes are truly functional, because QVs in these genes selected by a similar approach lead to lower levels of their encoded proteins, consistent with a LoF mechanism. We have included these findings in the new Figure 3, panel b. We have added descriptions to the Results (page 8, lines 184-189) and Supplementary Methods (page 5).

(4) It would be most helpful to see the concept of in silico knockout modelling and microbiome-personalised WBMS using e.g. pictorial representation

Response: We agree with the Reviewer and have now generated such a figure, added as new Supplementary Figure 10, highlighting the general workflow and depicting the underlying mathematical concepts.

(5) For rare alleles with demonstrated effect on metabolites; have the authors captured at least a few where they had rare homozygous genotype? If yes, was the effect on metabolites following the additive model of inheritance?

Response: Thank you for raising this relevant point. We have indeed three instances of autosomal genes where more than one individual was homozygous for a metabolite-associated qualifying variant: *ENOSF1* and urine ribonate, *SLC10A2* and urine glycocholate, and *SLC47A1* and urine acetylspermidine. In all three instances, a clear additive trend is observed. We have added a new Supplementary Figure 9 for visualization, and a short statement in the manuscript (page 11, lines 251-254).

(6) RE: X chromosome analysis: have pseudoautosomal regions been excluded from the study

Response: Yes, the results from genes in the PAR were not included in any interpretation, even if genes in the PAR were sequenced.

(7) Please add gene names to Tables ST3

Response: Gene names were already present in column B.

Reviewer #3:

Remarks to the Author:

The study reports genetic associations with serum and urine metabolites using gene-based test for rare WES variants in the German Chronic Kidney Disease (GCKD) study. WES variants were selected based on predictive annotations. Metabolites were generated using non-targeted mass spectrometry analysis from Metabolon. The study identified several significant metabolite-gene associations, with a small overlap of metabolite-gene associations among serum and urine metabolites. There are several follow-up analyses

including an attempt to identify variants that are drivers of the gene-based associations, and some exploration of in silico data and simulation approaches to uncover function as well as explorations in the UK Biobank for relationships with clinical traits. The authors are experienced in the field and have published similar studies using metabolites in this cohort. There are some interesting aspects of the project including the simulation of gene-knockouts and the report of one patient with a homozygous KYNU mutation. The paper reads like two different projects that have been combined in one report, instead of a sequential follow-up of results.

Response: We thank the Reviewer for the thorough review of our work and the many helpful suggestions for how to improve it. We have addressed all points as detailed below.

Major concerns.

1. The study participants have chronic kidney disease, and a low glomerular filtration rate (GFR) accounts for the largest variation in metabolite levels in both plasma and urine. Therefore, results are heavily influenced by the GFR and associations may be confounded by chronic kidney disease. Adjusting for GFR does not completely account for this. It would be important to provide results among participants with normal GFR for comparison.

Response: We agree that this point deserves attention. In our previous studies^{1,2} of associations between common genetic variants and metabolite levels, we had systematically compared genetic effect sizes on metabolite levels from the GCKD study to those estimated from participants without reduced eGFR in two different studies. Our comparisons for urine metabolites¹ and plasma metabolites² both showed that genetic effect sizes were highly similar: for example, the Pearson correlation coefficient of genetic effects on plasma metabolite levels from the GCKD study to those from participants of the ARIC Study with an eGFR >60 ml/min/1.73m² was 0.98².

We have now additionally performed two new analyses: first, we performed a sensitivity analysis in which we stratified study participants by eGFR ≤ 45 ml/min/1.73m² (N=2,185) and eGFR >45 ml/min/1.73m² (N=2,528). This eGFR cutoff yielded groups of similar sample size, which should maximize statistical power to detect any differences. We found that the genetic effect sizes on metabolites were very similar and highly correlated (Pearson correlation 0.97) across the two groups. Moreover, none of the gene-metabolite pairs showed significantly different effect sizes across groups. We have included these results in the Supplementary Results (page 6) and as a new Suppl. Figure 4a. Analyses stratifying at an eGFR of 60 ml/min/1.73m² (N=3,734 and N=979) yielded very similar results.

Secondly, prompted by requests for independent validation (see next comment) or effect size comparisons to previous studies (see Reviewer 1, point 4), we have now compared the effect sizes of rare variant associations with plasma metabolite levels from our study to those from the only previous study that used a comparable approach to metabolite quantification and shared their exome-wide summary statistics³. A detailed description of our findings is provided in response to Reviewer 1, point 4. In short, we found that effect sizes for variants present and informative in both studies were highly correlated (Spearman correlation coefficient of 0.81), despite differences in covariate selection and participant characteristics. Additionally, effect sizes on the gene-level were also highly correlated (Spearman correlation coefficient of 0.82), despite differences in burden tests, masks, and the definition of windows for aggregating QVs. Taken together, there is no indication that genetic effects on metabolite levels differ between individuals with and without reduced kidney function, neither for common nor rare variants.

2. The major limitation of the paper is the lack of replication of the gene-based results in an independent sample. This is even more important in the context of gene-based analysis focused on rare variant associations, and subsequent analyses of “variant drivers”. The follow-up analyses are based on these non-validated findings, and it is difficult to assess the validity of these additional results in the context of the initial findings.

Response: Replication of gene-based rare variant associations is challenging due to the fact that many qualifying variants are very rare and even private, so that an independent sample will have many different QVs, even the same criteria for selecting QVs are applied. Furthermore, large samples with urine metabolomics that could be used for replication are not available so far. In addition to the validation of our findings in whole body models of human metabolism, our findings are of high biological plausibility: many significant metabolites are known substrates/products of enzymes and transporters encoded by the associated gene. In addition, a large proportion of genes detected in our screen are known to harbor mutations causing IEMs that involve the same metabolites implicated in our screen. In fact, many QVs and especially driver variants detected in our study in the heterozygous state are known to cause IEMs in the homozygous state. Lastly, the follow-up analysis of the six mutations in the sulfate transporters SLC13A1 and SLC26A1 were based on experimentally proven loss-of-function and sulfate and disease-associated stop gain variants^{13–17}.

In response to the Reviewer’s point, we have additionally performed four new analyses to validate our findings (please also compare response to Reviewer 1, point 4, for details):

- 1) We performed gene-based tests for significant metabolites available in the UK Biobank, where we could align the statistical analysis with our study with regard to transformation of metabolite levels, selection of the same QVs, and the applied burden tests. These tests were feasible for histidine and phenylalanine, as the overlap with the Nightingale platform used in the UKB was very limited. These aligned burden tests showed very similar effect sizes for both associations, despite the differences in sample size and metabolic platforms. The results are displayed in panel a) of the new main Figure 3.
- 2) We compared effect sizes at the single QV level and at the gene-level to those from a large study of the plasma metabolome quantified with the Metabolon platform³ that greatly increased overlap in analyzed metabolites. As outlined in detail in our response to Reviewer 1, point 4, we observed excellent correlations between associations in the GCKD and the INTERVAL studies (Spearman correlation coefficient 0.81 at the variant level and 0.82 at the gene level), see panel c) and d) of the new main Figure 3.
- 3) We investigated the metabolite-associated QVs from our study using the plasma proteomics data from the UKB, and found that when aggregated at the gene level, they were significantly associated with lower plasma levels of the encoded proteins, supporting loss of function as a mechanism (for details please see response to Reviewer 2, point 3 and panel b) of the new main Figure 3.
- 4) We added new experimental data to the manuscript, showing that methionine sulfone is a new substrate of the SLC6A19 transporter, with which it was associated in our study in addition to several known substrates. While this can only serve as proof-of-principle for one finding, it highlights the potential for experimental follow-up studies for validation (panel a) and b) of the new main Figure 4).

Jointly, we believe these complementary lines of evidence strongly support the validity of our findings. We have included text related to these new findings in the Results section (page 8f, lines 178-204), Abstract, Discussion (page 18f, lines 436-440), and Methods (page 37f, lines 817-825; page 41f, lines 903-920; page 49f, lines 1087-1113; Supplementary Methods page 5).

3. Overall, a large amount of the first part of the paper is descriptive and focused on the number of associations with little detail on novel metabolite-gene associations, which are not clearly highlighted. The follow-up analyses also do not focus on novel associations and seem to change the focus on specific results such as patient mutation and in silico simulations.

Response: In response to this comment and to a comment by Reviewer 1, we have now quantified and added the proportions of novel associations (73%), and how many of these (94%) involved a metabolite analyzed in at least one of the evaluated previously published studies (for details, please compare response to Rev. 1, point 4). Novelty status is also marked in ST3 (column Y) and ST5 (column AC), as well as in Figure 2a, where gene-metabolite associations not previously reported in WES-based metabolite association studies are shown in bold black font. We have rearranged text in the first part of the manuscript to clearly highlight and quantify novel associations (page 7, lines 157-164). Moreover, we have

now added experimental validation for one new gene-metabolite association as proof-of-principle (see previous point).

Many of the significant gene-metabolite relationships are reported here for the first time from human sequencing-based association studies of the metabolome. Although they are novel in that sense, many of their connections are supported by decades of biochemical research in model systems. In the second part of the manuscript, we have therefore decided to emphasize a common feature of the detected relationships rather than highlighting individual findings: when looking at the results as a whole, the degree of information contained in heterozygote genotypes in a population not selected for the study of metabolic diseases emerged as a common theme. We used and now extended several showcases to substantiate that the study of heterozygous genotypes permits inferences about the relationship to a metabolite that can be detected from the study of homozygous individuals (with larger effect sizes). A new supplementary figure (Suppl. Fig. 9) that compares individuals heterozygous for QVs in a given gene with the few homozygous individuals in our study (please see point 5, Rev. 2, for details) further supports this point. Moreover, the study of heterozygous individuals has two important advantages: first, carriers are more frequent than individuals with IEMs, thereby enabling inferences about IEMs that are ultra-rare and for which patients have not been comprehensively characterized with respect to their metabolic profiles. Second, the presence of many heterozygous QVs in the same gene facilitates the construction of allelic series, which can promote drug development efforts by providing information about a range of target inhibition and an accompanying metabolic readout. We have now clarified these points in the Discussion section.

4. There are several assertions throughout the paper that are beyond the reported findings, and they should be either removed or clearly stated as hypothesis. The paper should clearly state that no functional assays were performed for any of the genes/variants.

Response: We have revised the language throughout the manuscript accordingly, as detailed in response to the respective specific comments below. During the course of the revisions, we have also added functional validation of one finding as proof-of-principle. A description of the functional validation of SLC6A19 as a methionine sulfone transporter has been added to the Results (page 8f, lines 193-204), panels a) and b) of new Figure 4, as well as in the Methods.

Specific questions:

1. The study used a burden test for gene-based analysis, and it is unclear why new methods for collapsing rare variants were not chosen.

Response: We decided to use a burden test for aggregating rare, putatively damaging variants within a gene because of the assumed loss-of-function as the mechanism underlying metabolic changes. In this scenario, burden tests have the best power to detect associations of aggregated variants pointing in the same direction, and deliver effect sizes, which facilitates interpretation of results¹⁸. Previous studies focusing on effects of rare variants on metabolite levels have shown that burden tests provide more associations than SKAT tests^{3,19}.

With regard to the selection of qualifying variants, our rationale for evaluating two complementary masks rather than many similar masks was to constrain the multiple testing penalty in our moderately sized sample while still allowing for detection of different genetic architectures. Moreover, rare variant association studies with metabolomics²⁰, clinical traits/phenotypes⁷, or proteomics¹² in the UKB, which used 9 to 11 different non-synonymous masks for collapsing analyses, have shown that most associations were detected based on masks including protein truncating variants and their combination with rare putatively damaging missense variants. These masks are very similar to the ones used in our study.

To empirically address the Reviewer's point of using alternative approaches of gene-based testing, we performed both SKAT and a SKAT-O test for all significant gene-metabolite associations, as well as a burden test using a LoF mask including only high confidence loss-of-function variants. We included the results from these sensitivity analyses in the new Supplementary Table 4 and describe them in the Supplementary Results (page 5f). The P-value provided by the burden test outperforms the one provided by the SKAT test for 369 of 382 associations (see the new Supplementary Figure 3). That was also the case when comparing our masks LoF_mis and HI_mis with the LoF only mask that resulted in lower power due to the low number of QVs. For more details on the results based on the LoF mask, please see Reviewer 1, point 8.

2. Details on the two approaches to select variants need to be included when results are first cited, and the type of variants selected clearly stated. There is some ambiguity in the description throughout the text that makes one think that all the variants selected were predictive damaging.

Response: We agree that it is helpful for readers to include the definitions of the masks used in the Results section rather than referring to them solely in the Methods section, and expanded the text accordingly (page 6, lines 132-137).

3. It would be important to know the number of variants tested per gene for each of the two approaches used for all significant associations.

Response: This information was already included in Supplementary Table 3 in column M called “N variants used”, which allows for comparison of the two masks used when looking at the same gene-metabolite association.

4. The whole paragraph under “Identification and properties of 192 significant gene-metabolite associations” is a long description of number of associations without any specific details on type of metabolites or genes, which does not provide many insights on findings. There is likely a high correlation among these metabolites, which could account for the multiple associations with the same gene, which needs to be explored.

Response: We have revised this section by including a comparison of our results to those from previously published studies, which provides better context for the reported associations, including information on the gene and metabolite level. For details about these findings and corresponding changes, please see response to point 2 above.

Moreover, we have addressed this comment by generating a correlation matrix of all significantly associated metabolites. **Reviewer Figure 3** shows that the majority of metabolites is not highly correlated. We would also like to note that some of the genes encode for enzymes for which several metabolites represent bona fide substrates, such as *ACADM* and medium chain fatty acids, in which case metabolites will be correlated but also reflect true causal associations. In fact, metabolites that share genetic architecture will be correlated, and a correlated metabolite that is more distantly related to a reaction would still be a true causal finding.

Reviewer Figure 3: Pairwise Spearman correlation between all significant plasma or urine metabolites detected in association with the gene-based aggregated effect of rare QVs in our study.

5. It was unclear how the comparison of significant gene associations between this study and published one was done. Contrasting the type of metabolomic platform and the strategy used for variant selection may provide more context on the comparisons.

Response: We agree with the Reviewer that this is important information, and have clarified in the Methods section how the comparison of significant gene-metabolite associations between this study and published ones was done (page 37f, lines 800-825). Details on metabolomics platform, cohort, statistical tests, masks, aggregation units, transformation, covariates, and significance thresholds used in the published studies are now given in the footnote of Supplementary Table 5. The comparisons in this table have now been substantially expanded in response to this Reviewer's point 2, above, as well as to Reviewer 1, point 4.

6. The focus of the paper is on inborn errors of metabolism (IEM) but the cohort includes older adults with chronic kidney disease and no known IEM.

Response: An important message of our manuscript is indeed that we can learn something about IEMs based on the association of metabolites with the aggregated effect of rare, heterozygous variants in a population that was not ascertained for recessively inherited IEMs. We substantiate this message through multiple lines of evidence (comparison of heterozygosity and hemizyosity for X-chromosomal genes, agreement with *in silico* gene knockout modeling, comparison to an IEM patient), and have now further strengthened this point by adding an additional example for a second IEM, phenylketonuria, as detailed in response to point 17 below. Together, our analyses show that links between genes and metabolites identified in our study of mostly heterozygous individuals do not only agree with inferences about such links from homozygous genotypes / full loss of function, but also that different rare, damaging heterozygous variants in the same gene can reveal allelic series. These messages are summarized in the Discussion (e.g., page 19f, lines 441-485).

7. On Page 6, the statement "The proportion of lipids was substantially higher among associated metabolites detected in plasma compared to urine, consistent with the absence of glomerular filtration of many lipids." Clarify if this means that some lipid metabolites were not available in the urine, and this explains the differences in associations for lipids.

Response: The number of lipids quantified only in plasma (N=329) is larger compared to the number of lipids quantified only in urine (N=48) or in both matrices (N=122). Therefore, many of the associations with lipids are due to the unavailability of the implicated metabolite in urine. However, even among lipids quantified in both plasma and urine, there were both plasma- and urine-specific associations. Prompted by the Reviewer's comment, we have generated a visual comparison of quantified and significantly associated metabolites by matrix across all pathways. We believe that it contains interesting information,

as it highlights the complementary information both matrices provide and how this differs across pathways (included as Reviewer Figure 4 below and as new Supplementary Figure 1, mentioned on page 6f, lines 148f).

Reviewer Figure 4: Number of significantly associated metabolites among metabolites quantified in plasma or urine only, as well as in both matrices.

8. Page 7, 2nd paragraph. The results for drug targets should focus on newly associated genes and not all the 73 identified genes, given several are already reported. Provide this information in the main text.

Response: Among the 55 unique genes, for which associations with a given metabolite have not been reported previously, five are targets of approved or currently developed drugs. Among the 32 genes that were not reported to be associated with any metabolite in any of the previous studies, two are drug targets. We have included this number in the manuscript (page 7, lines 164f).

9. Table S4 column “T” IEM Status, the interpretation of the listed numbers needs to be included and add a column showing which associations are new.

Response: We included the interpretation of the IEM evidence status in the footnote of Supplementary Table 5 (previously ST4). The columns AB and AC of ST5 already name the studies that previously reported the gene or the gene-metabolite association. Hence, blank entries indicate that the gene or the gene-metabolite association have not been reported in any of the seven previous plasma studies and one previous urine study.

10. Page 7, differences in the associations using the two approaches for gene-based analysis. The most likely explanation is power than differences in genetic architecture, given some variants were removed in the stricter definition based on annotations. The reduced number of variants within each gene is shown in Figure S3 and could be listed in the main text.

Response: We agree that differences in associations for several gene-metabolite pairs between the LoF_mis and HI_mis masks can be explained by better statistical power of the HI_mis mask that includes more putatively damaging variants and adapted the respective statement accordingly (page 9, line 211). However, there were also gene-metabolite associations, for which the stricter LoF_mis mask provided lower P-values compared to those based on the HI_mis mask: 16 of 235 associations were detected only by the LoF_mis mask, and 74 had a lower P-value using the LoF_mis as compared to the HI_mis mask. These observations likely reflect differences in genetic architecture, where the addition of variants in the HI_mis mask seems to add noise, resulting in increased association P-values. We have added the median of the number of variants per gene aggregated in each mask to the manuscript (page 6, lines 132-137); the number of variants aggregated in each mask was already shown in the Supplementary Table 3, column M.

11. Page 8, 1st paragraph, statement that some genes show differences in association with a metabolite in serum and urine, is not easy to verify in supplementary material. Perhaps include side to side results in supplementary figures? Same for statements 3 and 4 in the same paragraph, which are not easily queried in results.

Response: As suggested by the Reviewer, we improved the queries of the results with regard to the comparison between both matrices by combining both matrices into one figure and by reordering. The new Supplementary Figure 5 contains, for each significant gene-metabolite pair, the box plot of the association in plasma on the left and in urine on the right, if the pair was significant in both matrices.

Similarly, the new Supplementary Figure 6 shows, for each significant gene-metabolite pair, the contribution of individual QVs for both masks in plasma (top) and urine (bottom).

12. Include the definition of “driver variants” in the main text and clarify if there is more than one driver variant at a gene. It seems that these variants were selected based on lowest p-values. Figure S3 does not highlight the driver variants as mentioned in the text. It seems that some driver variants were outliers in the gene burden test. It would be of interest to report their allele count, which I could not find in Tables S5a and b.

Response: A definition of “driver variants” was included in both the main text and the Methods. We additionally clarified that each gene-metabolite association contains several driver variants (page 10, lines 222f). Driver variants were marked in Supplementary Figures 1 and 2 (now Supplementary Figure 5) in blue. In Supplementary Figures 3 and 4 (now Supplementary Figure 6) driver variants can easily be identified as the variants sorted first, which, when aggregated, provide the lowest possible P-value. We now clarified this in the figure legend. In Supplementary Table 3, the number of driver variants for each gene-metabolite association is shown in column Q. Furthermore, Supplementary Tables 6a and b indicate driver variants by their rank in the set of drivers for each gene-metabolite pair (column Q). Blank entries indicate non-driver variants.

The minor allele count (MAC) for all QVs including the driver variants with $MAC > 2$ is now added to the Supplementary Tables 6a and b in column P.

13. Page 9, analyses of common variants, clarify if variants were from WES or GWAS/imputed variants.

Response: Since most variants identified by GWAS are common intronic or intergenic variants, they are not present in WES data. We therefore used array-based variants that were imputed using state-of-the-art haplotype reference panels (TOPmed) and with good imputation quality (median 0.997, IQR 0.989-0.999), which we have clarified in the Methods section (page 38, lines 829-832).

14. The summary sentence on page 9, 2nd paragraph, seems to be out of place and is not based on results.

Response: We agree with the Reviewer and have removed this sentence.

15. Page 9, paragraph under “Heterozygous variants”. There is a change in the focus from driver variants to all variants identified in all genes for further assessments (hemizyosity, *in silico* knockout modeling, and allelic series). This is somewhat puzzling given the large effort in identifying driver variants and showing their stronger associations with genes.

Response: The *in silico* modeling is performed on the gene level rather than the variant level, because the purpose of these analyses is to compare the empirical data from gene-based testing of our discovery screen to the *in silico* knockout modeling. For the allelic series, all evaluated variants were driver variants. The analyses related to hemizyosity now separately show results for driver and non-driver variants, and the newly added association P-values are provided for driver variants. We have clarified this in the figure legends.

16. Page 10, related to sex differences in associations in X-chromosome genes, report the interaction p-value by sex.

Response: We have added the sex-specific association P-values based on the driver variants to Figure 5 (previously Figure 3), and included the P-value for testing sex differences in X-chromosomal genes to Supplementary Table 8.

17. Virtual IEM, this is interesting, but you will need to clarify why the number of genes queried was reduced to 25. The parameters for the simulation should be included as well as the rationale for testing separately in men and women. These simulations are not clear-cut validation of associations, so some rewording on statements is needed. Also, include some references on validation of these simulation approaches. The results from one patient homozygous KYNU mutation support predictions for one gene pathway but not the other genes investigated. This part of the paper seems to be a different focus from the first part and needs to be better integrated into the whole report.

Response: We agree that more details on this part are helpful and revised the manuscript accordingly. The virtual IEMs are based on the organ-resolved sex-specific whole body models published in Thiele et al. 2020²¹. As such, we can only model genes that encode for enzymes and transporters with gene-protein-reaction annotations in the human genome scale reconstruction RECON 3D²². However, only genes that were 1) exclusively causal (see page 42, lines 933-937 for formal definition), 2) for which the implied metabolites had excretion reactions into urine, and 3) for which annotated reactions could carry flux can be meaningfully modeled with the current pipeline. Thus, 24 genes before curation and 26 after curation

could be modeled with the updated version of the WBM, which we have now clarified in the description of the method (page 42, lines 923-938 and page 44, lines 976-982). Please note that simulations were updated in parts, as a new version of the underlying whole-body model was made available during the revision, which led to changes in mapping and changes in the QP modelling. However, the main findings remain unchanged.

The “parameters” of the simulation consist of the stoichiometric matrix, the constraint settings, and the parameters for the optimization procedures. The latter were already included in the manuscript (page 43, line 943-949), and we clarified the wording for better understanding. The former are available at

https://www.thielelab.eu/files/archives/f63236_3b89d6f5077848769fc3039d37cd6858.zip?dn=Harvey_1_03c.mat.zip and https://www.thielelab.eu/files/archives/f63236_323a1c166b97415aa5ede5f47551de04.zip?dn=Harvetta_1_03c.mat.zip.

All code for the performed simulation will be made publicly accessible via Github (https://github.com/SysPsyHertel/CodeBase/tree/main/Scripts_Scherer_WBM_QP) upon publication of the paper.

The rationale for separate *in silico* testing in men and women is that the models are sex-specific organ-resolved reconstructions. Thus, the female whole body model contains a different set of organs than the male model. In consequence, the stoichiometric matrix of the male and the female models are different, and thus we have two mechanistically distinct models.

We also carefully reviewed the wording. Indeed, in the sense that “*in silico* validation” is used in the setting of genome-wide population screens, it may be misleading. We therefore now refer to the *in silico* modeling results as “additional supporting evidence” and similar.

As requested, we now included some additional references on the validation of the simulation approaches (page 45, lines 991-993 and page 5, line 108). Although the use of this method is novel in our setting, constraint-based reconstruction and analysis as such is a well-proven computational approach with a very broad range of applications in biotechnological research and biotechnological industry.

Lastly, we now added another example for which we combine evidence from gene-based aggregate variant testing, *in silico* knockout modeling, and evidence from a patient with the corresponding recessively inherited IEM: using information from a patient with phenylketonuria caused by a homozygous loss-of-function mutation in *PAH* in our care, we show that the pipeline works well not only for the *KYNU* gene, supporting that the methodology is in principle generalizable. We included these new results on page 13f, lines 308-313 and 325-330 and as Figure 6b and Supplementary Figure 11b, and also improved the general flow of the results part.

18. The associations with health outcomes in the UK Biobank are focused on the original 73 genes, not novel associations. The UK Biobank has metabolomics and could be used for replication of gene-based results in addition to PHEWAS.

Response: We have performed investigations of all genes in the UKB, to permit readers both to focus on novel associations but also to assess whether there are novel phenotype associations for genes previously related to metabolites. Thank you for the good suggestion to validate our findings in the UKB metabolomics data. The overlap between metabolites was very limited, because the Nightingale platform, used in the UKB, focuses on lipids. However, for the available overlapping metabolites, we have now performed gene-based tests using exactly the same workflow as in our study, and found that effect sizes closely align with the ones identified in our study. We have added these new results in our new Figure 3, panel a. This figure summarizes validation of our findings based on independent, complementary sources of evidence. Please compare to our response to Reviewer 2, comment 3, for a detailed description of the figure and the corresponding additions to the manuscript in the Results, Methods and Discussion section.

19. Page 13, the 2nd paragraph starting with “We have previously shown”, clearly state what is newly reported and what is already published given this publication used the same dataset. Same for the description of allele series in Page 14, what is new and what is already published? The ambiguity in the statements makes one think that these are all new results.

Response: Regarding the first statement, we have now clarified that the cited previous study referred to an investigation of common variants ($MAF > 0.01$). The results in this manuscript are based only on variants with $MAF < 0.01$ and have thus not been studied previously. Note that this section has been moved to the Supplementary Results (page 11f) in order to improve the focus on the main messages.

Regarding the second statement, we have clarified that our previous experimental work¹³ established loss-of-function as the mechanism of the tested *SLC26A1* variants, but that the lowest possible p-value from the aggregation of driver variants in *SLC26A1* was from this study (page 15, lines 344-347). Moreover, our previous work did not investigate the relationship of the experimentally confirmed loss-of-function alleles in *SLC26A1* with human height.

20. Page 15, related to allelic series: I am not sure if Pearson correlation is the best test for testing the genetic effect sizes on plasma sulfate levels in the GCKD study and sitting and standing height in the UKB and concluding that there is a “causal effect”. Perhaps use a causal model? This analysis is using driver

variants, so clarify why some analysis used all variants and others just the driver variants. The last paragraph on this page is a comment on a published study and it needs to be moved to the discussion.

Response: We would have preferred to use a causal model, but sulfate is currently not measured in the UKB. Our statement about causality was motivated by the fact that our observations are based on driver variants selected for an experimentally confirmed loss-of-function mechanism and the growth defect / lower size described in the two sulfate transporter knockout mouse models^{8,9}. We have now included a statement that once sulfate becomes available in the UKB, conditional analyses to statistically investigate causality should be performed (page 21, lines 484f).

The last paragraph contains results from analyses that we performed in our study, and compares our findings to those from a publication describing the phenotypic presentation of a completely unrelated individual homozygous for one of the variants we investigated. We believe that the citation of that study at this point is useful underscore that findings from heterozygous individuals relate to observations made upon full loss of gene function. We have now edited to clearly indicate which analyses were carried out as part of our study.

21. Page 17, the conclusion that “these findings provide convincing evidence that lower transcellular sulfate reabsorption is associated with numerous adverse musculoskeletal traits and diseases” is an overstatement as the evidence is based on associations and not functional assays.

Response: We have rephrased the sentence as “these findings provide strong support that genetic variants that proxy lower transcellular sulfate reabsorption are associated with human height, as well as with several musculoskeletal traits and diseases” (page 17, lines 404-406). We believe that this statement is justified, as the evaluated variants have been confirmed by functional assays previously.

22. Page 17, discussion. The sentences related to “heterozygous variants identified in a population sample permit insights into graded effects of impaired gene function without the need to identify patients with a corresponding biallelic IEM” is an overstatement of the paper findings, given the gene-based results were not replicated and the study is using relative levels of metabolites. Therefore, except for the one KYNU patient results that used absolute metabolite quantification, inference related to disease do not apply.

Response: Our statement refers to the graded effects of impaired gene function between heterozygous carriers in population-based studies and (effectively) homozygous individuals (hemizygous men for *TMLHE*, and the newly added Supplementary Figure 9 showing more extreme metabolite values for the

few instances, in which more than one individual homozygous for qualifying variants in a given gene was observed in comparison to heterozygous individuals (compare Reviewer 2, comment 5)). These comparisons also hold semi-quantitative data. Further, the statement is supported by the observation that many of the heterozygous QVs are known causative mutations for known IEMs when present in the homozygous state (see ClinVar entries in columns AE-AG in Supplementary Tables 6a and 6b). Lastly, as detailed in response to point 2 above, we have now added several complementary and independent lines of evidence to replicate or validate findings from our study. We therefore believe that our original sentence is justified.

23. I may have missed this but the “impaired epithelial transcellular sulfate reabsorption” was not tested in assays and is based on association findings with phenotypes. I suggest rephrase this statement.

Response: We have modified our wording to clarify that we are referring to genetically inferred impaired epithelial transcellular sulfate reabsorption (page 20, lines 469-471).

Methods:

24. Report number of missing metabolites in serum and urine and how missing was handled.

Response: There is no information on missing metabolites provided by Metabolon, because the quantification is not based on a fixed metabolite panel, but rather compares measured spectra to a very large reference library, the content of which is not disclosed. We have however included a comprehensive Supplementary Table 2 that shows, for each metabolite returned, whether it was quantified in plasma, urine, or both. Moreover, we have analyzed only metabolite levels above the level of detection of each metabolite and with information from at least 300 individuals, with the corresponding sample size clearly indicated in column T of Supplementary Table 3 and as described on page 34 of the Methods.

25. GFR was estimated using the race-based equation, which is no longer used. The updated 2021 equation should be used instead.

Response: Since all participants of the GCKD study are of European ancestry, the race term in the equation was effectively not applied.

26. Add references for the need for adjustments for serum albumin and urine albumin for serum and urine metabolite analyses.

Response: We have now added a reference to our previous analysis of common variant associations with metabolites with corresponding adjustment variables² which were chosen again to facilitate comparisons of common and rare variant effects.

27. A sensitive analysis excluding participants with low GFR should be done to confirm the main findings, as associations may be driven by chronic kidney disease.

Response: The Reviewer raises a valid point, which we have addressed analytically by performing stratified analyses among participants with low and high eGFR. Effect sizes were similar and highly correlated (Pearson correlation 0.97) across both groups (new Suppl. Figure 4a). For more details, see Reviewer 1, point 2 and Reviewer 3, point 1.

28. Which threshold for significance was determined in the single rare variant analysis?

Response: The primary discovery analysis in our manuscript was a gene-based analysis, to maximize power to detect significant metabolite-gene associations. For each gene with a significant association with at least one metabolite ($P\text{-value} < 5e-9$), we subsequently tested each qualifying variant for association with the corresponding metabolite(s) in single-variant analyses. These single-variant results were not filtered by p-value, because we wanted to characterize and compare the effect sizes and p-values of all qualifying variants, and because variants with a MAC of 1 are not expected to have very low individual p-values.

Our strategy is supported by observations reported for rare variant associations with plasma protein levels in the UK Biobank¹², where only a minority of significant single variant associations pointed towards genes that were not also found in gene-based tests, whereas 26% (across all tested models) resp. 62% (for the pvt model, similar to our masks) of the gene-protein associations identified with gene-based tests were not found in single variant analyses.

29. Results for conditional analysis at genes need to be included to sort out if the associations with common variants explain the associations with rare variants.

Response: Thank you for this suggestion, which we have now addressed analytically. The results of the conditional analyses as well as measurements for assessing differences between both un- and conditional analyses are added to Supplementary Table 7 (previously ST6). Moreover, the highly correlated effect sizes for both un- and conditional analyses (Pearson correlation 1) are visualized in the new Supplementary Figure 8, panel b. There was no gene-metabolite association for which conditioning on the associated common variant within the gene region resulted in significantly changed effect sizes. These new observations are included in Results (page 10f, lines 242-245) as well as in methods (page 38, lines 833-840).

Figures:

30. Figure 1. change “Novel genes” to “novel gene associations”.

Response: With the term “novel genes” we wanted to describe genes that were not reported as associated with any metabolite in the previous studies, in order to distinguish them from “novel gene-metabolite associations”, where the gene-metabolite combination was not reported previously. As changing “novel genes” to “novel gene associations” could lead to confusions with the term “novel gene-metabolite associations”, we would like to retain our original wording.

31. Figure 2a circus plot is hard to read the metabolite names and the color code is too complex. I find this type of graph unhelpful as they mask important details. Consider simplifying it.

Response: We chose the color coding in the plot with the intent to convey precisely the information that the Reviewers have requested in several other comments: the information whether associations were known or novel (comment 3 above, Reviewer 1, comment 4) is encoded in the color of the gene:metabolite labels. All genes and metabolites are of course also provided in Supplementary Table 3 for magnification. The color of the circles corresponds to the matrix (plasma red, urine blue), and the shade of the color to the effect direction (dark: positive, light: negative). This permits an easy comparison of whether associations were unique to plasma or urine or shared, and if shared, whether effect directions are concordant. The metabolite classes in the inner circle are informative because they convey, for instance, that many associations uniquely detected in plasma arise from metabolites that belong to the lipid pathway. We would therefore prefer not to simplify the figure, as we believe that these aspects contain important information.

32. Figure 2b, it is not very helpful to show broad categories of metabolites and you could move this to a supplemental material.

Response: We believe that the plot is informative in that it gives an overview of two points the Reviewer was raising above, i.e., the lower number of lipid associations detected in urine and how often plasma and urine associations are unique. In addition, it also compares the number of findings across masks, and how many findings are unique to each mask and matrix. We have modified the figure to make it more aesthetic and, hopefully, have succeeded in emphasizing these points.

33. Figure 3, remove statement about the enzyme function as this was not measured in the study. Measured metabolites are relative levels and not absolute levels so one cannot infer if they are within normal or abnormal range.

Response: We have made several adjustments to Figure 3 that also include modifications based on other comments that this Reviewer made, including the presentation of adjusted rather than unadjusted metabolite levels (see comment 37), the inclusion of association p-values for driver variants (comment 16), and an improvement of the legend. We included a schematic representation about the function of the enzyme in the figure, which is known based on a broad body of literature and biochemical experiments, because we believe it is helpful for readers to emphasize that indeed the levels of the substrate are higher in carriers of presumed loss-of-function variants compared to non-carriers, indicative of lower turnover, and vice versa for the produced metabolite. To the best of our knowledge, we did not make any statements about metabolite levels being in the normal or abnormal range, just compared their levels relative to each other (e.g., higher in carriers compared to non-carriers), which can be done with semi-quantitative levels. We have addressed the Reviewer's point in the figure legend, clearly stating that the known enzyme function is a conceptual addition to the figure and was not tested again in this study.

34. Figure 5a, this type of graph is usually shown when there is a functional experiment performed, which does not seem to be the case and can mislead the interpretation of results. I suggest that this is removed. Figure 5c, label that results are from the UK Biobank.

Response: Panel a is a conceptual figure to illustrate how these two transport proteins work together in a common pathway, transcellular sulfate reabsorption. We have now clearly indicated in the figure legend

that panel a) is a conceptual model. As the Reviewer suggests, we have now included in Figure 5c that these results are based on the UKB.

35. Figure 6a, label for NaS1 p.R272C has two het categories. Fig 6b, add the number of participants that had the NaS1 p.Arg272Cys variant and an outcome. Is this the “Number of QV carriers with disease”? it would be better to show the estimates and 95% confidence intervals and N cases/controls for these results as the display is confusing.

Response: Thank you for pointing out that the labeling in Figure 8a (previously panel 6a) may cause confusion. We have now clarified that there are two boxplots for individuals that are heterozygous for NaS1 p.R272C, as we separated them additionally by carrier status of SAT1 p.L348P. This underscores that carriers of LoF alleles in both transporters tend to have lower plasma sulfate levels than heterozygous carriers of NaS1 p.R272C only, as shown in the legend of Figure 8a. In panel 8b, we have added 95% confidence intervals, as suggested by the Reviewer. We have also added the absolute number of carriers with a given disease for clarity. The numbers of individuals in the remaining three fields of the corresponding 2x2 table (carriers without disease, non-carriers with disease, non-carriers without disease), as well as additional information, is provided in Supplementary Table 15.

Supplementary material:

36. Table S1. Add % chronic kidney disease based on GFR and urine albumin.

Response: We have added the proportion of individuals with eGFR 60 ml/min/1.73m² and with UACR >300 mg/g to Supplementary Table 1.

37. Figures S1 and S2. The statement citing these figures says that most of the variants were observed in heterozygous state, which is not shown. The significant gene-metabolite associations are from collapsing all variants within the gene, but the figures include each allele, and there is no way to know if a participant has multiple alleles. Although the overall mean metabolite level between groups is different, there is a large overlap in the distribution of the metabolite, and results are likely unadjusted. These figures need to be revised.

Response: We have revised Supplementary Figures 1 and 2 (now Supplementary Figure 5) as suggested: we have plotted metabolite levels that were adjusted for the same covariates as used in the main

discovery analyses. Individuals homozygous for a QV are colored in orange, whereas individuals heterozygous for a QV are colored in gray or blue depending on the driver status of the QV. Individuals carrying multiple QVs are depicted with an asterisk, whereas the symbol shape among individuals carrying just a single QV within the gene is based on variant consequence. Lastly, we have now included the panels for each significant gene-metabolite relationship in both plasma and urine as well as for both masks on one page, to facilitate their comparison, as suggested in comment 11 above.

38. For Figure S5, provide the comparison for the effect estimates or p-values for driver/non-driver variants by allele count as I suspect some of the associations are driven by a very low allele count.

Response: We appreciate this helpful suggestion and have added a new panel c to Supplementary Figure 7 (previously SFig. 5; mentioned on page 10, lines 234-236), which shows the effect estimates for driver/non-driver variants by minor allele count bins. In each MAC bin, median absolute effect estimates of driver variants are clearly larger than those of non-driver variants, even for the most common MAC bin.

39. Figure S6, include the definition of shared and unique associations. A conditional analysis likely would be a better approach to sort out this. This figure does not provide insights on results at the genomic regions including the relation between the absolute aggregated effect size of rare variants with the presence of a GWAS signal in the region, as stated in the main text.

Response: We agree with the Reviewer that conditional analyses are a valuable addition. We have now performed the conditional analyses suggested in comment 29 above, and updated Suppl. Figure 8 (previously SFig. 6) to include a new panel b) showing that adjustment for common variants does not alter the aggregated rare variant association signals. We have included a description of the conditional analyses in the Methods (page 38, lines 833-840), and refer to the new analyses in the Results section (page 10f, lines 242-245).

Data availability

40. "Further data is available on personal request." Is not appropriate and authors should include link to the genotype, metabolite and phenotype data for the GCKD and UK Biobank.

Response: We have included specific links, and also updated the "code availability" section with specific links to relevant GitHub repositories.

41. A role of pharma on this study needs to be clarified given funding source of genotype and metabolomics. Conflicts of interest will need to be updated based on these clarifications.

Response: As stated in the manuscript, pharma supported the generation of the genetic and the metabolomics datasets used in this study, but the companies were not involved in this project. GCKD investigators obtained the generated datasets, and are free in their scientific use without the need for industry approval. Therefore, no conflicts of interest were declared. During the revision work of the article, we initiated a collaboration with Maze Therapeutics, which contributed the newly added experimental data showing that methionine sulfone is a new substrate of SLC6A19. Maze investigators contribute in this study as scientists. We have made the link to Maze clear in the affiliations and the updated conflicts of interest section.

References

1. Schlosser, P. *et al.* Genetic studies of urinary metabolites illuminate mechanisms of detoxification and excretion in humans. *Nat. Genet.* **52**, 167–176 (2020).
2. Schlosser, P. *et al.* Genetic studies of paired metabolomes reveal enzymatic and transport processes at the interface of plasma and urine. *Nat. Genet.* **55**, 995–1008 (2023).
3. Bomba, L. *et al.* Whole-exome sequencing identifies rare genetic variants associated with human plasma metabolites. *Am. J. Hum. Genet.* **109**, 1038–1054 (2022).
4. Feofanova, E. V. *et al.* Whole-Genome Sequencing Analysis of Human Metabolome in Multi-Ethnic Populations. *Nat. Commun.* **14**, 3111 (2023).
5. Yousri, N. A. *et al.* Whole-exome sequencing identifies common and rare variant metabolic QTLs in a Middle Eastern population. *Nat. Commun.* **9**, 333 (2018).
6. Yu, B. *et al.* Loss-of-function variants influence the human serum metabolome. *Sci. Adv.* **2**, e1600800 (2016).
7. Wang, Q. *et al.* Rare variant contribution to human disease in 281,104 UK Biobank exomes. *Nature* **597**, 527–532 (2021).
8. Dawson, P. A., Beck, L. & Markovich, D. Hyposulfatemia, growth retardation, reduced fertility, and seizures in mice lacking a functional NaSi-1 gene. *Proc. Natl. Acad. Sci. U. S. A.* **100**, 13704–13709 (2003).
9. Whittamore, J. M., Stephens, C. E. & Hatch, M. Absence of the sulfate transporter SAT-1 has no impact on oxalate handling by mouse intestine and does not cause hyperoxaluria or hyperoxalemia. *Am. J. Physiol. Gastrointest. Liver Physiol.* **316**, G82–G94 (2019).
10. Eales, J. M. *et al.* Uncovering genetic mechanisms of hypertension through multi-omic analysis of the kidney. *Nat. Genet.* **53**, 630–637 (2021).
11. Tomaszewski, M. *et al.* Kidney omics in hypertension: from statistical associations to biological mechanisms and clinical applications. *Kidney Int.* **102**, 492–505 (2022).
12. Dhindsa, R. S. *et al.* Rare variant associations with plasma protein levels in the UK Biobank. *Nature* **622**, 339–347 (2023).
13. Pfau, A. *et al.* *SLC26A1* is a major determinant of sulfate homeostasis in humans. *J. Clin. Invest.* **133**, (2023).
14. Lee, S., Dawson, P. A., Hewavitharana, A. K., Shaw, P. N. & Markovich, D. Disruption of NaS1 sulfate transport function in mice leads to enhanced acetaminophen-induced hepatotoxicity. *Hepatology. Baltim. Md* **43**, 1241–1247 (2006).
15. Tise, C. G. *et al.* From Genotype to Phenotype: Nonsense Variants in *SLC13A1* Are Associated with Decreased Serum Sulfate and Increased Serum Aminotransferases. *G3 Bethesda Md* **6**, 2909–2918 (2016).

16. Ao, X. *et al.* Rare variant analyses in large-scale cohorts identified SLC13A1 associated with chronic pain. *Pain* **164**, 1841–1851 (2023).
17. Bjornsdottir, G. *et al.* Rare SLC13A1 variants associate with intervertebral disc disorder highlighting role of sulfate in disc pathology. *Nat. Commun.* **13**, 634 (2022).
18. Povysil, G. *et al.* Rare-variant collapsing analyses for complex traits: guidelines and applications. *Nat. Rev. Genet.* **20**, 747–759 (2019).
19. Cheng, Y. *et al.* Rare genetic variants affecting urine metabolite levels link population variation to inborn errors of metabolism. *Nat. Commun.* **12**, 964 (2021).
20. Nag, A. *et al.* Effects of protein-coding variants on blood metabolite measurements and clinical biomarkers in the UK Biobank. *Am. J. Hum. Genet.* **110**, 487–498 (2023).
21. Thiele, I. *et al.* Personalized whole-body models integrate metabolism, physiology, and the gut microbiome. *Mol. Syst. Biol.* **16**, e8982 (2020).
22. Brunk, E. *et al.* Recon3D enables a three-dimensional view of gene variation in human metabolism. *Nat. Biotechnol.* **36**, 272–281 (2018).

Decision Letter, first revision:

16th Apr 2024

Dear Professor Köttgen,

Your Article, "Coupling of metabolomics and exome sequencing reveals graded effects of rare damaging heterozygous variants on gene function and human traits and diseases" has now been seen by 3 referees. You will see from their comments below that while they find your work of interest, some important points are raised. We are interested in the possibility of publishing your study in Nature Genetics, but would like to consider your response to these concerns in the form of a revised manuscript before we make a final decision on publication.

We therefore invite you to revise your manuscript taking into account all reviewer and editor comments. Please highlight all changes in the manuscript text file. At this stage we will need you to upload a copy of the manuscript in MS Word .docx or similar editable format.

*2) If you have not done so already please begin to revise your manuscript so that it conforms to our Article format instructions, available here.

*3) Include a revised version of any required Reporting Summary:

Please be aware of our guidelines on digital image standards.

[redacted]

We hope to receive your revised manuscript within 2 to 3 months. If you cannot send it within this time, please let us know.

Sincerely,
Wei

Wei Li, PhD
Senior Editor
Nature Genetics
New York, NY 10004, USA
www.nature.com/ng

Reviewers' Comments:

Reviewer #1:

Remarks to the Author:

Overall, the authors have done extensive work to try and address our comments (and also, that of the other reviewers). A few remaining comments following the author responses.

- Major comment no. 3: For the significant gene-metabolite relationships reported in the study, the authors have shown quite convincingly that adjustment for medication use has minimal impact on the associations (Reviewer Fig. 1). However, I would encourage the authors to conduct a separate genome-wide analysis adjusting for medication use might help the authors detect additional gene-metabolite associations that were not significant in their original unadjusted analysis.
- Major comment no. 4: It is still remarkable that such a high proportion of the findings in this study are novel despite for the plasma metabolites the majority have been analysed in one of the previous studies. The authors cite low sample size / power of previous studies as one possible explanation, but a quick look at couple of the studies cited suggests that they had comparable / greater sample size (ref no 3: N~4K & ref no 4: N~12K) compared to this study (N=4,737). The authors have tried to address by conducting rigorous comparisons of effect sizes between their study and previous studies to demonstrate that when the signals were observed in both then they are correlated. Some commentary on the likely source of the differences would be reassuring for the readership.
- Minor comment no. 6: It is well established that there is greater statistical power for quantitative than binary regression tests. For genetic signals with a significant metabolite relationship, the authors should relax the significance threshold for clinical endpoints (leveraging the prior from the strength of the quantitative trait genetic signal) to see if an enrichment analysis would then be feasible? It remains noteworthy that some of the metabolite-associated genes highlighted as vignettes (e.g., SLC7A9 / SLC6A19) influence only renal metabolite / biomarker levels (correlative signals) rather than being kidney disease (driver) genes. As per Suppl. Tables 13 and 16 that the authors have now added, these genes don't show any association with kidney disease endpoints at $p < 1 \times 10^{-5}$. This remains the key outstanding clarification in the article as genetic signals observed to have large effects on human biomarker levels can often lack clear clinical/therapeutic effects. So, distinguishing metabolite/biomarker genetic signals from human disease drivers will enhance the robustness of the article.

Reviewer #2:

Remarks to the Author:

Thank you very much for conducting additional experiments and providing responses to my comments - I am satisfied that my suggestions have been largely addressed.

Reviewer #3:

Remarks to the Author:

Changes in the reviewed manuscript are not provided, so they are not easily verified. Some issues remained.

I am not comfortable with the general statements provided in the abstract which suggest that functional studies were done for several metabolite-gene associations in relation to inborn errors of metabolism. It would be important if authors provide an objective description of results including specific data instead of general overview. This kind of statement without details can add to the confusion in the field on what was studied. The last sentence "We present a powerful approach to identify new players in incompletely characterized human metabolic reactions, and to reveal metabolic readouts of disease risk to inform disease prevention and treatment" is a high overstatement of the paper results.

Page 116: "a novel computational method based on WBMs". In the answer to reviewer 3 (question17), the method was based on ref 22, which is not listed in this sentence.

The authors stated that they did not refer to absolute values for relative quantities of metabolites, but paragraph starting in page 265 description can be misinterpreted as clinically relevant absolute low values. Example, line 269: "In plasma, male QV carriers showed 1.15 standard deviations (SD) lower levels of plasma 270 hydroxy-N6,N6,N6-trimethyllysine as compared to non-carriers (P-value=6e-44)". Line 273: "Levels were higher in women than men, suggesting that X-23436 is a metabolite downstream of the reaction catalyzed by the encoded regucalcin".

Last sentence in the conclusion is an overstatement and needs to be reworded: In conclusion, the exome-wide study of rare, putative loss-of-function variants can establish causal relationships with metabolites, and highlight metabolic biomarkers that reflect the degree of impaired gene function and result in graded, adverse effects on human health. "

Answer to R1, question 4, it seems that most of the blood-based metabolites were previously tested in one of 4 published studies, but only significant in this study. This is not reassuring, and one cannot rule out false positive given the GCKD study has also limited power for rare variants, even in aggregates. Did you consider including power analysis for this study?

I am not convinced that the new analyses constitute real validation. For example, it is surprising that the gene-metabolite analysis shows similar effects sizes in the context of differences in covariant selection, units and so on compared with results from Bomba et al. The conclusion that genetic effects on metabolite levels do not differ between individuals with and without reduced kidney function should not be based only on correlation agreement of estimates with validation data. It would be important to know the distribution of metabolites across strata, the number of heterozygous variants contributing to the test and report a formal test of interaction.

Reviewer Figure 2 shows a very large effect size for associations, and it would be of interest to know the variance of these estimates. This is related to the statement that QVs are only detected in one or two carriers, so the estimates are likely unstable.

Answer to R3

Question 1. Sensitivity analysis across eGFR categories highlighted the effect size only and did not include the distribution of the metabolites across these strata or the number of variants that contributed to analyses. Where are the results for the analyses comparing to an eGFR>60?

Question 2. Thank you for the additional analyses. However, it would be important to know specifically which associations were validated and which were not, instead of providing some proof of concept examples. This is important for future studies to replicate findings. Clearly point to the table with these results.

Question 4. Reviewer Figure 4 labels are not readable. Based on colors, it seems that there are two clusters of metabolites that are positive correlated, and you could just plot them.

Questions 6 and 22. I am concerned that the conclusions related to IEM variants can be interpreted as disease-related variants in participants who do not clearly have IEM disease. The associations are not related to absolute values of metabolites, so there is no way to establish thresholds, and relative changes may be within the normal range. Statements related to IEM should be clearly discussed as hypothesis. See also concerns related to abstract and conclusion sentences.

Question 25. The authors should be aware that the new race-free equation has also change the calculation for GFR in individuals of European ancestry and therefore, the calculated eGFR is not the same as per equation used in this study.

Author Rebuttal, first revision:

Dear Reviewers and Editors,

We would like to thank you for the thorough review of our revised work, and the additional constructive suggestions. We have addressed each of these as outlined below. The location of the changes refers to the PDF version of our revised manuscript that contains tracked changes. Please note that we have not tracked changes related to revisions of grammar and style, and to renumbering of display items due to additions or shifts. We hope that these changes meet your expectations, and are looking forward to hear from you.

Best wishes,

Nora Scherer, Daniel Fässler, Johannes Hertel, and Anna Köttgen

Reviewers' Comments:**Reviewer #1:***Remarks to the Author:*

Overall, the authors have done extensive work to try and address our comments (and also, that of the other reviewers). A few remaining comments following the author responses.

Major comment no. 3: For the significant gene-metabolite relationships reported in the study, the authors have shown quite convincingly that adjustment for medication use has minimal impact on the associations (Reviewer Fig. 1). However, I would encourage the authors to conduct a separate genome-wide analysis adjusting for medication use might help the authors detect additional gene-metabolite associations that were not significant in their original unadjusted analysis.

Response: We agree that these are interesting analyses, and have now performed separate genome-wide analyses for both urine and plasma metabolites and both masks, adjusting for the use of medications commonly prescribed in CKD as detailed in our previous rebuttal letter (ACE inhibitors, ARBs, beta blockers, calcium channel blockers, anti-diabetic medications, diuretics, and lipid-lowering medications). The results showed very little differences compared to our main analysis that did not adjust for medication use. Not unexpectedly, very few (seven of 235 gene-metabolite associations detected in plasma or urine using both masks) associations no longer met the significance threshold when adjusting for medication use. However, as shown in **Reviewer Figure 1**, their association p-values were still very significant and barely missed the significance threshold. Conversely, there were 11 associations that were significant only when adjusting for medication use, but all of their association p-values were close to the significance threshold in our main analysis. Moreover, the effect sizes of associations that were only significant in the main or in the medication-adjusted analysis were very similar and highly correlated (**Reviewer Figure 1**), and statistical tests for differences in effect sizes all had p-values >0.66. Because the manuscript contains a lot of information already and the new analyses did not indicate strong differences when adjusting for medication, we decided not to add these analyses to the manuscript.

Reviewer Figure 1: Comparison of effect sizes (a) and $-\log_{10}(\text{P-values})$ (b) of gene-metabolite associations detected in the main analysis with those found in the analysis additionally adjusting for the use of commonly prescribed medications. Colors indicate the analysis in which an association met the significance threshold.

Major comment no. 4: It is still remarkable that such a high proportion of the findings in this study are novel despite for the plasma metabolites the majority have been analysed in one of the previous studies. The authors cite low sample size / power of previous studies as one possible explanation, but a quick look at couple of the studies cited suggests that they had comparable / greater sample size (ref no 3: $N \sim 4K$ & ref no 4: $N \sim 12K$) compared to this study ($N=4,737$). The authors have tried to address by conducting rigorous comparisons of effect sizes between their study and previous studies to demonstrate that when the signals were observed in both then they are correlated. Some commentary on the likely source of the differences would be reassuring for the readership.

Response: We thank the Reviewer for raising this point again, and have used the opportunity to provide more clarifications. For all four previous studies that performed sequencing-based rare variant analyses of metabolite levels quantified with the same technique as ours, we included all

reported results from the analysis of rare exonic variants at the gene-level as well as at the single-variant-level (if reported) into our comparison.

Differences to the study from Bomba *et al* (ref. no. 3 in the Reviewer's comment above) largely arise from differences in the selection of qualifying variants to define the tested masks / windows, resulting in the presence of fewer putatively damaging variants and hence lower power to detect associations as compared to our study. We have now included a new **Supplementary Table 8** that provides details on the comparisons and shows that especially for gene-based tests that did not achieve significance in the study by Bomba *et al*, fewer variants were aggregated as compared to our study. This is consistent with their approach to test windows, which often did not span the entire gene.

Differences to the study from Feofanova *et al* (ref. no. 4 in the Reviewer's comment above) largely arise from their staged study design. Specifically, Feofanova and colleagues only performed gene-based tests for those 230 metabolites that were involved in significant single-variant-associations annotated as novel, resulting in less metabolites evaluated at the better-powered gene-based level as compared with our study. To be as inclusive as possible, we have revised our comparisons to also consider associations reported by Feofanova *et al* that were based on single exonic rare variants. As a result, we find that our study detects 88 newly reported gene-metabolite associations in plasma as compared to previous studies (before: 93), and that 82 of these 88 associations involved metabolites analyzed by one of the previous four studies. We have updated these numbers in the results section of the manuscript (page 7, lines 157, 159, 162), and added the need to consider differences in study design, QV selection, and statistical tests when comparing our results to those of previous studies of the plasma metabolome to the limitations section of the discussion (page 21, lines 493-495). We have included a reference to the new **Supplementary Table 8** in the manuscript (page 8, line 190).

Minor comment no. 6: It is well established that there is greater statistical power for quantitative than binary regression tests. For genetic signals with a significant metabolite relationship, the authors should relax the significance threshold for clinical endpoints (leveraging the prior from the strength of the quantitative trait genetic signal) to see if an enrichment analysis would then be feasible? It remains noteworthy that some of the metabolite-associated genes highlighted as vignettes (e.g., SLC7A9 / SLC6A19) influence only renal metabolite / biomarker levels (correlative signals) rather than being kidney disease (driver) genes. As per Suppl. Tables 13 and 16 that the authors have now added, these genes don't show any association with kidney disease endpoints at $p < 1 \times 10^{-5}$. This remains the key outstanding clarification in the article as genetic signals observed to have large effects on human biomarker levels can

often lack clear clinical/therapeutic effects. So, distinguishing metabolite/biomarker genetic signals from human disease drivers will enhance the robustness of the article.

Response: Following the Reviewer's comment, we have repeated the enrichment analyses based on associations with any binary trait at p-value $<1e-5$ in the UK Biobank. Based on 16,291 genes assessed for gene-based rare variant associations in both the UK Biobank and in our study, there were 3,745 associated with binary traits in the UKB at p-value $<1e-5$, of which 22 (out of 72) were also significantly associated with at least one metabolite in our study. The resulting enrichment odds ratio of 1.48 is not significant (Fisher test p-value=0.085). We conclude that genes that are strongly linked to altered metabolite levels by rare variant aggregation studies do not translate into enrichment for associations with the binary traits and diseases studied in the UK Biobank as a whole (at a significance level of p-value $<1e-5$). We note, however, that many of the binary traits and diseases are not expected to be caused by altered metabolite levels (or to even have a strong genetic component), and that individual significant associations with diseases, when present, can be informative even in the absence of enrichment across all diseases evaluated in the UK Biobank. We have added these results and a sentence along the lines suggested by the Reviewer to the corresponding part of the manuscript (Supplement page 12f).

Reviewer #2:

Remarks to the Author:

Thank you very much for conducting additional experiments and providing responses to my comments - I am satisfied that my suggestions have been largely addressed.

Response: We thank the Reviewer for the positive feedback.

Reviewer #3:

Remarks to the Author:

Changes in the reviewed manuscript are not provided, so they are not easily verified. Some issues remained.

1. I am not comfortable with the general statements provided in the abstract which suggest that functional studies were done for several metabolite-gene associations in relation to inborn errors of metabolism. It would be important if authors provide an objective description of results including specific data instead of general overview. This kind of statement without details can add to the confusion in the field on what was studied. The last sentence “We present a powerful approach to identify new players in incompletely characterized human metabolic reactions, and to reveal metabolic readouts of disease risk to inform disease prevention and treatment” is a high overstatement of the paper results.

Response: We have rephrased the abstract to explicitly clarify that experimental proof-of-principle validation was performed for one finding, and modified the last sentence.

2. Page 116: “a novel computational method based on WBMs”. In the answer to reviewer 3 (question17), the method was based on ref 22, which is not listed in this sentence.

Response: Both manuscripts (Thiele 2020 and RECON3D) were cited in the Methods part of our manuscript. We have now added both papers as references again at the position the Reviewer points out, page 5, line 118.

3. The authors stated that they did not refer to absolute values for relative quantities of metabolites, but paragraph starting in page 265 description can be misinterpreted as clinically relevant absolute low values. Example, line 269: “In plasma, male QV carriers showed 1.15 standard deviations (SD) lower levels of plasma 270 hydroxy-N6,N6,N6-trimethyllysine as compared to non-carriers (P-value=6e-44)”. Line 273: “Levels were higher in women than men, suggesting that X-23436 is a metabolite downstream of the reaction catalyzed by the encoded regucalcin”.

Response: The sentences pointed out by the Reviewer do not contain any absolute values, and the standard deviation units refer to inverse normal transformations of relative metabolite quantification and are valid as such. We therefore respectfully disagree that these sentences require changing. To ensure that there is no confusion about the interpretation of the mentioned standard deviations, we have added an explanatory sentence on page 35, lines 749-751, and

very clearly stated throughout the manuscript the few instances in which we refer to absolute metabolite levels. Moreover, we have now added explicitly to the limitations section on page 21, line 500f that we are analyzing semi-quantitative population metabolomics data that do not allow for conclusions whether metabolite levels are outside the clinical reference range.

4. Last sentence in the conclusion is an overstatement and needs to be reworded: In conclusion, the exome-wide study of rare, putative loss-of-function variants can establish causal relationships with metabolites, and highlight metabolic biomarkers that reflect the degree of impaired gene function and result in graded, adverse effects on human health. “

Response: We have reworded this sentence to tone down the conclusion.

5. Answer to R1, Question 4: it seems that most of the blood-based metabolites were previously tested in one of 4 published studies, but only significant in this study. This is not reassuring, and one cannot rule out false positive given the GCKD study has also limited power for rare variants, even in aggregates. Did you consider including power analysis for this study?

Response: We have used this comment, as well as a comment about the comparison to previous studies by Reviewer 1 (see comment 2 above), to clarify why most of the previously studied metabolites were not reported as significant in those studies in more detail. As outlined above, this can be largely attributed to differences in study design (Feofanova *et al* used a staged design and tested much fewer metabolites in gene-based tests) as well as in qualifying variant selection and aggregation unit (e.g., Bomba *et al* used windows as units, resulting in the aggregation of fewer qualifying variants and less power). We are now adding a new **Supplementary Table 8** showing the numbers of aggregated variants, the used masks and tests, so that readers can appreciate how differences arise.

Limited power increases type 2 error in studies (i.e., limits the ability to detect true associations), rather than increasing type 1 error (false positive findings). Moreover, the detected associations are overwhelmingly supported either through existing biological/biochemical knowledge about the connection of the gene to the metabolite, based on decades of research into the implicated reactions, and/or by known inborn errors of metabolism when biallelic variants are present. Moreover, the high correlation between genetic effect sizes from our study with those reported by Bomba *et al* further supports the validity of these associations, even when their association p-values did not achieve statistical significance in the Bomba study after correction for multiple

testing. We therefore do not believe that false positive findings are a major concern in our study. With regard to power analyses, these are commonly not performed for rare variant aggregation studies in the literature, because they require many assumptions (e.g., regarding effect size, variance explained by multiple variants within a locus, number of causal variants, and variant consequence) that cannot be generalized to all genes, since their individual genetic architecture differs with respect to the presence, number, and types of rare damaging variants.

6. I am not convinced that the new analyses constitute real validation. For example, it is surprising that the gene-metabolite analysis shows similar effects sizes in the context of differences in covariant selection, units and so on compared with results from Bomba *et al*. The conclusion that genetic effects on metabolite levels do not differ between individuals with and without reduced kidney function should not be based only on correlation agreement of estimates with validation data. It would be important to know the distribution of metabolites across strata, the number of heterozygous variants contributing to the test and report a formal test of interaction.

Response: Our findings related to kidney function were not only based on the comparison to the findings by Bomba *et al*, but also based on the eGFR-stratified analyses in the GCKD study. In addition to the new **Supplementary Table 8**, in which we now present the comparison of effect sizes to those of Bomba *et al* along with the numbers of aggregated variants, the used masks and tests (see above), we agree that some readers may be interested in genetic effect size comparisons between individuals with $eGFR \geq 45$ ml/min/1.73m². We have therefore now included a new **Supplementary Table 5** with these results mentioned on page 5, line 154, which also contains the number of heterozygous variants in each stratum as well as tests for effect size differences.

We do not consider it surprising that the genetic effect sizes are similar in the context of different covariates, including kidney function. In fact, this is precisely what one would expect if these covariates do not represent confounders. As most of the identified associations are based on connections between enzymes and their metabolite substrates or products, it is quite plausible that there is little confounding, collider bias, or reverse causation. As expected given the important role of the kidney in the clearance of metabolites from blood, the median of differences in plasma metabolite levels between individuals with $eGFR < 45$ ml/min/1.73m² compared to those with higher eGFR is positive (0.39 SD). This does not mean, however, that genetic effects on metabolite levels only occur in one of the two strata or are of different magnitude. In fact, our test for difference in effect size across strata, included in new **Supplementary Table 5**, shows that there are no significant difference in genetic effect sizes.

7. Reviewer Figure 2 shows a very large effect size for associations, and it would be of interest to know the variance of these estimates. This is related to the statement that QVs are only detected in one or two carriers, so the estimates are likely unstable.

Response: The previous Reviewer Figure 2 already included standard errors of the effect sizes to quantify the uncertainty of these estimates (left-hand part of the figure). We are therefore not entirely sure what the Reviewer is referring to – maybe the gray font used for the standard errors made them appear not very prominently. The figure shows that effect sizes are more stable for the masks used in our study than for the LoF-only mask, which we evaluated based on the first round of reviews. To provide estimates that are as stable as possible, we performed gene-based aggregation tests as the primary approach and only assessed genes with at least four qualifying variants. The right-hand part of the Figure presents p-values, so estimates of uncertainty do not apply.

8. Answer to R3, Question 1: Sensitivity analysis across eGFR categories highlighted the effect size only and did not include the distribution of the metabolites across these strata or the number of variants that contributed to analyses. Where are the results for the analyses comparing to an eGFR>60?

Response: As outlined in response to comment 6 above, we have now included a new **Supplementary Table 5** with these eGFR-stratified results, which also contains the number of variants in each stratum as well as tests for differences in effect sizes.

When stratifying eGFR at 60 ml/min/1.73m² instead of at 45, we obtain essentially the same findings. **Reviewer Figure 2** shows that the genetic effect sizes across strata are very highly correlated (correlation coefficient 0.94). Across all tested metabolites, significant differences were only detected for two out of all findings (see labels). These differences were driven by differences in qualifying variants in unequally sized groups (3660 vs. 960 individuals), such that very influential variants were only present in one stratum. This hinders systematic comparisons of the effects of aggregated variants across strata, which is why we chose to not perform stratified analyses in our original analysis plan. We believe that readers will benefit most from the comparison of equally sized groups obtained when stratifying at an eGFR of 45, as they are most robust. We therefore provide new **Supplementary Table 5** stratified at this cutoff.

Reviewer Figure 2: Comparison of effect sizes of gene-metabolite associations across individuals with eGFR below and above a cutoff of 45 (a) and 60 mL/min/1.73m² (b).

9. Answer to R3, Question 2: Thank you for the additional analyses. However, it would be important to know specifically which associations were validated and which were not, instead of providing some proof-of-concept examples. This is important for future studies to replicate findings. Clearly point to the table with these results.

Response: We agree that this is of interest and are now including a new **Supplementary Table 8** that contains comparisons to the study of Bomba *et al*, the only previous study that released also their non-significant findings on the aggregation level, for all associations significant in our study. Moreover, our previous Supplementary Table 5 (now **Supplementary Table 6**) contains columns AA-AC that can be filtered to show which metabolites were assessed by which of the

previous studies, whether a given gene was identified as significant, and whether this significant association occurred with the same metabolite as found in our study.

10. Answer to R3, Question 4: Reviewer Figure 4 labels are not readable. Based on colors, it seems that there are two clusters of metabolites that are positive correlated, and you could just plot them.

Response: As requested, we are including a new **Reviewer Figure 3**, in which we only plot the highly correlated (correlation coefficient >0.8) metabolites that include the two clusters shown in the previous plot and that are now more easily readable. Because our main point in the previous figure was the fact that most metabolites are not highly correlated, we had not focused on readability of the individual labels. As the Reviewer can see, the two main correlated clusters correspond to a group of phosphatidylethanolamine metabolites in plasma that are all associated with rare variants in *APOC3* (upper cluster), and to a group of medium-chain acyl carnitine metabolites in plasma and acyl glycine metabolites in urine, which are all associated with rare variants in *ACADM* (bottom cluster). This is biologically plausible given the role of *ACADM* (medium-chain specific acyl-CoA dehydrogenase) in mitochondrial fatty acid beta-oxidation, where it acts specifically on acyl-CoAs with saturated 6 to 12 carbons long primary chains (i.e., they all represent substrates). In conclusion, we do not believe that associations with correlated metabolites arise as a result of confounding, and we also believe that our multiple testing correction was appropriate, which was based on the estimated number of uncorrelated metabolites (see Methods).

Reviewer Figure 3: Correlation matrix for significant plasma or urine metabolites detected in our study that were correlated with at least one other metabolite with a Spearman correlation coefficient >0.8.

11. Answer to R3, Questions 6 and 22: I am concerned that the conclusions related to IEM variants can be interpreted as disease-related variants in participants who do not clearly have IEM disease. The associations are not related to absolute values of metabolites, so there is no way to establish thresholds, and relative changes may be within the normal range. Statements related to IEM should be clearly discussed as hypothesis. See also concerns related to abstract and conclusion sentences.

Response: To clearly emphasize that metabolite estimates from Metabolon are semi-quantitative (relative) measurements that do not allow for conclusions whether metabolite levels are outside the clinical reference range, we have added a new statement on page 21 to the manuscript as detailed in response to comment 3 above. Throughout the manuscript, we are careful not to imply that heterozygous variant carriers are affected by an inborn error of metabolism. We always refer to them as carriers and not as patients, and we explicitly state that heterozygous carriers of any variants causative for IEMs show only milder changes of the same or related metabolic phenotypes (page 4). We never refer to thresholds or abnormal levels outside the reference range, for any findings based on the GCKD study. The abstract and concluding sentences have been modified. Together, these precautions should avoid that readers may think that heterozygous variant carriers are affected by IEMs and show metabolite levels outside the reference range.

12. Answer to R3, Question 25: The authors should be aware that the new race-free equation has also change the calculation for GFR in individuals of European ancestry and therefore, the calculated eGFR is not the same as per equation used in this study.

Response: We are aware of this. Because all GCKD participants are of European ancestry and eGFR is not a confounder of gene-metabolite associations, we did not believe that updating the GFR estimating equation would have an appreciable effect on our results. However, to address this point empirically, we have now performed new sensitivity analyses adjusting all significant gene-metabolite relationships for GFR estimated by the new, race-free equation instead of the CKD-EPI equation. As shown in **Reviewer Figure 4**, the results are virtually identical: genetic effect size correlation is 0.9999996, and the P-value for effect size differences across both sets of analyses is >0.97 for all associations.

Reviewer Figure 4: Comparison of effect sizes of gene-metabolite associations based on adjustment using the race-free equation for the estimation of GFR and those based on the original analysis using the CKD-EPI equation.

Decision Letter, second revision:

18th Jul 2024

Dear Dr. Köttgen,

Thank you for submitting your revised manuscript "Coupling of metabolomics and exome sequencing reveals graded effects of rare damaging heterozygous variants on gene function and human traits and diseases" (NG-A63793R1). It has now been seen by the original referees and their comments are below. The reviewers find that the paper has improved in revision, and therefore we'll be happy in principle to publish it in Nature Genetics, pending minor revisions to comply with our editorial and formatting guidelines.

Sincerely,
Wei

Wei Li, PhD
Senior Editor
Nature Genetics
www.nature.com/ng

Reviewer #1 (Remarks to the Author):

I find that the authors have sufficiently addressed the comments from previous reviews.

Reviewer #3 (Remarks to the Author):

The authors addressed most of my concerns.

Final Decision Letter:

27th Sep 2024

Dear Dr. Köttgen,

I am delighted to say that your manuscript "Coupling metabolomics and exome sequencing reveals graded effects of rare damaging heterozygous variants on gene function and human traits" has been accepted for publication in an upcoming issue of Nature Genetics.

Your paper will be published online after we receive your corrections and will appear in print in the next available issue. You can find out your date of online publication by contacting the Nature Press Office (press@nature.com) after sending your e-proof corrections.

Please note that *Nature Genetics* is a Transformative Journal (TJ). Authors may publish their research with us through the traditional subscription access route or make their paper immediately open access through payment of an article-processing charge (APC). Authors will not be required to make a final decision about access to their article until it has been accepted. Find out more about Transformative Journals

Authors may need to take specific actions to achieve compliance with funder and institutional open access mandates. If your research is supported by a funder that requires immediate open access (e.g. according to Plan S principles) then you should select the gold OA route, and we will direct you to the compliant route where possible. For authors selecting the subscription publication route, the journal's standard licensing terms will need to be accepted, including <https://www.nature.com/nature-portfolio/editorial-policies/self-archiving-and-license-to-publish>. Those licensing terms will supersede any other terms that the author or any third party may assert apply to any version of the manuscript.

To assist our authors in disseminating their research to the broader community, our SharedIt initiative provides you with a unique shareable link that will allow anyone (with or without a subscription) to

read the published article. Recipients of the link with a subscription will also be able to download and print the PDF.

If you have not already done so, we strongly recommend that you upload the step-by-step protocols used in this manuscript to protocols.io. protocols.io is an open online resource that allows researchers to share their detailed experimental know-how. All uploaded protocols are made freely available and are assigned DOIs for ease of citation. Protocols can be linked to any publications in which they are used and will be linked to from your article. You can also establish a dedicated workspace to collect all your lab Protocols. By uploading your Protocols to protocols.io, you are enabling researchers to more readily reproduce or adapt the methodology you use, as well as increasing the visibility of your protocols and papers. Upload your Protocols at <https://protocols.io>. Further information can be found at <https://www.protocols.io/help/publish-articles>.

Sincerely,
Wei

Wei Li, PhD
Senior Editor
Nature Genetics
www.nature.com/ng